# A leading-edge based method for correction of slope-induced errors in ice-sheet heights derived from radar altimetry

Weiran Li [1], Cornelis Slobbe [1], and Stef Lhermitte [1]

[1]Department of Geoscience and Remote Sensing, Delft University of Technology, Delft, The Netherlands

**Correspondence:** Weiran Li (w.li-7@tudelft.nl)

**Abstract.** Satellite radar altimetry has been an important tool for cryospheric applications such as measuring ice-sheet height or assessing anomalies in snow and ice properties (e.g., the extensive melt in Greenland in 2012). Although accurate height measurements are key for such applications, slope-induced errors due to undulating topography within the kilometre-wide beam-limited footprint can cause multi-metre errors. Two main correction methods that have been developed (referred to as the slope- and point-based methods) either neglect the actual topography or the actual footprint that can be estimated by a combination of the leading edge and topography. Therefore, a Leading Edge Point-Based (LEPTA) method is presented that corrects for the slope-induced error by including the leading edge information of the radar waveform to determine the impact point. The principle of the method is that only the points on the ground that are within the range determined by the beginning and end of the leading edge are used to determine the impact point. Benchmarking of the LEPTA method against the slope- and point-based methods based on CryoSat-2 LRM acquisitions over Greenland in 2019 shows that, when compared to ICESat-2 observations, the LEPTA method has a stable performance both in the flat, interior regions of Greenland and in regions with more complex topography. The median difference between the slope-corrected CryoSat-2 heights using LEPTA and the ICESat-2 heights is at millimetre level, whereas the slope and point-based methods can respectively have a 0.21 m and 0.48 m difference, and the Level-2 data provided by ESA have a 0.01 m difference. The median absolute deviation of height differences between CryoSat-2 and ICESat-2, which we use as an indicator of the variation of errors, is also the lowest for LEPTA (0.09 m) in comparison to the aforementioned methods (0.19 m for slope method and 0.10 m for point-based method) and ESA Level-2 data (0.14 m). Although ESA Level-2 products and the point-based method have good performance in either median and median absolute deviation, LEPTA shows a good performance in both metrics. Based on that, we recommend considering LEPTA for obtaining accurate height measurements with radar altimetry data, especially towards the margins of the LRM coverage where the surface slopes increase.

## 1 Introduction

Satellite radar altimetry is a key tool for assessing the status and dynamics of the cryosphere as it allows constructing digital elevation models (DEMs) (Slater et al., 2018), deriving height change of ice sheets (Hurkmans et al., 2012; Helm et al., 2014a), understanding seasonal variations of snow (Adodo et al., 2018), and estimating snowpack properties (Lacroix et al., 2008). To obtain accurate information on heights, altimetry processing involves correction for instrument errors, atmospheric effects, tidal

effects, and slope-induced errors (Helm et al., 2014a; Hai et al., 2021). Of crucial importance is the correction for slope-induced errors as they can affect the obtained height measurements significantly. For example, according to the error propagation in Brenner et al. (1983), the CryoSat-2 satellite at an altitude of 717 km can give a vertical offset of approximately 39 m and a horizontal offset of 7.5 km when measuring heights of a terrain with a 0.6° slope,.

To correct for the slope-induced errors, different methods have been developed (Brenner et al., 1983; Remy et al., 1989; Bamber, 1994; Roemer et al., 2007). The most widely used methods involve both a correction to the height as well as a relocation of the satellite measurement location from nadir to the expected impact point (i.e., the radar reflection point) on the terrain. Two implementations of this so-called 'relocation method' are known as the 'slope' method and the 'point-based' method (Bamber, 1994; Roemer et al., 2007). The slope method assumes constant surface slope parameters within the beam-

limited altimeter footprint and calculates the relocated latitude, longitude, and height according to trigonometry (Levinsen et al., 2016). The point-based method uses a topographic model within the beam-limited satellite footprint and searches for minimum range between the satellite and a surface area in the size of the pulse-limited footprint (Roemer et al., 2007; Levinsen et al., 2016).

Although both methods have been refined and applied with reliable results, they both show methodological shortcomings.

The slope method, for example, tends to ignore the local topography within the footprint and therefore may not be accurate enough in undulating areas (Levinsen et al., 2016). The point-based method of Roemer et al. (2007), on the other hand, is more accurate in the undulating regions (Roemer et al., 2007; Levinsen et al., 2016) as it considers the detailed topography, but by assuming a fixed footprint size, it neglects the actual footprint illuminated by the satellite on the terrain. For example, by taking the averaged range within the assumed footprint, this method may ignore part of the terrain that actually contributes to the return

signal, or assumes that part of the terrain not visible to the satellite could contribute to the return signal (Fig. 1). The recent availability of high-resolution DEM products provides the opportunity to determine the part of the terrain contributing to the rise of the leading edge, and therefore can determine the actual footprint of the radar altimeter. To overcome the shortcomings of both methods, we present a Leading Edge Point-Based (LEPTA) method that exploits high-resolution DEM information to correct for the slope-induced error by including the leading edge information of the radar waveform to determine the impact

point. The principle of the method is that only the points on the ground that are actually within the range interval determined by the beginning and end of the leading edge are used to compute the impact point.

The paper is organised as follows. Section 2 describes the data used for radar altimetric processing and assessment of the results. In Section 3, the different methods used for the correction of the slope-induced errors as well as the assessment workflow are introduced. To assess the performance of the LEPTA method, we apply it to all CryoSat-2 LRM acquisitions

over Greenland in 2019 and benchmark it to the slope and point-based methods by comparing it with laser altimeter ICESat-2 height measurements. In Sections 4 and 5 we present, analyse, and discuss the results. Finally, we conclude by emphasising the main findings.

## 2 Data and pre-processing

### 2.1 CryoSat-2 observations

On the interior of the Greenland ice sheet, data acquired by CryoSat-2 are in Low Resolution Mode (LRM). LRM is the conventional pulse-limited mode that requires correction for slope-induced errors. The pulse-limited LRM footprint is approximately 1.65 km in diameter, and the beam-limited footprint is approximately 14.39 km in diameter (Hai et al., 2021). Our evaluation employs all data acquired from Jan. 1 to Dec. 31, 2019, resulting in approximately $2.2 \times 10^6$ measurements. In particular, we use level-1b (L1b) Baseline D data (European Space Agency, 2019a; Meloni et al., 2020).

To process the waveform information and obtain height estimations, the L1b waveforms are retracked using the offset centre of gravity (OCOG) method (Wingham et al., 1986) documented in Bamber (1994). We use OCOG because of its precision and robustness (Bamber, 1994; Schröder et al., 2019). According to Davis (1997), a 10% threshold is ideal for detecting ice-sheet height change (or strong volume scattering (Aublanc et al., 2018)), a 20% threshold is the most appropriate for estimating the absolute or true ice-sheet height, and a 50% threshold is the most appropriate for estimating the absolute height when the waveform is dominated by surface scattering (Davis, 1997; Aublanc et al., 2018). In this study, we follow the recommendation of Davis (1997) and use a 20% threshold to obtain estimates of the true ice-sheet elevation. This allows a comparison with ICESat-2 data. Aublanc et al. (2018), who used a 25% threshold, highlighted that this choice is a compromise between pure surface scattering (in which case the threshold should be around 50%) and volume scattering (10%). In the first case, one would underestimate the true elevation, and in the other overestimate it. Hence, as pointed out by Davis (1997), '*the 20% retracking point provides a reasonable estimate of the true ice-sheet elevation in only an average sense*'. In addition, waveforms are removed if they meet one of the following empirically derived criteria: i) the integrated normalised power exceeds 150, ii) the normalised power in the first 10 range bins is larger than 0.2, or iii) no peaks are identified in the waveform.

To benchmark our results, level-2I (L2I) height data obtained with the OCOG retracker from European Space Agency (2019b) are used. In the L2I products the slope-induced error is corrected with the Helm et al. (2014b) DEM, which has a resolution of $1 \times 1$ km (Helm et al., 2014a). To enable a fair comparison with our in-house processed L2 data, all L2I height measurements are removed for which the waveforms meet one of the criteria mentioned above.

### 2.2 ArcticDEM

To compute a correction for the slope-induced errors, a DEM is needed. Here, the slope method uses a low-resolution DEM as it assumes a constant slope within the pulse-limited footprint (Levinsen et al., 2016). On the contrary, the point-based methods (i.e., LEPTA and the point-based method proposed by Roemer et al. (2007)) require DEMs with higher resolution, to provide the full information of the local terrain.

In this study, ArcticDEM is used as reference DEM as it is constructed from recent stereo satellite imagery and is available in high resolution ($2 \times 2$ m) (Porter et al., 2018). The systematic error of ArcticDEM is less than 5 m (Noh and Howat, 2015) and the DEM has been updated since 2016. ArcticDEM is low-pass filtered to 2 km resolution by applying a block-mean filter for the slope-based method and to 100 m resolution for the point-based and LEPTA methods. The use of 100 m resolution instead

of 2 m is a compromise to computational efficiency. To assess the impact of DEM resolution on the correction methods, we vary the resolutions from 100 m (200 m for the slope method, for computational efficiency) to 900 m with a 100 m interval, and from 1 to 8 km with a 1 km interval.

## 2.3 ICESat-2 observations

For validation of the different slope correction methods, the ICESat-2 L3A Land Ice Height (ATL06) product (Smith et al., 2020a) is used. ICESat-2 uses the Advanced Topographic Laser Altimeter System (ATLAS) which emits green light pulses and counts the received photons (Abdalati et al., 2010). The laser beams are configured in a $2 \times 3$ array. The distance between and within beam pairs is $\sim 3.3$ km and $\sim 90$ m, respectively (Smith et al., 2019). The along-track resolution of the land ice height product is $\sim 20$ m (Smith et al., 2020b). The ATL06 products have a known geolocation accuracy (or bias) of less

than 10 m (National Snow and Ice Data Center (NSIDC), 2021). A comparison between ICESat-2 and ArcticDEM is shown in Appendix A. The results show that the median ICESat-2 height for the different beam pairs is up to 0.21 m higher than ArcticDEM. The median absolute deviation of the differences is 0.72 m for all beam pairs.

## 3 Methods

### 3.1 Slope correction methods

The different slope-induced error correction methods are conceptually illustrated in Fig. 1. The impact points estimated from the slope method, the point-based method, and LEPTA are represented by $P_s$, $P_p$ and $P_l$. The 'low-resolution DEM' (2 km) is only used by the slope method, whereas the point-based method and LEPTA use a 'high-resolution DEM' (100 m). The slope method computes a correction based on the surface slopes obtained from a DEM, whereas the point-based method and LEPTA are based on the range between the satellite and the terrain.

### 3.1.1 Slope correction method

The slope method uses the slope of the low-resolution DEM at the nadir point to compute the impact point. It assumes that the slope within the CryoSat-2 pulse-limited footprint is constant, and is defined by direction $\theta$ and magnitude $\Phi$ (Cooper, 1989; Bamber, 1994). In our implementation, $\theta$ and $\Phi$ are computed in the same map projection and grid as ArcticDEM. The gridded $\theta$ and $\Phi$ are then interpolated to the satellite nadir point. The corrected height ($h_C$), corresponding to the height of the impact point $P_s$, can then be obtained by (Bamber, 1994):

$$h_C = \frac{R_s \sin(\Phi - \Gamma)}{\sin \Phi} - R_\alpha,$$ (1)

where

$$\Gamma = \sin^{-1} \left( \frac{R \sin \Phi}{R_s} \right),$$ (2)

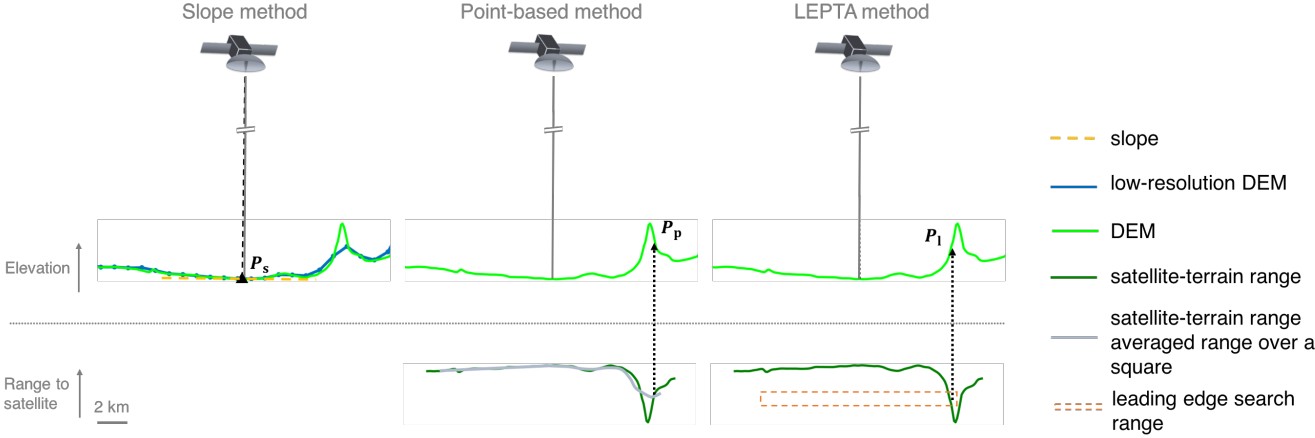

**Figure 1.** Conceptual illustration of different slope-induced error correction methods. The impact points estimated from the slope method, the point-based method, and LEPTA are represented by $P_s$, $P_p$ and $P_l$. The slope method computes a correction based on the surface slopes obtained from a DEM, whereas the point-based method and LEPTA are based on the range between the satellite and the terrain.

$$R_s = R_\alpha + h_S, \tag{3}$$

$$R_\alpha = \frac{\rho\nu}{\nu\cos^2\theta + \rho\sin^2\theta}, \tag{4}$$

$$\nu = \frac{a}{\sqrt{1 - e^2\sin^2\phi}}, \tag{5}$$

$$\rho = \frac{a(1 - e^2)}{\sqrt{(1 - e^2\sin^2\phi)^3}}, \tag{6}$$

$R$ represents the retracked range, $a$ and $e$ the semi-major axis and eccentricity of the reference ellipsoid being used, and $\phi$ is the latitude of the satellite. The corrected location of the impact point in latitude $\phi_C$ and longitude $\lambda_C$ (in radians) are computed as

$$\phi_C = \frac{\pi}{2} - 2\sin^{-1}\left(\frac{X + \Delta x}{2R_\alpha\cos\lambda_C}\right), \tag{7}$$

$$\lambda_C = \tan^{-1}\left(\frac{Y + \Delta y}{X + \Delta x}\right), \tag{8}$$

where $X$ and $Y$ define the position of the satellite in Cartesian coordinates,

$$\Delta x = R_\alpha \Gamma \cos\theta, \tag{9}$$

and

$$\Delta y = R_\alpha \Gamma \sin\theta. \tag{10}$$

Application of the slope-method in Fig. 1 shows that the impact point will be assumed at the position $P_s$. Inaccuracies usually occur when this method is applied to complex terrains, due to the simplification of the complex topography to a constant slope Levinsen et al. (2016).

### 3.1.2 Point-based correction method

The point-based method directly uses the topographic information from the a priori DEM to find the impact point ($P_p$). It does so by minimising the mean distance $\bar{R}_P$ to the satellite over a pre-defined fixed-size rectangular footprint area (e.g., $1.65 \times 1.65$ km in Hai et al. (2021)). Assuming the pre-defined rectangular footprint with area $A$ consists of $n$ DEM grid cells, $\bar{R}_P$ is computed by (Roemer et al., 2007):

$$\bar{R}_P = \frac{1}{A} \sum_{j=1}^{n} A_{Pj} \bar{R}_{Pj}, \tag{11}$$

where $A_{Pj}$ and $\bar{R}_{Pj}$ are the area of and range to each grid cell $j$. The point for which $\bar{R}_P$ is minimal is referred to as $P_p$ with latitude $\phi_c$ and longitude $\lambda_c$. The range between the satellite and $P_p$ is referred to as $r_p$. In line with Roemer et al. (2007), we use the 100 m DEM to find an approximate position. The final point is obtained by a second search in the vicinity of the approximate position for which we use an up-sampled DEM of $10 \times 10$ m. The corrected height $h_C$ is computed as (Roemer et al., 2007)

$$h_C = h_N + r_p - (h_S - h_I), \tag{12}$$

where $h_N$ is the surface height of the nadir point relative to the reference ellipsoid (i.e., the ellipsoidal height of the satellite $h_S$ minus the retracked range $R$), and $h_I$ is the DEM height of $P_p$. Eq. (11) also shows, however, that this approach can take DEM points into account that actually do not contribute to the rise of the leading edge (i.e., points that fall outside the pulse-limited footprint).

### 3.1.3 Leading Edge Point-Based (LEPTA) correction method

The LEPTA method is similar to the point-based method as it also uses the topographic information from the a priori DEM to find the impact point ($P_l$), but differs in the search method of the impact point. Instead of pre-defining a fixed pulse-limited footprint size, the LEPTA method identifies the parts of the terrain within the beam-limited satellite footprint that contribute to

the rise of the leading edge. To identify these points, we use a beam-limited satellite footprint of $14.39 \times 14.39$ km (Hai et al., 2021) centred around the nadir point and a search range bounded by $r_{\text{begin}}$ and $r_{\text{end}}$:

$$r_{\text{begin}} = \max(r_{1\%}, r_{20\%} - \Delta r), \tag{13}$$

$$r_{\text{end}} = \min(r_{90\%}, r_{20\%} + \Delta r), \tag{14}$$

where $r_{1\%}$ and $r_{90\%}$ refer to the retracked ranges obtained using a 1% and 90% threshold retracker (Davis, 1997), respectively, $r_{20\%}$ is the OCOG retracked range using a 20% threshold to obtain the firn-air interface, and $\Delta r$ is a user-defined threshold. $\Delta r$ is used to avoid the search range ($r_{\text{end}} - r_{\text{begin}}$) becoming unrealistically large. For all experiments, we use a value of $1.25$ m based on an empirical optimisation of $\Delta_r$. If no DEM grid points are identified within the search range, we add the difference between the range to the closest DEM point and $r_{\text{begin}}$ to $r_{\text{begin}}$ and $r_{\text{end}}$.

The location of $P_l$ is computed as the average of all $K$ identified DEM grid points. Finally, the corrected height $h_C$ is computed by

$$h_C = h_N + \frac{1}{K} \sum_{i=1}^{K} (r_{\text{DEM}}^i - (h_S - h_{\text{DEM}}^i)), \tag{15}$$

where $h_{\text{DEM}}^i$ is the ellipsoidal height of the $i$th identified DEM grid point and $r_{\text{DEM}}^i$ the range between the satellite and the $i$th identified DEM grid point. By using averaging to compute $P_l$, it is theoretically possible that the average location is outside

the actual pulse-limited footprint (e.g. when the impact points form a donut shape or two equally large but disjoint sets of points). These occurrences can be easily identified.

One of the advantages of the LEPTA method compared to the point-based method is that it includes points that contribute to the rise of the leading edge signal but are outside the fixed (square) pulse-limited footprint, and rejects points that do not contribute to the rise of the leading edge signal but are inside the pre-defined pulse-limited footprint. An additional advantage

of LEPTA is that it does not apply the recursive computation process as the point-based method, therefore it speeds up the processing.

## 3.2   Performance assessment

To assess the performance of the LEPTA method, we benchmark the different methods by comparing their accuracy relative to reference data. First, we directly compare the corrected heights ($h_C$) for each method with the reference height from the 100 m

ArcticDEM. To compare $h_C$ with the DEM, we bi-linearly interpolate the DEM heights to the CryoSat-2 locations ($h_{\text{DEMC}}$). Then, the CryoSat-2 measurements are grouped in $25 \times 25$ km tiles. For each tile, we compute the median and median absolute deviation of the $h_C - h_{\text{DEMC}}$ values. This assessment cannot be considered as a validation as ArcticDEM is not an independent dataset. However, it is insightful especially when the CryoSat-2 points do not have an ICESat-2 point nearby.

Secondly, we compare the corrected height measurements with the ICESat-2 heights for each method. This comparison is

done per month, i.e., we compare the CryoSat-2 heights acquired in a particular month to the ICESat-2 heights acquired in

the same month. For each point, we first identify all ICESat-2 points within $50$ m of the CryoSat-2 point. If ICESat-2 points are available in each quadrant surrounding the CryoSat-2 point, the ICESat-2 heights are interpolated to the CryoSat-2 point using a natural-neighbour interpolation ($h_{\text{ICE2}}$). Otherwise a nearest neighbour interpolation is applied. A natural neighbour interpolation provides a smoother solution (Bobach, 2009) yet requires weighting functions based on the surrounding points.

To correct for the height difference between the locations of the CryoSat-2 and ICESat-2 points over a potentially sloping terrain, we apply a correction computed as the height difference between the $100$ m ArcticDEM evaluated at the CryoSat-2 ($h_{\text{DEMC}}$) and ICESat-2 ($h_{\text{DEMI}}$) locations. Hence, the differences between the CryoSat-2 and ICESat-2 heights ($\Delta h$) become

$$\Delta h = h_C - h_{\text{ICE2}} - (h_{\text{DEMC}} - h_{\text{DEMI}}) \tag{16}$$

Similar to the comparison with ArcticDEM, we compute the median and median absolute deviation of $\Delta h$ for each $25 \times 25$ km tile.

When benchmarking the methods, two aspects of accuracy are assessed. First, we determine the difference between the slope-corrected CryoSat-2 measurements and the reference heights ($h_{\text{DEMC}}$ or $h_{\text{ICE2}}$) using standard statistical parameters (median, median absolute deviation, mean, and standard deviation). Second, we assess the variability of the statistics for the different methods. The statistical parameters are computed with and without outliers. Cumulative functions are provided mainly to visualise the percentiles that indicate the distribution of the results and determine the outliers. Here, we consider $h_C - h_{\text{DEM}}$ or $\Delta h$ outside the 10–90th percentile range as an outlier. Probability distribution functions are provided to visualise the overall distribution of results. The skewness parameter is provided as long tails of the probability distribution are not completely visualised. In addition, tiles including less than 10 measurements are rejected for visualisation and interpretation, as the statistics of these tiles do not represent sufficient data and cannot be informative.

## 3.3 Sensitivity analysis

The LEPTA method is potentially sensitive to i) the definition of $r_{\text{end}}$ and $r_{\text{begin}}$ and hence $\Delta r$ (Eqs. (13)–(14)), ii) a potential bias in the DEM, and iii) the resolution of the DEM. Another aspect that may impact the height estimates of all methods is the adopted OCOG threshold. To assess how our choices impact the results, we conduct a number of sensitivity analyses in which we:

- Vary $\Delta r$ (Eqs. (13) and (14)) from $0.5$ to $5$ m in steps of $0.5$ m to define an optimal choice.

- Vary the adopted OCOG threshold to determine $R$ and hence $h_N$ (Eq. (16)) from 10% to 90% in steps of 20%, using an optimal choice of $\Delta r$ for LEPTA.

- Add a bias to the DEM from -7.5 to 2.5 metres in steps of 2.5 m, using a 20% OCOG threshold and an optimal choice of $\Delta r$ for LEPTA.

- Vary the DEM resolution from 200 to 900 m in steps of 100 m, and from 1 to 8 km in steps of 1 km, using a 20% OCOG threshold and an optimal choice of $\Delta r$ for LEPTA.

## 4 Results

### 4.1 Comparison with ArcticDEM

The cumulative distribution of $h_C - h_{\mathrm{DEMC}}$ for all methods (Fig. 2a) shows that most values are within the [-1.0, 3.0] m interval
(as shown by 10th and 90th percentiles), although outliers have an impact on the interpretation of the results. These outliers
have most impact on the overall standard deviation and skewness of $h_C - h_{\mathrm{DEMC}}$, as shown in Table 1 and Fig. 3. Although the
distribution curves show a positive bias, the skewness is negative for all methods, showing more or larger negative outliers, as
also shown in Fig. B1. Comparison of the methods, however, shows that LEPTA is least affected by such negative outliers.

**Table 1.** Statistics of the height difference between slope-corrected CryoSat-2 measurements and ArcticDEM and ICESat-2 ($h_C - h_{\mathrm{DEMC}}$ or
$\Delta h$ as computed by Eq. (16)). Height statistics are in unit of metres. The parameters are shown with and without outliers (referred to as w/
outlier and w/o outlier) using 10th and 90th percentiles. E, S, P and L represent ESA L2I, slope method, point-based method and LEPTA,
respectively.

| CryoSat-2 | | vs. ArcticDEM | | | | vs. ICESat-2 | | | |
|---|---|---|---|---|---|---|---|---|---|
| | | E | S | P | L | E | S | P | L |
| No. of data | w/ outlier | 2.2e6 | 2.2e6 | 2.2e6 | 2.2e6 | 8.2e4 | 8.3e4 | 8.3e4 | 8.2e4 |
| | w/o outlier | 1.8e6 | 1.8e6 | 1.8e6 | 1.8e6 | 6.6e4 | 6.6e4 | 6.6e4 | 6.6e4 |
| median | w/ outlier | 0.43 | 0.69 | 0.71 | 0.24 | 0.01 | 0.21 | 0.48 | 0.00 |
| | w/o outlier | 0.43 | 0.69 | 0.71 | 0.24 | 0.01 | 0.21 | 0.48 | 0.00 |
| median absolute deviation | w/ outlier | 0.58 | 0.66 | 0.45 | 0.45 | 0.18 | 0.24 | 0.14 | 0.12 |
| | w/o outlier | 0.44 | 0.50 | 0.35 | 0.34 | 0.14 | 0.19 | 0.10 | 0.09 |
| mean | w/ outlier | 0.58 | 1.27 | 0.47 | 0.22 | 0.39 | 1.14 | 0.39 | 0.06 |
| | w/o outlier | 0.51 | 0.87 | 0.70 | 0.22 | 0.13 | 0.51 | 0.50 | 0.00 |
| standard deviation | w/ outlier | 2.87 | 3.11 | 2.75 | 1.72 | 2.73 | 3.27 | 1.78 | 1.64 |
| | w/o outlier | 0.64 | 0.82 | 0.46 | 0.46 | 0.35 | 0.70 | 0.15 | 0.13 |
| skewness | w/ outlier | -14.29 | -2.87 | -22.03 | -1.93 | -2.58 | 2.86 | -21.13 | -15.60 |
| | w/o outlier | 0.49 | 0.90 | -0.11 | -0.17 | 1.77 | 1.93 | 0.50 | -0.15 |

Removing the outliers significantly reduces the standard deviation of $h_C - h_{\mathrm{DEMC}}$ and skewness for all methods and brings the
mean closer to the median. Comparison of the mean and median values (Table 1) and probability distribution (Fig. 3a) moreover
indicates that LEPTA performs better than other methods when compared with ArcticDEM, with a mean height difference of
0.22 m and a median difference of 0.24 m. The slope method results in the largest mean difference of 0.87 m, while the point-
based method gives the largest median of 0.71 m. The standard deviation (0.46 m) and median absolute deviation (0.34 m) from
LEPTA are also the smallest, same as those obtained from the point-based method. The largest $h_C - h_{\mathrm{DEMC}}$ deviation values
after outlier removal are given by the slope method, with the standard deviation being 0.82 m and median absolute deviation

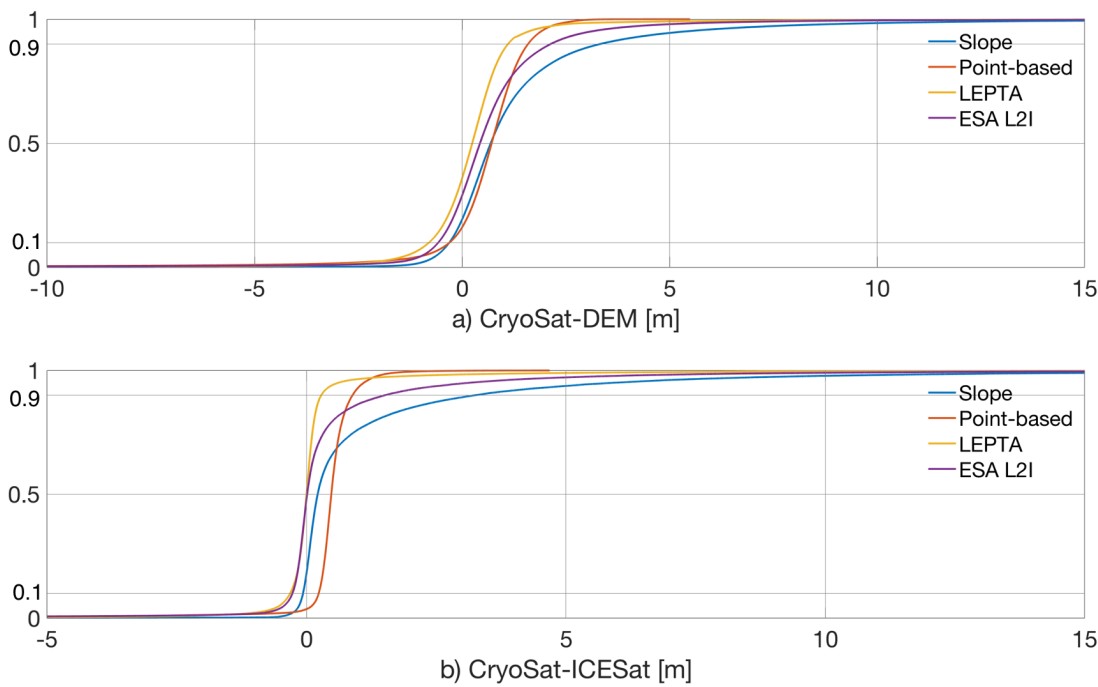

**Figure 2.** Cumulative distribution figures of a) the difference between CryoSat-2 and ArcticDEM ($h_C - h_{\text{DEM}}$) and b) the difference between CryoSat-2 and ICESat-2 ($\Delta h$), including outliers. 10th and 90th percentiles are shown in the figures for outlier removal. For visualisation, x-axis is restricted to [-10, 15] m for figure a and to [-5, 15] m for figure b.

being 0.50 m. An additional note is that the mean and median from all methods are positive, which implies that the heights obtained by these methods are generally higher than ArcticDEM heights.

Comparison of the spatial patterns of median and median absolute deviation (Fig. 3) shows large spatial differences in both pattern and magnitude among the different methods. In general, the largest median and median absolute deviation values occur near the margins of the LRM coverage, where the terrain is steeper. For the point-based method and LEPTA, the median values on the western side are generally lower than on the eastern side. This spatial pattern is similar to that of the differences between ICESat-2 and ArcticDEM (shown in Fig. A1). For ESA L2I products and the slope method, the largest median values occur on both eastern and western sides of Greenland, and those from the slope method largely exceed those of the ESA L2I products, the point-based method, and LEPTA. So far, we lack a conclusive explanation for the spatial differences between the methods. Regarding the median absolute deviation values, we observe in general higher values on the western side of the ice sheet than in the interior. For the ESA L2I products and the slope method, the median absolute deviation values are also high on the eastern side. These median absolute deviation values show that topography affects the different performances of the methods, and the point-based method and LEPTA are less affected on the eastern side. In addition, for the point-based method, removing the outliers results in most missing data close to Jakobshavn. Combining the statistics in Table 1 and the spatial

distribution of median and median absolute deviation in Fig. 3, it can be concluded that LEPTA performs best when compared with ArcticDEM.

Using averaging in Eq. (15) to compute $P_l$ results in 5.2% of the impact points being outside the actual footprint. Removing these points as 'unreliable data' minimally affects the median and mean (0.26 m and 0.25 m), but improves the median absolute deviation (0.32 m) and standard deviation (0.40 m).

**4.2    Validation with ICESat-2 observations**

Comparison of CryoSat-2 and ICESat-2 heights (Fig. 2b) shows again the impact of outliers on the results, although the outliers are generally lower than for the ArcticDEM comparison. ESA L2I products, the point-based method and LEPTA have more impacts from negative outliers, while the slope method results in more positive outliers.

With the outliers removed, the standard deviation of $\Delta h$ values from all methods is greatly reduced, especially for the ESA

L2I and slope-method which show the largest outliers (Fig. 2b). The lowest median (0.00 m), mean (0.00 m), median absolute deviation (0.09 m) and standard deviation (0.13 m) of $\Delta h$ are obtained by LEPTA, showing that the LEPTA method again outperforms the other methods. The largest median (0.48 m) is obtained by the point-based method, and the largest mean (0.51 m), median absolute deviation (0.19 m) and standard deviation (0.70 m) are from the slope method.

The comparison of the height differences between CryoSat-2 vs. ArcticDEM and CryoSat-2 vs. ICESat-2, respectively,

shows moreover that the height differences with ICESat-2 are smaller, probably due to the better quality of ICESat-2 data compared to ArcticDEM and the longer time gap between CryoSat-2 and ArcticDEM, as satellite imagery data for generating ArcticDEM were gathered since 2007 (Noh and Howat, 2017; Howat et al., 2019) and co-registered to ICESat from before 2009, whereas ICESat-2 measurements were obtained in the same month as CryoSat-2 data. The comparison between CryoSat-2 and ICESat-2 also results in less data points, as not all CryoSat-2 measurements have corresponding nearby ICESat-2

measurements within the 50 m criterion.

The spatial distribution of the median and median absolute deviation of $\Delta h$ (Fig. 4) shows clear spatial patterns. For the ESA L2I products, the slope method, and the point-based method, the median differences with respect to the overall median difference are generally negative (positive) in the central part (margins of the LRM zone). For LEPTA, the variability of negative or positive differences is smaller (especially vs. ESA L2I and the slope method), but with a slightly reversed pattern.

This reversed pattern can be explained by LEPTA's definition of $r_{\text{begin}}$ and $r_{\text{end}}$ that may result in an asymmetry around $r_{20\%}$ that can spatially vary. Figure 4 also shows that LEPTA has the lowest spatial variability of the median absolute deviation, whereas the slope method shows the largest contrast between the interior and the margins of the LRM zone.

**4.3    Sensitivity to the definition of the search range**

The performance of the LEPTA method relies on the definition of $r_{\text{begin}}$ and $r_{\text{end}}$ and hence $\Delta r$. To assess the sensitivity of LEPTA

to the choice of $\Delta r$, we repeat the performance assessment by varying $\Delta r$, as introduced in Section 3.3. The results of this $\Delta r$ sensitivity assessment are summarised in Fig. 5. It shows that while $\Delta r$ changes at metre level, the median and the median absolute deviation values of $\Delta h$ only change at centimetre level. More specifically, the median and median absolute deviation

increase with increasing $\Delta r$. From Fig. 5, we can also conclude that $\Delta r = 1.25$ results in a near-zero median difference compared to ICESat-2. Hence $\Delta r = 1.25$ is used for all experiments.

However, it is not sufficient to conclude that LEPTA is robust to the choice of $\Delta r$ by merely assessing $\Delta h$. The reason is that different $\Delta r$s might result in different horizontal locations, which are then compared to potentially different ICESat-2 measurements. Therefore, Fig. 6 shows the differences in the ellipsoidal height and horizontal position of the impact points obtained using $\Delta r = 2$ m ($\Delta r_2$) and $\Delta r = 1$ m ($\Delta r_1$). This comparison shows whether a $\Delta r$ change of 1 m can result in large horizontal and vertical offsets. In the interior of the ice sheet this effect is small as the vertical and horizontal offsets resulting from $\Delta r_2$ vs. $\Delta r_1$ are close to 0. In the margin regions of LRM coverage, however, increasing $\Delta_r$ results in lower elevation of impact points and horizontal offsets with mean values up to 20 m and standard deviations up to 250 m.

## 4.4 Assessment of OCOG retracker threshold dependence

Changing the OCOG retracker threshold from 10% to 90% results in retracked points further away from the satellite and hence lower height estimates (Fig. 7). For all methods, this behaviour is apparent, as the median of $\Delta h$ is reduced by approximately 1.2 m when the threshold increases from 10% to 90%. Changing the OCOG retracker threshold in LEPTA results only in a change of the height of $P_l$ and does not affect the selection of the DEM points that contribute to $P_l$. This means that increasing the OCOG retracker threshold actually corresponds to increasing the depth of the radar return within the snowpack or firn. Moreover, Fig. 7 highlights that the adopted OCOG retracker threshold of 20% for LEPTA results in a near-zero median difference compared to ICESat-2, indicating that on average it effectively detects the absolute ice sheet height.

## 4.5 Sensitivity to potential biases in the DEM

To assess the sensitivity of the methods to potential constant ice sheet elevation changes, we perform a sensitivity analysis in which we add biases to the DEM. Figure 8 shows that the slope and the point-based methods are not affected by these DEM biases, while it does affect LEPTA. The impact, though, depends on the sign of the bias. Adding a bias between $-7.5$ and $-2.5$ m (which corresponds to ice sheet lowering) only changes the median $\Delta h$ by approximately 2.3 cm, while adding a bias of 2.5 m (which corresponds to an increase in ice sheet elevation) results in a median $\Delta h$ that is 8.8 cm higher. A similar observation holds for the median absolute deviation of $\Delta h$. This dependency on the sign of the bias can be easily understood. The impact point is typically in the area where the range between the satellite and the terrain is smallest. Lowering the DEM and thereby increasing the range to the satellite hence results in a reduced number of DEM grid points within the search range ($r_{\mathrm{end}} - r_{\mathrm{begin}}$). If no points are found, the search range is adjusted. Applying a positive bias, on the other hand, will result in other parts of the terrain being within the search range.

Despite LEPTA's sensitivity to a potential bias in the DEM, however, the median and median absolute deviation of $\Delta h$ remain lower than the other methods for negative biases up to $-7.5$ m. With a positive bias of 2.5 m, the median absolute deviation of $\Delta h$ from LEPTA is approximately 8 mm higher than that from the point-based method. In Appendix C, we present the results of a similar analysis as shown in Fig. 6. It shows that the impact of a potential bias in the DEM is largest

at the western side of the LRM zone, resulting in vertical and horizontal offsets with mean values up to 2 m and 50 m and standard deviations up to 3.5 m and 700 m, respectively.

## 4.6 Sensitivity to the resolution of the DEM

Figure 9 shows the effect of changing the DEM resolution on the median and median absolute deviation of $\Delta h$ for different slope correction methods. For both the slope and the point-based method, the smallest median $\Delta h$ is obtained at 2 km resolution. For the slope method, the median $\Delta h$ increases from 0.21 to 0.30 m when the DEM resolution increases from 2 to 8 km. For the point-based method, the variation of median $\Delta h$ for DEM resolutions between 100 m and 2 km is within millimetre level. Lowering the resolution down to 8 km increases the median to 0.62 m. For LEPTA, the variation of the median $\Delta h$ for DEM resolutions between 100 m and 1 km is within millimetre level. For lower resolutions, the median $\Delta h$ increases to 0.23 m (8 km resolution). The smallest median absolute deviation for the slope method (0.19 m) and the point-based method (0.09 m) are obtained at 2 km resolution. For LEPTA, the smallest median absolute deviation is obtained when using a 1 km resolution, though the values between resolutions of 100 m and 2 km vary at millimetre-level. For resolutions lower than 2 km, the median absolute deviation for both the point-based method and LEPTA increases by approximately 10 cm. For the slope method, the increase is 6 cm.

## 5 Discussion

The comparison with ArcticDEM and validation based on ICESat-2 show that the presented LEPTA method outperforms the slope and point-based methods as well as the ESA L2I product in accuracy with lower median, mean, and median absolute deviations. Especially in the margin regions of the LRM zone, heights derived from LEPTA correspond more closely to ICESat-2 height measurements, compared to the slope method being used by ESA. This indicates that including leading edge information to determine the impact point results in an important improvement of the accuracy of CryoSat-2 LRM height estimations. By showing the importance of accurately determining the impact points over steeper margin areas, our results confirm earlier work of Levinsen et al. (2016) in the margin regions, where they also showed that the point-based method outperforms the slope method in median absolute deviation values. The improved performance of the point-based method and LEPTA method can be explained by the assumption of a constant slope within the footprint in the slope-based method, which results in a biased impact point further away from the satellite than the optimal location (Levinsen et al., 2016). An explanation for the improved performance of LEPTA over the point-based method can be found in the design of the method which only takes into account areas that contribute to the rise of CryoSat-2 LRM waveform leading edge (Fig. 1).

Our results also show that the ESA L2I product outperforms our self-implemented slope correction method. This agrees with Levinsen et al. (2016) who attributed the different performance between the ESA's Envisat Radar Altimetry-2 products and their self-implementation of the slope correction method to the Doppler slope correction step implemented in ESA L2I products (Blarel and Legresy, 2012) and differences in the used DEM. We must admit that at this stage an explanation for the

difference we obtained is lacking. Detailed analysis (not shown in this paper) shows that the differences cannot be explained by the fact that in our study we use another DEM.

The first sensitivity analysis shows that in terms of bulk statistics, LEPTA is quite robust for the definition of the search range. Compared to ICESat-2, the change in the median is $< 0.1$ m for the interval over which we changed $\Delta r$, while the change in the median absolute deviation is at millimetre level. Regionally, the impact may be larger. In particular, we observe changes up to $1.46$ m in the vertical and $231$ m in the horizontal position of the impact points towards the margins of the LRM zone. In these areas, the mean and standard deviation of the leading edge width are larger. This, in turn, suggests using a larger $\Delta r$ locally. The use of a spatially varying $\Delta r$ is hence considered as a potential further improvement of the method.

Increasing the OCOG retracker threshold lowers the height estimates for all methods. For both LEPTA and the point-based method, the horizontal position of the impact points does not change. This means that increasing the OCOG retracker threshold actually corresponds to increasing the depth of the radar return within the snowpack or firn. That is, the adopted threshold controls the observed penetration. Our results confirm that using a 20% threshold gives on average comparable height estimates as ICESat-2. It is meanwhile worth noting that the probable scattering of ICESat-2 photons within the snowpack cannot be neglected (Smith et al., 2021).

Differently from the slope and point-based methods, LEPTA shows sensitivity to a bias in the DEM. The presence of a bias in the DEM does not affect the slope or the relative differences between the DEM points, which are key to the slope method and the point-based method respectively. However, in the case of LEPTA, when the DEM heights are biased and the search range determined by the waveform leading edge is unchanged, the DEM points used to calculate the impact point of LEPTA are changed. According to Appendix C, this bias mainly affects the margins of the LRM coverage. Overall these bias effects indicate that it is key to have up-to-date, time-varying DEMs when applying LEPTA to correct for slope-induced errors. Changes in the elevation over time will affect the applied correction as well as the location of the impact point. However, in case of non-homogeneous elevation changes (which will result in slope changes) this also holds for the other methods.

Sensitivity to DEM resolution shows that the slope and point-based method perform best with an intermediate DEM resolution (2 km), which is consistent with Levinsen et al. (2016). However, differently from Levinsen et al. (2016) who obtained stable performance for the point-based method between 2 and $4$ km DEM resolution, our results show that the performance of the point-based method is stable when the DEM resolution is finer than 2 km. This can be attributed to differences in i) the study area, ii) the used altimeter data, iii) the used DEM to compute the corrections, and iv) the reference data and methods for validation. In principle, the point-based method should perform better with a finer DEM resolution because it has the advantage of using full topography rather than assuming a constant slope, as used by the slope method. While Levinsen et al. (2016) attributed the optimal 2 km resolution of other methods to the radar altimetry's ability to resolve small-scale surface features, our results show that $\Delta r$ used by LEPTA to define the pulse-limited footprint may have a different impact (e.g. asymmetry around $r_{20\%}$). Therefore, for future studies, fine-tuning the impact of $\Delta r$ is still of high importance.

Moreover, our experiment focuses on the performance of LEPTA in the CryoSat-2 LRM-covered regions over the Greenland ice sheet, therefore it remains to be studied how it performs over more complex terrains and Antarctica. Since the topography

and DEM quality in other regions of the Earth are different from those in Greenland, we expect LEPTA to perform differently, and the impact of $\Delta r$ can also vary. This phenomenon provides more aspects for future works.

Finally, while we use CryoSat-2 Baseline D data, Baseline E is available. However, we do not expect changes that significantly affect the conclusions of this study, as main changes in Baseline E are associated with the sea ice products (European Space Agency, 2021).

## 6 Conclusions

Reducing slope-induced errors is a key correction algorithm when processing LRM data over ice sheets. To correct for this error, different methods have been developed to determine the impact point, which all rely on footprint assumptions: e.g. slope-method, which assumes a constant slope within the footprint, or the point-based method, which assumes a fixed footprint size to determine the impact point by minimising the mean distance. Each of these methods has shortcomings as they either neglect
the actual topography or the actual footprint that can be estimated by a combination of the leading edge and topography. To overcome these shortcomings, we present a Leading Edge Point-Based (LEPTA) method that corrects for the slope-induced error by including the leading edge information of the radar waveform to determine the impact point. The principle of the method is that only the points on the ground that are within range determined by a specific search range that contributes to the rise of the waveform leading edge are used to determine the impact point.

Different methods for correcting the slope-induced errors are used in this study using CryoSat-2 measurements over the Greenland ice sheet. Statistics show that the LEPTA method outperforms all other methods with the smallest median and variability of errors. The median difference between ICESat-2 heights and CryoSat-2 heights derived by LEPTA using a 20% OCOG threshold and $\Delta r = 1.25$ m search range is 0.00 m. Spatially, LEPTA has a good improvement compared to the traditional slope method on the margins of the LRM-covered regions of the ice sheet, as it derives heights generally more than
2 m closer to ICESat-2 measurements. LEPTA is sensitive to the definition of the search range, and the bias in the DEM used to correct for the slope-induced error, mainly in the horizontal location of the impact points. However, comparison with ICESat-2 measurements generally shows centimetre-level sensitivity. Therefore, LEPTA is a method worth considering to obtain accurate height measurements with radar altimetry, especially in regions with complex topography.

### Appendix A: Comparison between ICESat-2 measurements and ArcticDEM

ICESat-2 ALT06 Land Ice Height data include a large amount of measurements between Jan. 1 and Dec. 31 2019. Therefore, we compute the statistics of the differences between ICESat-2 heights ($h_{\mathrm{ICE2}}$) and ArcticDEM interpolated to the corresponding locations ($h_{\mathrm{DEMI}}$) per beam pair. In this process, data outside the CryoSat-2 LRM zone have been excluded. The statistics are summarised in Table A1. All differences are computed as $h_{\mathrm{ICE2}} - h_{\mathrm{DEMI}}$. The median difference between ICESat-2 and ArcticDEM for all beam pairs is 0.21 m, showing good agreement. The mean differences are around 5 cm.

**Table A1.** Difference between ICESat-2 measurements and ArcticDEM values interpolated to ICESat-2 locations.

| Beam pair of measurements | vs. ArcticDEM [m] | | | |
|---|---|---|---|---|
| | mean | standard deviation | median | median absolute deviation |
| pair one | 0.05 | 1.13 | 0.21 | 0.72 |
| pair two | 0.06 | 1.13 | 0.21 | 0.72 |
| pair three | 0.05 | 1.13 | 0.21 | 0.72 |

**Appendix B: Probability distribution functions of the height differences showing skewness**

Probability distribution functions of all methods are provided in Figs. B1-B2 to illustrate the underlying skewness in Table 1 and Fig. 3-Fig. 4 within [-10 m, 10 m] range. However, the skewness can also be affected by large outliers, as also shown in the figures. For the slope method, the DEM resolution is 2 km. For LEPTA, the $\Delta r$ is 1.25 m. For the slope method, point-based method and LEPTA, the retracker is the OCOG retracker with a 20% threshold.

**Appendix C: Impact of a bias in the ArcticDEM on the 3D location of the LEPTA impact points**

Figures C1- C2 show the three-dimensional difference between using the original ArcticDEM and the vertically displaced DEM to correct for the slope-induced error. The vertical and horizontal differences are calculated using the difference between the location of impact points $P_l$ of the biased DEM minus the location of the impact points of the original ArcticDEM. Figure C1 shows that when the DEM used has a negative bias, the corrected heights are higher, the horizontal locations on the western
side of the ice sheet are in general biased towards the northeast, and the horizontal locations on the northeast side of the ice sheet are biased towards the southwest. Figure C2 shows an inverse pattern, when the DEM shows a positive bias. On the interior of the ice sheet, however, the effects of the DEM biases are small.

*Author contributions.* WL conducted data management, processing and analysis, produced the figures, and provided the manuscript with contributions from all co-authors. CS designed the study and provided expertise and software for radar-altimetry processing. SL provided
support on statistical analysis and data visualisation.

*Competing interests.* The authors declare they have no conflict of interest.

*Acknowledgements.* The research is supported by the Dutch Research Council (NWO) on the ALWGO.2017.033 project.

ArcticDEM is provided by the Polar Geospatial Center under NSF-OPP awards 1043681, 1559691, and 1542736. DEM of Greenland used for result visualisation is provided by Helm et al. (2014a, b) under license Creative Commons Attribution 3.0 Unported. The CryoSat-2 L1b and L2I data are provided online by ESA and the ICESat-2 L3A data are provided online by NSIDC (https://nsidc.org/data/atl06).

The authors would also like to thank Professor Roland Klees, Dr Bert Wouters, Katarzyna Sejan, Jan Haacker and Dr Lorenzo Iannini for valuable discussions, and Dr Louise Sandberg Sørensen for the review and editing of this manuscript. Finally, we would like to thank the referees for reviewing and providing recommendations to improve this manuscript.

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

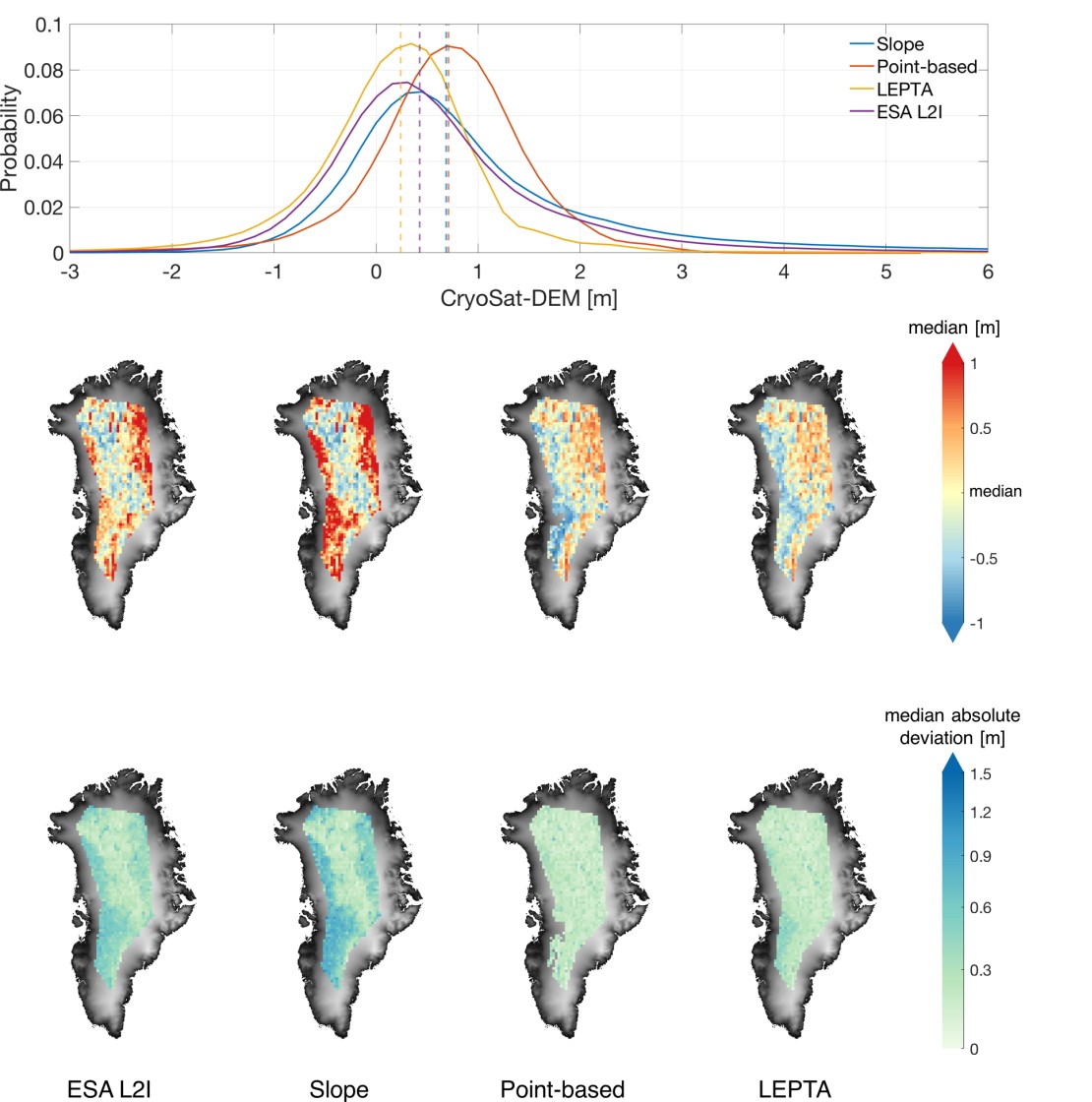

**Figure 3.** Upper panel: probability distribution of height difference between CryoSat-2 and the ArcticDEM, before removing the outliers with 10th and 90th percentiles. The probability distribution is plotted with all data samples, but restricted to [-3m, 6m] for visualisation (for better illustration of the skewness and large outliers, please refer to Appendix B). Vertical lines show median value per method. Middle and lower panels: spatial distribution of median and median absolute deviation of the height difference per tile of $25 \times 25$ km, after removing the outliers. To enhance the visibility of the maps, the median value of each method is subtracted in the middle panel. The colours of the median absolute deviation plots are in logarithmic scale to enhance contrast. The spatial distribution results from left to right are obtained by ESA L2I products, the slope method, the point-based method and LEPTA, with the $1 \times 1$ km DEM covering Greenland (Helm et al., 2014a, b) as background.

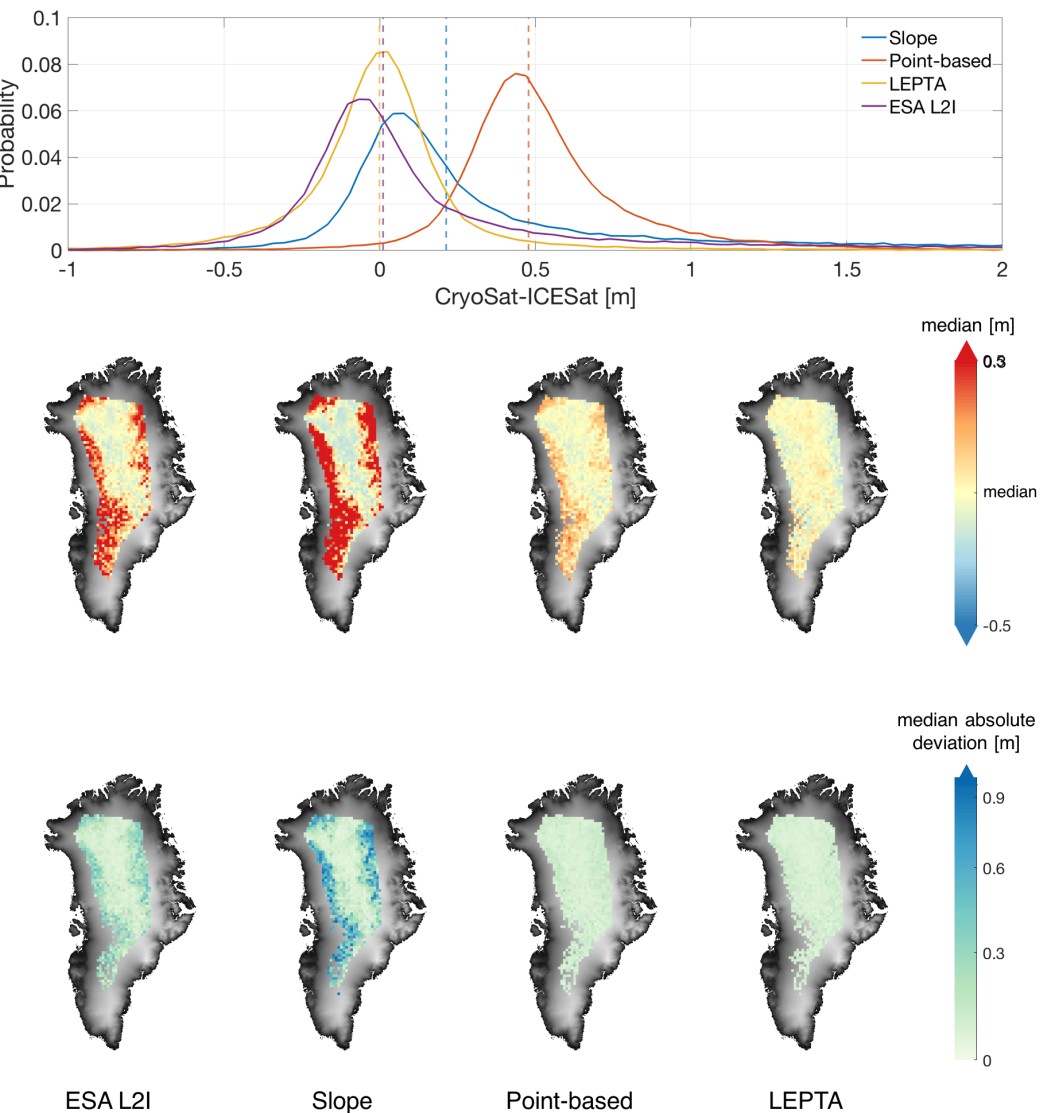

**Figure 4.** Upper panel: probability distribution of height difference between CryoSat-2 and the ICESat-2, before removing the outliers. The probability distribution is plotted with all data samples, but restricted to [-1 m, 2 m] for visualisation (for better illustration of the skewness and large outliers, please refer to Appendix B). Vertical lines show median value per method. Middle and lower panels: spatial distribution of median and median absolute deviation of the height difference per tile of $25 \times 25$ km, after removing the outliers. To enhance the visibility of the maps the median value of each method is subtracted in the middle panel. The colours of the median absolute deviation plots are in logarithmic scale. The spatial distribution results from left to right are obtained by ESA L2I products, the slope method, the point-based method and LEPTA, with the $1 \times 1$ km DEM covering Greenland (Helm et al., 2014a, b) as background.

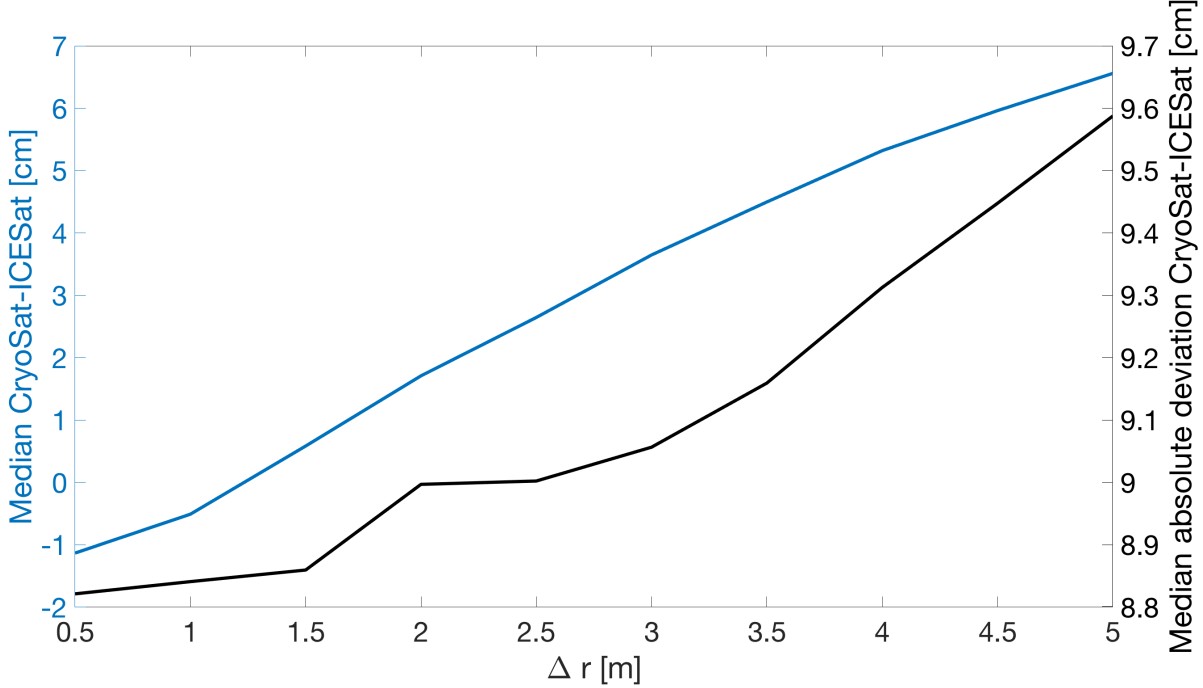

**Figure 5.** Median (left axis) and median absolute deviation (right axis) of the height differences between CryoSat-2 and ICESat-2 ($\Delta h$ calculated with Eq. (16)) as function of $\Delta r$. Outliers are removed using 10th and 90th percentiles.

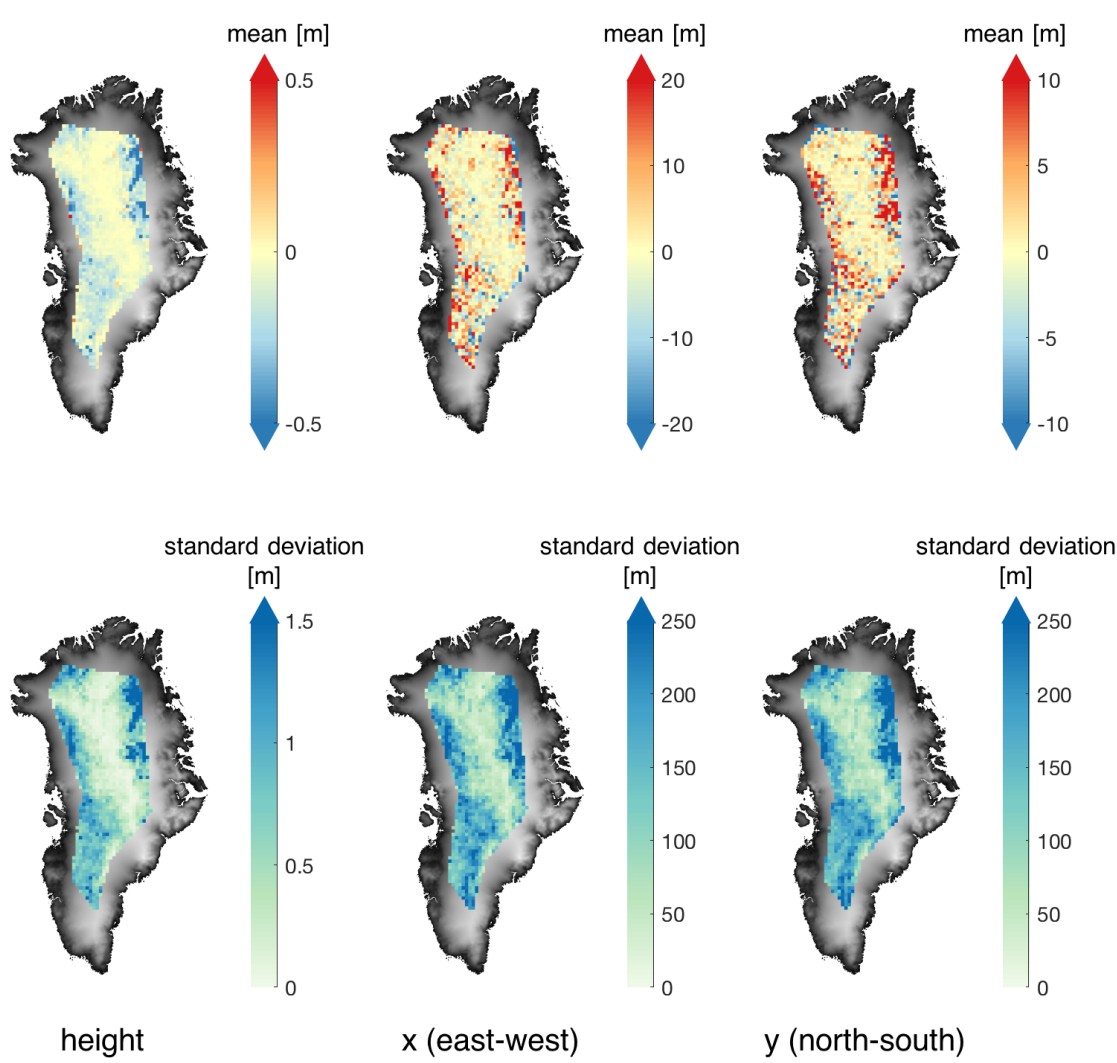

**Figure 6.** Mean and standard deviation of the differences between the height and horizontal location of the impact point obtained using $\Delta r = 2$ m ($\Delta r_2$) and $\Delta r = 1$ m ($\Delta r_1$). The mapped locations are based on the horizontal locations (x and y) derived from $\Delta r_1$, tiled by the $25 \times 25$ km grid same as in Fig. 4.

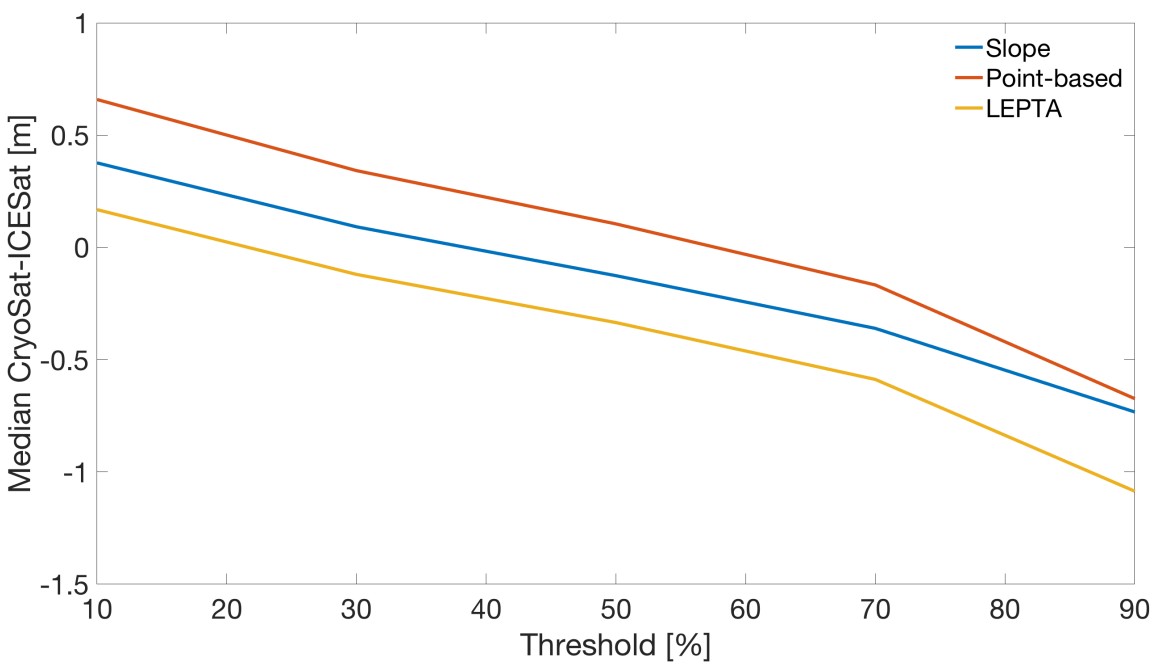

**Figure 7.** Median of height differences between CryoSat-2 and ICESat-2 ($\Delta h$ calculated with Eq. (16)) as function of the OCOG retracker threshold. Outliers are removed using 10th and 90th percentiles.

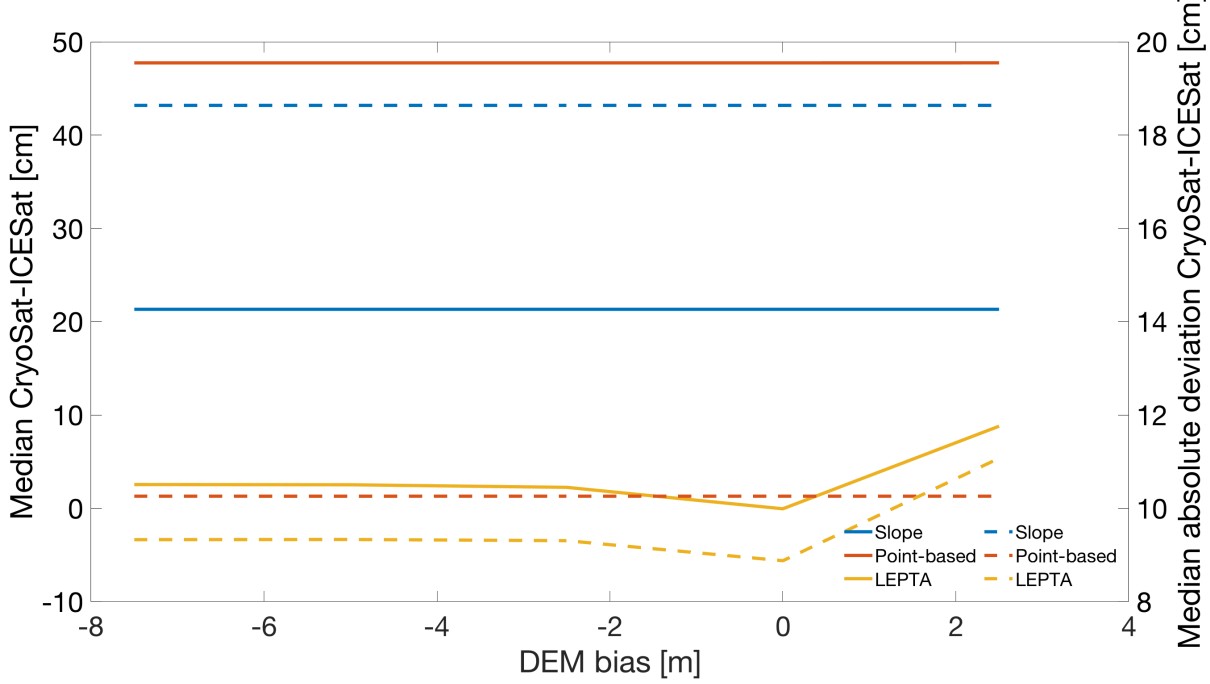

**Figure 8.** Median (left axis, solid curves) and median absolute deviation (right axis, dashed curves) of height differences between CryoSat-2 and ICESat-2 ($\Delta h$ calculated with Eq. (16)) as function of a bias in the DEM. Outliers are removed using 10th and 90th percentiles.

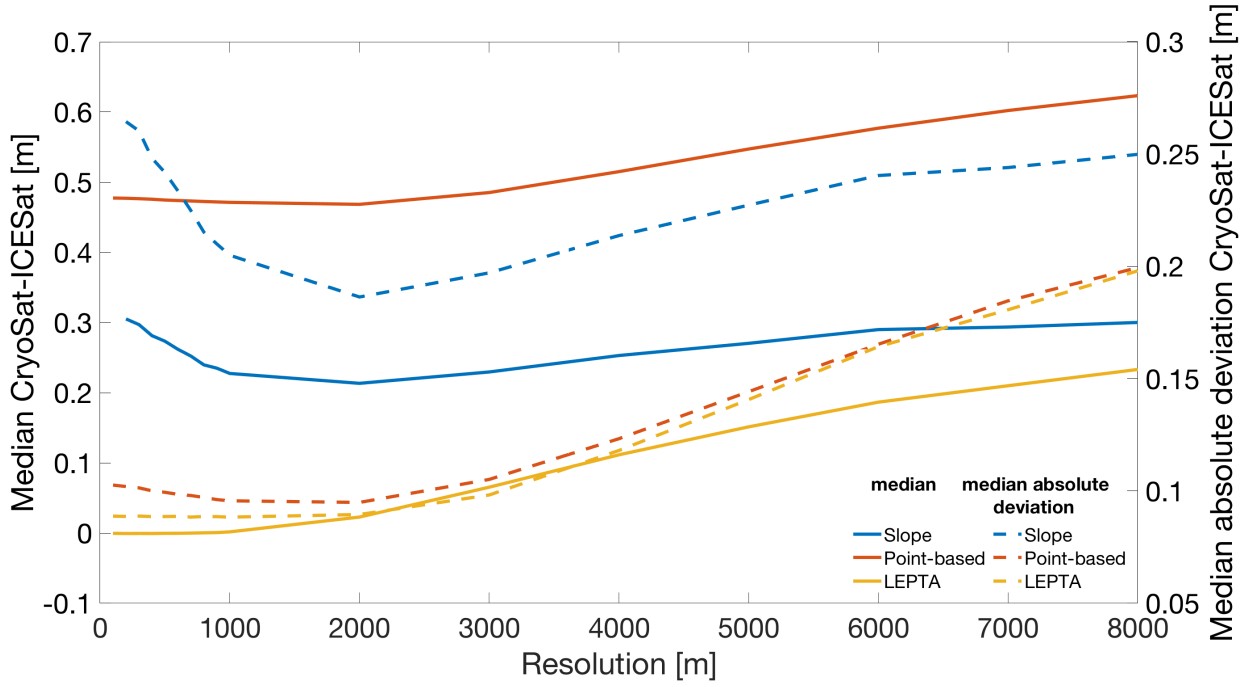

**Figure 9.** Median (left axis, solid curves) and median absolute deviation (right axis, dashed curves) of height differences between CryoSat-2 and ICESat-2 ($\Delta h$ calculated with Eq. (16)) as a function of DEM resolution. Outliers are removed using 10th and 90th percentiles.

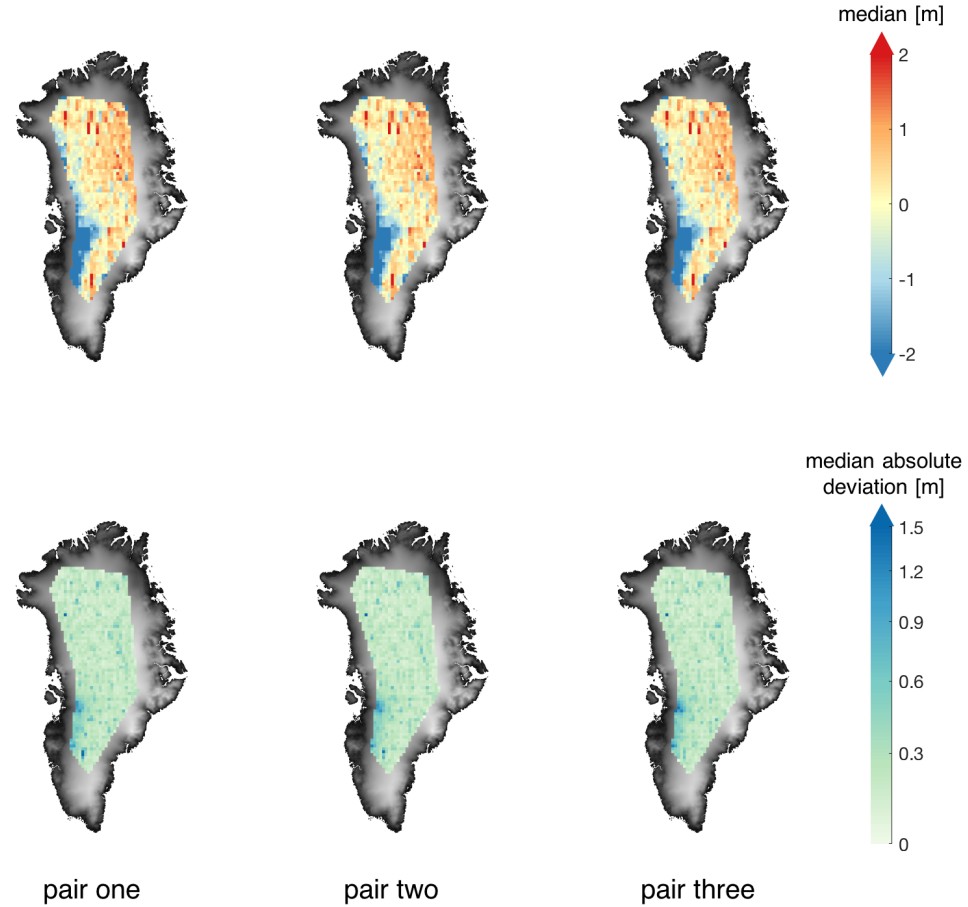

**Figure A1.** Spatial distribution of median and median absolute deviation of the height difference between each pair of ICESat-2 beams and ArcticDEM (tiled in $25 \times 25$ km). The colours of the median absolute deviation plots are in logarithmic scale.

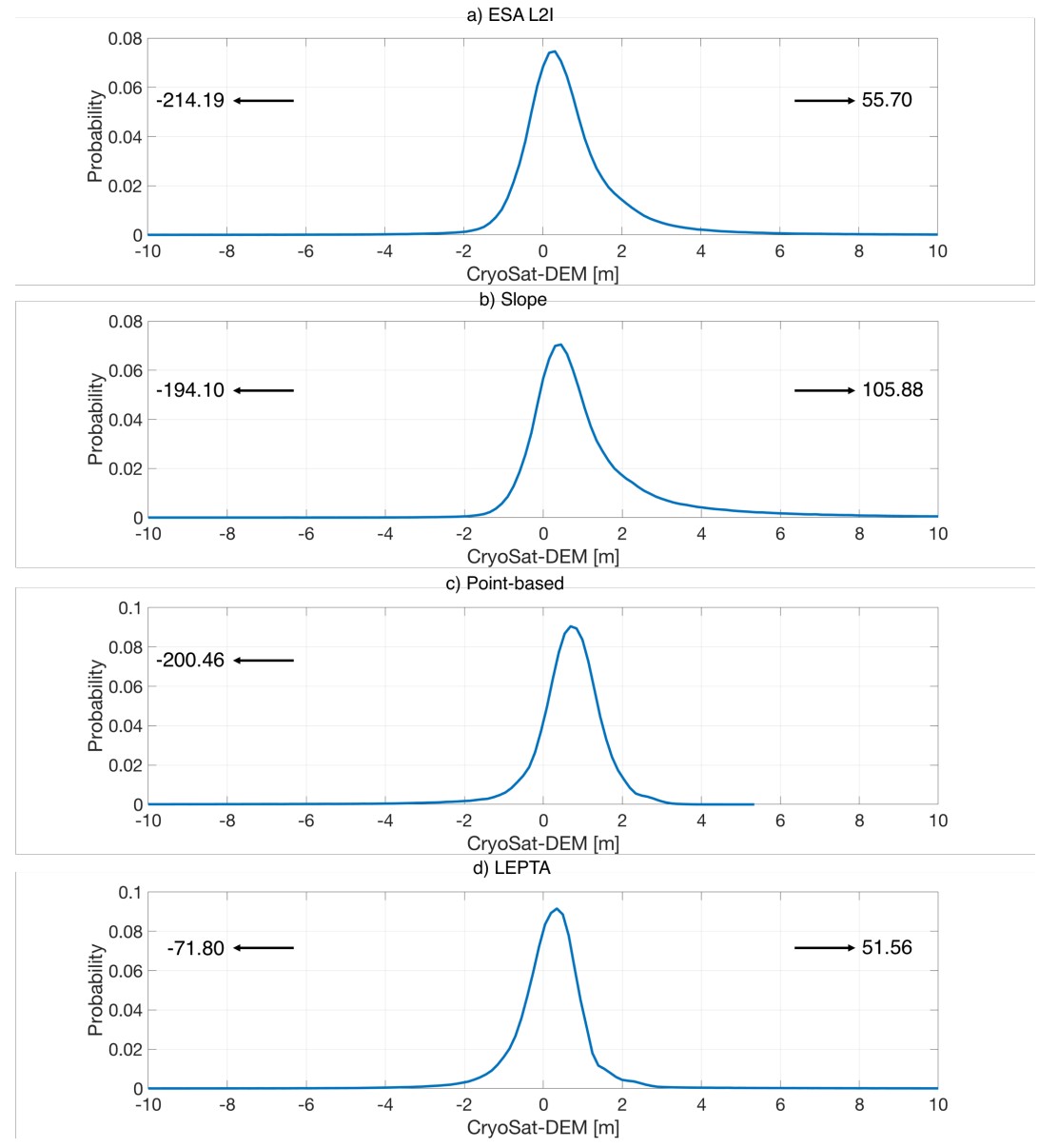

**Figure B1.** Probability distribution functions of heights between CryoSat-2 and ArcticDEM derived from a) ESA L2I, b) slope method, c) point-based method and d) LEPTA centred between [-10 m, 10 m]. To clearly show minimum and maximum values (values displayed with arrows), the curves are not displayed in the same panel.

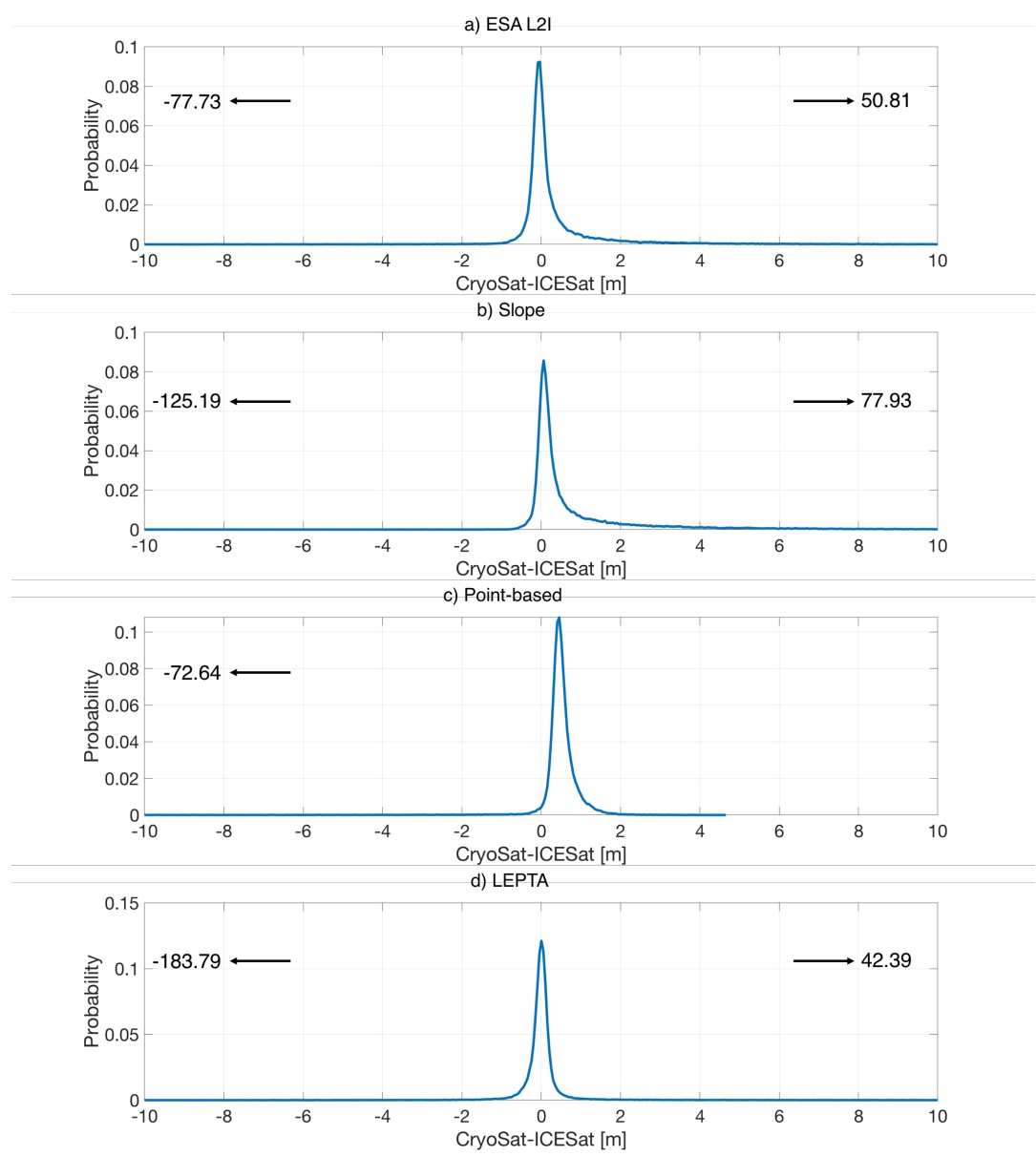

**Figure B2.** Probability distribution functions of heights between CryoSat-2 and ICESat-2 derived from a) ESA L2I, b) slope method, c) point-based method and d) LEPTA centred between [-10 m, 10 m]. To clearly show minimum and maximum values (values displayed with arrows), the curves are not displayed in the same panel.

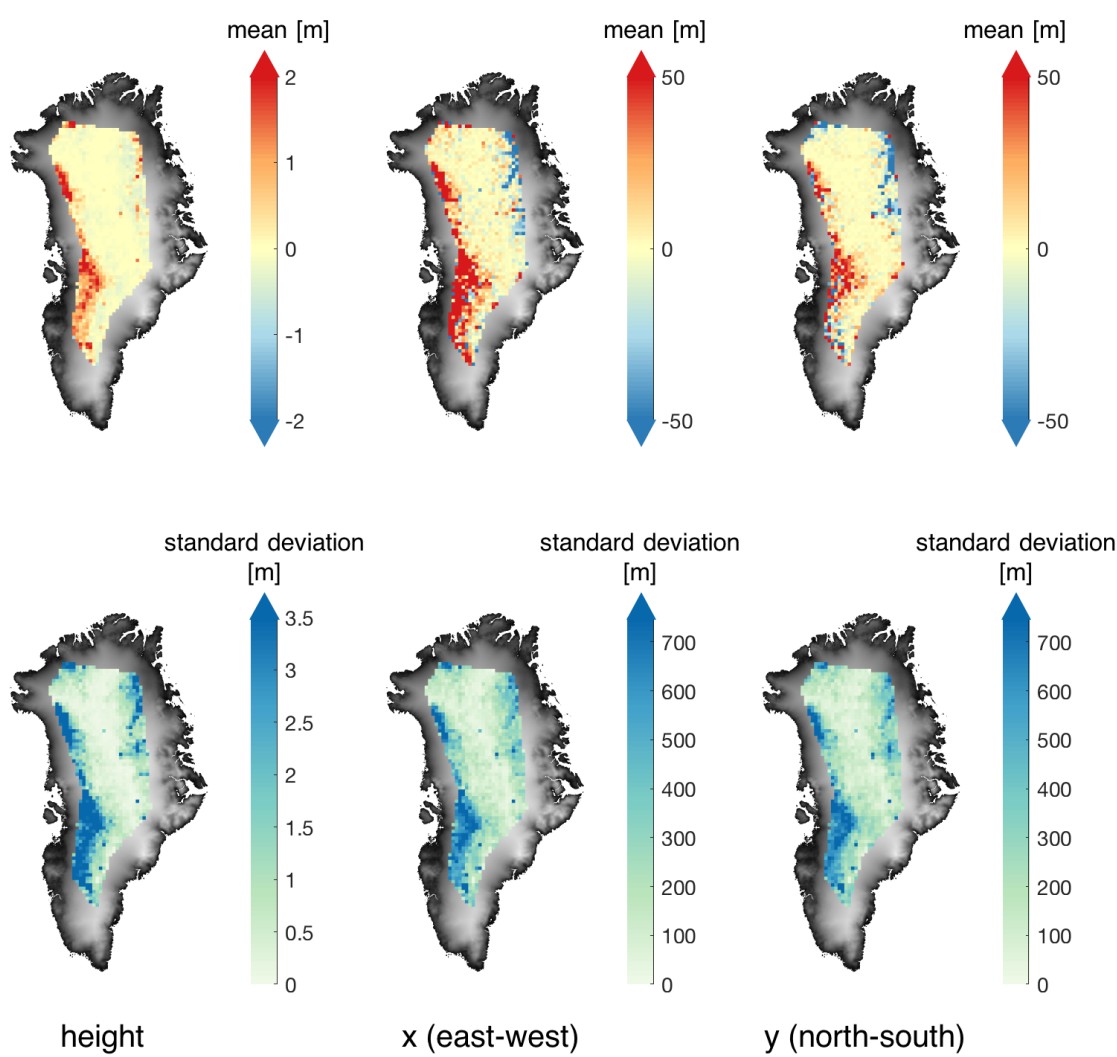

**Figure C1.** Mean and standard deviation of vertical and horizontal difference of derived impact point $P_l$ between i) using the DEM with a homogeneous vertical displacement $\Delta h_{\text{DEM}} = -2.5$ m ($\Delta h_{\text{DEM1}}$) and ii) using the original ArcticDEM (DEM$_{\text{orig}}$). The mapped locations are based on the horizontal locations (x and y) derived from DEM$_{\text{orig}}$, tiled by the $25 \times 25$ km grid same as in Fig. 6.

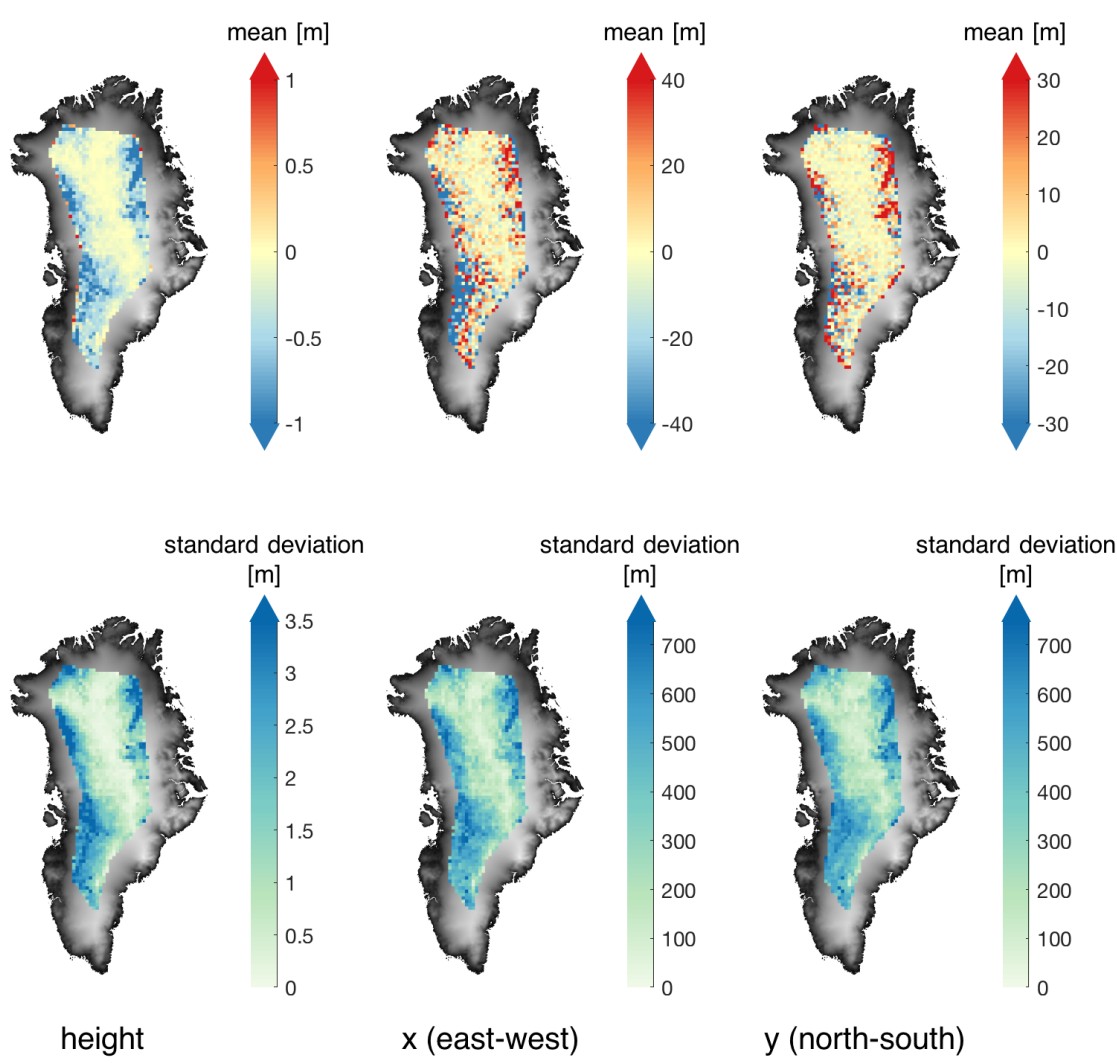

**Figure C2.** Mean and standard deviation of vertical and horizontal difference of derived impact point $P_l$ between i) using the DEM with a homogeneous vertical displacement $\Delta h_{\mathrm{DEM}} = 2.5$ m ($\Delta h_{\mathrm{DEM1}}$) and ii) using the original ArcticDEM (DEM$_{\mathrm{orig}}$). The mapped locations are based on the horizontal locations (x and y) derived from DEM$_{\mathrm{orig}}$, tiled by the $25 \times 25$ km grid same as in Fig. 6.