# Peer review of "A leading-edge based method for correction of slope-induced errors in ice-sheet heights derived from radar altimetry"

_The Cryosphere, 2021_

## Author Comment (AC1)

Response to Referee 1 on tc-2021-176

First, we would like to thank the referee for reviewing and commenting the manuscript. Please find the item-by-item reply below, with the original comments in *italics* and the responses in blue. The suggested changes will be implemented in the revised text.

*This manuscript describes a new approach for relocating radar altimetry measurements acquired over ice sheets; one of the most important processing steps for retrieving reliable surface elevation measurements. The authors outline the method, together with a proof-of-concept study whereby the approach is applied to one year's worth of CryoSat-2 LRM measurements over the interior of Greenland. They perform validation relative to ICESat-2 measurements and an independent DEM, alongside a sensitivity analysis to explore some of the inherent assumptions within their approach.*

*I found the manuscript very interesting; the proposed methodology is novel and definitely has the potential to improve upon current approaches documented in the scientific literature and implemented within ESA's ground segment. I therefore believe that it will be of interest to the subsection of The Cryosphere's readership that have an interest in radar altimetry processing techniques over ice sheets, ice caps and glacier surfaces. That being said, I believe that there is still some additional work required to (1) convincingly demonstrate the superior performance of the method relative to existing approaches, and (2) to provide the necessary level of methodological detail required to adequately document this promising new method. Without this, I am left feeling that I have a glimpse of an exciting new approach, but have many unanswered questions that prevent me from being fully convinced that it delivers the improvements that the authors claim. I hope that, by addressing these points, the authors will be able to provide a more compelling demonstration for The Cryosphere's readership. I have detailed these major comments below and would like to see each of them addressed in the revisions. Following these comments I have also listed a number of more minor points, which I hope will help to improve the clarity of the manuscript. Finally, I would recommend that the manuscript undergoes a thorough check for grammatical errors, as there were a considerable number throughout.*

We thank the referee for the constructive review and suggestions. We regret that our explanations were not always clear. Based on some of the comments, we found that some important (technical) details regarding the methods and the setup of the experiments were missing. We will elaborate on this below and clarify that with relevant details in the revised manuscript. We also found a small flaw in our implementation of the point-based method (Roemer et al., 2007). This caused that the refined search of the impact point on a 10 meter grid did not work as expected. Hence, the impact point we found was the impact point on a 100 m grid. After correcting the code, we found an improved performance of the point-based method. Still, though, our LEPTA method shows the best performance when compared to ICESat-2 data. Please see our detailed responses below.

**Major comments**
- ***Performance of LEPTA relative to other approaches.***
*The authors compare LEPTA to the ESA L2I product, and their own in house versions of the slope correction method and the Roemer et al. (2007) relocation method. Whilst the statistics*

*show the superior performance of LEPTA, I am left with several important questions relating to the implementation of other approaches, which make it difficult to determine whether they have been implemented optimally; i.e. whether a better implementation could have yielded improved results more closely matching the performance of LEPTA. Specific points that I would like to see addressed are as follows:*

- *For ESA L2I – have any of the quality flags included within the product been applied? More L2I data are available than for the in house methods and this makes me wonder whether stricter quality control has been applied in the latter, e.g. the waveform filtering mentioned on line 289. In other words, that some of the improvement of LEPTA relative to L2I is not due to the method used slope correction, but simply down to the quality control applied.*

This is a good point. The quality flags indicated in Section 4.3.3 of Bouzinac (2012) are used to exclude flagged data. Furthermore, we reject a waveform if i) the integrated power exceeds a threshold (defined as 150 by the software default) ii) in case the normalised power in the first 10 range bins is larger than 0.2, or iii) our peak-algorithm fails to identify a peak in the waveform. In the revised version of the manuscript, we will apply the latter criterion also to the L2I data (was not the case earlier). Preliminary analysis shows that these quality flags result in the removal of some large outliers and reduce the number of L2I data points to the same amount as obtained for the in-house processed L2 datasets. As such, also the statistics of Table 1 will improve (which, according to the referee's comments, will be replaced by cumulative distribution figures). So, indeed, part of the improvement was due to the applied quality control. At the same time, however, LEPTA still shows significant improvement compared to ESA L2I. We will add the necessary details regarding the applied quality control to Section 2.1 and update all the figures.

- *For the authors' in house 'slope correction' method – the results, e.g. as shown in Fig 4, indicate far worse performance than the ESA L2I implementation, and make me concerned that their slope correction method has been implemented sub-optimally. This, combined with point 1 above, means that I do not think that a convincing case has been made to justify the level of improved performance of LEPTA relative to the slope correction approach. This is not to say that LEPTA is not an improvement, but just that I feel that more work is required to justify this convincingly. Specifically, if the authors really believe that the difference between L2I and their in-house implementation relates to the Doppler slope correction, then I would like to see further analysis to demonstrate (1) that this really is the case (i.e. that the Doppler slope correction can be responsible for a difference of this magnitude), and (2) why it does not affect LEPTA in the same way (and should not be incorporated into the LETPA L2 processing). I would also like the authors to state the DEM resolution used for the slope correction (I couldn't seem to find it anywhere), and if it is 900 m or less, to justify why this is an appropriate choice. From my perspective, the 'resolution' should be comparable to the beam limited footprint (i.e. 10's of km), not the pulse limited footprint, because it is preferable to relocate using the large scale slope across the illuminated area. If you use the '900 m' slope at nadir, then there is the risk that the slope you use will not be representative of the average slope across the illuminated area. Indeed I think you could be seeing this effect in Figure 7, where performance improves up to a resolution of 900 m, and raises the question as to whether you would see further improvements if the resolution was increased any more. As such, I would like the*

*authors to either provide a justification to counter the above concerns, or to test this by computing the slope over a larger length scale (comparable to the beam limited footprint) and re-evaluating the performance of their slope-based method.*

Indeed, the DEM resolution we used was 900 m. This choice was motivated by the fact that we wanted to use a resolution that is close to the resolution used by ESA (1 km). In addition, we relied on the results obtained by Levinsen et al. (2016) who showed that over the interior ice sheet, the DEM resolution has little impact. Note that they tested resolutions up to 8 km. Anyway, the concern of the referee is valid. Hence, in the updated manuscript, the chosen DEM resolution for the slope method will be the one for which the differences compared to ICESat-2 results in the lowest median and median absolute deviation. Here, we searched over an interval from 1 to 5 km with steps of 500 m. The optimal resolution we found this way was 2 km. We will add the results to the revised manuscript.

Our statement that '*the difference between L2I and our in-house implementation relates to the Doppler slope correction*' originates from the discussion section of Levinsen et al. (2016). However, in their study they used Envisat Radar Altimetry-2 data. As such, we agree that the statement is not applicable and will be removed in the revised manuscript. To interpret the differences between our in-house slope correction method and the ESA L2I implementation, we used the DEM suggested in CryoSat-2 Baseline D Handbook *(https://earth.esa.int/documents/10174/125272/CryoSat-Baseline-D-Product-Handbook, last access: Dec. 3 2021)* and a 25% threshold to try to obtain similar results to ESA L2I products. However, the agreement compared to the ESA L2I product did not change (please see the attached figures). This makes us conclude it does not depend on the DEM or the OCOG threshold.

At this stage, we can only state that our results are in line with Levinsen et al. (2016). They report differences (Table I+II Levinsen) between 'ESA relocation' and their slope method using a 1 km DEM of approximately 25/10 m (median) in a steep/smooth region, respectively.

[Figure]

*Figure 1. Comparison between in-house implementation using 2 km resolution ArcticDEM and 20% threshold for OCOG retracker (left panels), in-house implementation using the 1 km resolution DEM derived from Helm et al. (2014) as described in the CryoSat-2 Baseline D Handbook and 25% threshold for OCOG retracker (middle panels), and ESA L2I products (right panels). The visualised height difference is defined as CS2-ICE2.*

- *For the authors' point-based approach, I find the magnitude of the bias surprising, e.g. as shown in Figure 4, and that there is a general lack of detail or discussion required to assess whether this is due to the implementation of the approach. In particular, I cannot find any information relating to the search area that the authors have used; i.e. the illuminated area on the ground where they assume the leading edge reflection could have come from. It would be reasonable to base this upon the 3 dB beamwidth of the instrument, but it is not clear to me what the authors have used. As such, my concern is that an inappropriate choice could lead to a bias in the 'point-based' solution; for example if the criteria used is too strict, and does not allow for the POCA to be sufficiently far away from nadir. I would therefore like to see the authors (1) state what criteria is used, (2) justify why it is appropriate and not impacting the accuracy of the results, and (3) dependent upon these points, consider whether the performance of their point-based approach should be re-evaluated with a refinement to the allowed relocation distance.*

We agree that the description of the search area should be improved. In the manuscript, we assumed a square shape of the PLF (similar to Roemer et al., 2007) with width of 2 km. The BLF was assumed to be a square of 16 x 16 km. These values are close to what is reported by Hai et al., 2021: '*Considering the average altitude of 730 km, an average antenna beam widthof 1.1296° (the antenna shape of CryoSat-2 is an ellipse (Bouzinac, 2015)), the pulse length of 3.125 ns, and flat terrain, the beam-limited illuminated area (BLF) should be 14,393 m in diameter, and the smaller PLF size should be 1,654 m in diameter*'. In the revised

manuscript, we will use values in line with Hai et. al., (2021): PLF = 1.65 km and BLF = *14.393 x 14.393 km*, as we consider them more accurate. We will add this information to the revised manuscript and the references. So, the bias in the point-based method is not related to inapproriate choices of the BLF/PLF.

We agree that the magnitude of the bias in the point-based method is large. This bias, though, seems to be partly introduced by a flaw in our implementation of the method. The method published by Roemer et al. (2007) includes a refined search step based on a high-resolution grid of 10x10 m which was skipped in our implementation. Preliminary results (please see below) show that with the new settings and revised code, both the median differences and median absolute deviation values are lower.

[Figure]

*Figure 2. Difference between the original in-house implementation (left panels) and the improved in-house implementation (right panels) of the point-based method. The visualised height difference is defined as CS2-ICE2. Both the median and median absolute deviation values have improved.*

- **Choice of delta-r.**
*The choice of delta-r seems rather arbitrary, yet central to the LEPTA approach, and so I would like to see some more discussion relating to this point within the manuscript:*

- *From a theoretical perspective, clearly it would make sense to let delta-r vary according to the width of the leading edge of each waveform. I assume the authors have practical*

*considerations for why they chose not to implement this approach, and I think it would be helpful for readers if they could therefore expand on this within the manuscript, to explain why such an approach was not selected.*

Agreed. First of all, we regret that our description in the manuscript was limited. In fact, what we referred to as the 'end of the leading edge' ($r_{end}$) was defined as $r_{end} = \min(r_{60\_rt}, r_{rt} + \text{delta-r})$, where $r_{60\_rt}$ is the range obtained with a 60% threshold retracker and $r_{rt}$ is the retracked range obtained with the OCOG retracker. Here, the use of $r_{rt} + 3.5$ was to avoid that the search window becomes too large. This typically happens in case the waveform has multiple peaks before it reaches its maximum. Hence, in most cases we already used a width that depends on the waveform as suggested by the referee.

To avoid confusion in the revised version, we will no longer refer to $r_{end}$ as the 'end of the leading edge'. Based on this comment and a later comment regarding the choice of $r_0$, we will modify the definition of $r_0$ and $r_{end}$ to the following: $r_0$ will be defined as $\max(r_{1\_rt}, r_{rt} - \text{delta-r})$ and $r_{end}$ as $\min(r_{90\_rt}, r_{rt} + \text{delta-r})$. In doing so, we have just one parameter in the sensitivity analysis, namely delta-r.

- *I appreciate this is extra work, and therefore I would not insist upon it, but given the central role that the leading edge plays in the LEPTA approach, I think it would be really valuable for the authors to provide some quantitative measures relating to the characteristics of the CryoSat-2 LRM leading edge over Greenland. For example, can you provide statistics relating to the mean and standard deviation of the range spanned by the leading edge? This would provide really helpful context for judging the validity of the range of delta-r considered.*

The mean and standard deviation of the leading edge width (defined as the difference between bin at the peak of the waveform and the bin where the normalised waveform power exceeds 0.05) is attached (please see the figure below). The tiling resolution is again 50 x 50 km (same as height differences). It is certainty true that the leading edge width can vary per location, and this discussion will be added in the supplementary material of the revised manuscript.

[Figure]

*Figure 3. Mean and standard deviation of leading edge width (LeW) defined as $X_{peak}$-$X_{norm>0.05}$, where $X_{peak}$ is defined as the bin index where the normalised power reaches the peak and $X_{norm>0.05}$ is defined as the bin index where the normalised power exceeds 0.05. This range is larger in the margin regions of the LRM coverage than the inland regions.*

- *Without point 2 being addressed, it's not clear to me why delta-r of 2 metres is a reasonable lower bound. I would therefore like to see the sensitivity analysis expanded below 2 metres, or a justification for why this is not appropriate; as, in theory, choosing a lower threshold would seem a sensible approach to ensuring that you always identify terrain corresponding to the leading edge.*

Please see our response above regarding the definition of $r_0$ and $r_{end}$. We agree that a lower delta-r makes sense. In the revised manuscript, we will change the range over which we varied delta-r between 0.5-5 m (instead of 2-5 m in the original manuscript). Note that 0.5 is pretty close to the range resolution (0.469 m). Moreover that in case delta-r = 0.5, i.e., when $r_{end}$ = min($r_{60\_rt}$, $r_{rt}$ + 0.5), $r_{end}$ in most cases equals $r_{rt}$ + 0.5 (as indicated by the dashed black line in the figure below). Preliminary results based on the old definition of $r_0$ and $r_{end}$ show that the use of delta-r = 0.5 influences the median difference by ~1cm compared to ICESat-2 (please see attached figure).

[Figure]

*Figure 4. Median (upper solid curve) and percentage of points where $r_{end} = r_{rt}+delta\text{-}r$ (upper dashed curve), varying as a function of delta-r, and median absolute deviation as a function of delta-r. Outliers based on $10^{th}$ and $90^{th}$ percentiles are removed.*

- *I also suspect that the optimal choice of delta-r might vary significantly spatially; yet this is impossible to assess based upon the median statistics presented. For example, that a delta-r of 2 m or lower might perform much better over simple topography. Given the central role of delta r in terms of the LEPTA approach, I think it would be interesting to produce spatial maps of the type shown in Figure 4 for a LEPTA-delta-r of 1 m and 2 m, to see the extent to which this can improve upon the 3.5 m case already plotted.*

We appreciate the suggestion, and the spatial maps will be added in supplementary materials.

- ***Impact of penetration***

*Throughout the manuscript, the issue of penetration into the snowpack is never mentioned. I do not think it requires further analysis, but I do think it would be helpful to include some discussion related to this phenomenon, and whether or not it has any implications for the LEPTA method; given that LEPTA uses range information from the leading edge, and the leading edge of LRM measurements can be modified by subsurface scattering.*

Agreed to mention the issue. The reason to not mention it was that we carefully selected the threshold in our retracker to avoid penetration. According to Davis et al., (1997), a 20% retracker is a proper choice to determine the absolute ice sheet height. The motivation behind the sensitivity experiment in which we increased the threshold to 50% was in fact to

assess whether in this case we would observe penetration. As can be seen from the results, this is indeed the case. In the revised manuscript we will explicitly mention the issue (in particular in view of the results of the sensitivity experiment).

**Manuscript minor comments**
*Line 2: anomalies in what – mass change, physical properties?*
Physical properties – this mainly refers to the massive melt event in Greenland in 2012, where meltwater was produced and subsequent sub-surface ice lenses were observed. We will clarify this in the revised manuscript.

*Line 4-5: Perhaps I'm misunderstanding, but I think the 'slope' method and 'point-based' (I assume Roemer?) are correcting for undulating topography within the \*beam-limited\* footprint rather than the pulse limited footprint?*
Correct. Will be corrected in the revised manuscript.

*Line 9: Begin -> beginning.*
This will be corrected in the revised manuscript.

*Line 13: 'slope corrected' – I assume this relates to those using LEPTA? This should be made clear.*
True. This will be clarified.

*Line 13: 'Almost negligible' is rather vague – please be quantitative, especially as you quote statistics for the other methods and ESA L2. E.g. is it better than ESA L2 at 0.01 m difference?*
Agreed. Will be changed in the revised manuscript.

*Line 14: Which methods exactly do 0.22 m and 0.69 m refer to? Currently it is not clear. Same applies on line 16.*
Slope and point-based methods. This will be clarified.

*Line 14: Median absolute deviation from what?*
CryoSat-2 vs. ICESat-2. This will be clarified.

*Line 31: surface \*slope\* parameters.*
This will be changed in the revised manuscript.

*Line 31: Beam limited footprint?*
Agreed. This will be changed in the revised manuscript.

*Line 32: 'full height information' is not particularly clear for readers not familiar with the subject – perhaps something like 'uses a topographic model...' would be clearer?*
In the revised manuscript, we will use the suggestion of the referee.

*Line 35: Not clear whether you are referring to pulse limited or beam limited footprint. As a more general point, I would recommend that you make sure that through-out the manuscript that it is unambiguous which you are referring to.*
Agreed.

*Line 54: Please state which CS2 product baseline was used.*
Baseline D. This will be specified in the revised manuscript.

*Line 56: Is the data used inclusive of these end months? Important for future reproducibility.*
We use all data between Jan. 1, 2019 – 31 Dec. 2019. This will be clarified in the revised manuscript.

*Line 66: use -> used.*
This will be corrected in the revised version.

*Line 66: What is 25% more realistic than? Do you have any supporting evidence for this statement?*
More realistic than 10% and 50% as discussed by Davis (1997) and Aublanc et al. (2018). We will clarify the sentence and add the references.

*Line 68: Please explain what you mean by 'a distinguishable noise' and what criteria exactly were used to identify the waveforms that failed this and the 'beginning of leading edge' tests; i.e. so that the reader has sufficient information to be able to reproduce your method, should they wish.*
A waveform is rejected in case: i) the integrated power exceeds a threshold (defined as 150 by the software default) ii) in case the normalised power in the first 10 range bins is larger than 0.2, or iii) our algorithm to identify peaks in the waveform does not return a peak.

*Line 70: 'as \*a\* benchmark'.*
This will be corrected.

*Line 71: 'which has a resolution'?*
This will be corrected.

*Line 84: Please also mention which ATL- product was used.*
ATL06. Will be added to the revised manuscript.

*Figure 1. I find this figure pretty hard to interpret and I think it would benefit from some more attention:*
- *Why is the low resolution DEM only given in the slope method panel?*
Because the low-resolution DEM is only used by the slope method. The other methods rely on a 'high-resolution DEM.'
- *I don't think 'apply the satellite-terrain range' really makes sense.*
We will change the last sentence of the caption into 'The slope method computes a correction based on the surface slopes obtained from a DEM, whereas the point-based method and LEPTA are based on the range between satellite and terrain'.
- *I think 'block mean averaged' could do with more explanation in the caption – presumably you mean the average range over either a square, rectangular or circular search window? What is the radius used in these graphs, or is it just a cartoon drawing to illustrate the concept?*
We will change 'block mean averaged' to 'averaged range over a square'.

Indeed, this figure is just a cartoon to illustrate the concept. We will clarify it by replacing 'Illustration…' by 'Conceptual illustration…'

- *Might be also worth annotating with the true POCA as well?*

We decided not to do so as we do not show the 'true' terrain to avoid the plots getting too busy. Moreover, it is just a conceptual illustration.

*Line 98: 'is the central angle between the satellite and Ps'. I don't think this is very clear. I'm not sure what the 'central angle' means, and also that it is correct. Doesn't it depend upon the instrument boresight, which might not necessarily be pointing at nadir?*

$\Gamma$ = asin( (R sin(slope magnitude))/$R_s$ ) where R is the retracted range measured by the altimeter and $R_s$ the radius of curvature of the ellipsoid at the sub-satellite point and the altitude. To avoid any confusion, we will include the equation given by Bamber 1994.

*Figure 2: This figure feels somewhat rough and not ready for publication:*
- *Where is this data from? A location map would be helpful.*
- *Why does the header say natural neighbour but the caption say nearest neighbour? What is h_ICE? ICESat-2 elevation? If so, how should the statistics be interpreted given that point-based method is much further from the IS-2 track than LEPTA? For example, has a correction been applied to account for the effect of surface slope between the CS2 and IS2 locations?*
- *What is d_min?*
- *How were the ICESat-2 tracks that are plotted selected?*
- *Visually, I think it would be easier for the reader to interpret if the DEM was displayed as a contour map; but this is only a recommendation, not essential.*

The figure will be improved according to the comments.

*Line 117: It's not clear to me why you are dividing by R? Also equation is not numbered.*

This is a mistake from our side; R should have been the area of the PLF ($1.65^2$ km$^2$). We will correct the equation in the updated manuscript. We will also add an equation number.

*Line 123: I think it would be helpful to expand upon this final sentence slightly, as I think it is important to convey this point, as it's your main argument relating to the limitation of Roemer. It is not clear what 'It' refers to in 'It also shows'. For example, I don't think that Roemer uses DEM points other than the POCA within equation 4, rather it is only in identifying the location . This distinction is not clearly articulated with the current wording.*

Agreed. 'It' refers to the not numbered equation. We will clarify the sentence in the revised manuscript.

*Line 134: Please provide justification for why 8 x 8 km is chosen as the search radius for the intersection points. It seems quite an arbitrary choice, with no justification given. For example, why not use something closer to the 3dB beamwidth, which would seem to have a much better physical justification? Otherwise, how can you be confident that you are not incorrectly locating measurements where POCA is greater than 4x4 km from nadir but still within the 3dB beamwidth, and therefore sensitive to the antenna gain pattern? At the very least, I would like to know how many measurements fail to identify DEM points within the search window?*

We agree with the referee that our justification should be improved. In the revised manuscript, we will use a BLF that is consistent to what is used in the point-based method.

That is, we have used an approximately 14.393 x 14.393 km BLF (please see the attached figure).

[Figure]

*Figure 5. LEPTA using the original 8x8 km beam-limited footprint (BLF) and delta-r of 3.5 m (left panels), and the improved 14.393x14.393 km BLF and delta-r of 1.5 m (right panels). The visualised height difference is defined as CS2-ICE2. Median values are improved mainly on the western side of the ice sheet.*

*Line 136: 'In case no DEM grid points are identified...'. Please provide a clearer explanation of what you are doing here, as it seems important but I cannot understand exactly what you are doing here. Is the interval expanded? Or is it shifted? Is this the same as finding the DEM points that are \*closest\* to the retracked range, even if they are not within the search interval? If this is the case then I would like to see some more analysis to support this approach; e.g. are these points commonly at the edge of the 8 x 8 search window? Is there a systematic bias in terms of whether the retracked range is normally higher or lower than the DEM range? I think this is required because it seems like this is somewhat at odds with the central tenet of your method which is to only use points within the leading edge interval, so it's not clear to me why this is justified. It relates to the previous point too – in that the underlying issue might be that in these cases POCA lies beyond the 8 x 8 km search window – and it isn't clear to me that what you are doing here is an appropriate way to correct for this issue.*

The interval is shifted. In the revised manuscript, we will use a BLF of 14.393 x 14.393 km. In case no DEM grid points are identified, we define $r_0$ by the range to the closest DEM point. We will clarify this in the text and add an analysis showing how often we encounter this situation.

*Line 137: Do you mean here that P(x,y) is computed as the average of the x and y coordinate values? If so, is this the mean, median or mode? Using this approach, I guess you could get a P(x,y) that is located outside of the LEPTA search area? Can you comment on this; e.g. how often it occurs and what the implications are?*

$P(x,y)$ *i* is the mean of the x and y coordinate values. This point will always be in the 8x8/14.393x14.393 km search area (in the original/revised manuscript, respectively). What may happen (for all methods) is that the position is in between two equally large but disjoint sets of points. By assessing the minimal distance between the computed impact point and the set of identified DEM points we found that this happens in about 5.7% of the cases. In the revised manuscript, we will assess the impact of this on the error statistics.

*Line 139: Should there be a 1/K averaging in equation 5?*

Correct. This will be corrected in the revised manuscript.

*Line 141: Slight aside and not essential, but do you have any statistics relating to the size of the LEPTA footprint – i.e. the intersect between the leading edge and DEM – it would be really interesting to see how much the reality diverges from the classical footprint size over a flat smooth surface.*

We agree this is very interesting. Apart from some limited case studies we did not assess this in full detail. We would like to make the code publicly available. We welcome the referee to study this aspect.

*Line 142: 'but \*are\* outside'.*

This will be corrected.

*Line 151: Please explain what a 'conceptual assessment' actually means.*

ArcticDEM is not an independent DEM (it is used in computing the corrections). Hence, we do not consider this comparison as a validation. Still we think it is insightful, especially when CryoSat-2 points do not always have a validating ICESat-2 point within the 50 m search range. We thought about clarifying this using the phrase 'conceptual assessment'. In principle, we meant to say that comparing to ArcticDEM is not a real validation, since it is not an independent dataset (which has been used to correct for the slope). We will clarify that in the revised version.

*Line 153: It's not clear to me how meaningful the median statistic is, given the effective timestamp of the ArcticDEM. I.e. isn't ArcticDEM referenced to ICESat, and in which case surely you need to account for the intervening elevation change of the surface?*

We agree that some care is needed in the interpretation of the median difference when comparing the ArcticDEM elevations and the Crysosat-2 elevations as i) there are indeed time stamp differences (e.g. due different moment of to ArcticDEM observations, co-registration with Icesat, etc) and ii) ArcticDEM is not an independent DEM validation data set as it is being used in the corrections. Therefore, we opted to call it DEM *evaluation* instead of *validation*. Nevertheless, we think the comparison with ArcticDEM elevations still adds useful information. Therefore, we opt to keep it in the paper, but we will add some discussion to the revised manuscript about the difference between validation/evaluation. A correction for

temporal elevation changes is not possible at the offered ArcticDEM resolution. In the validation with ICESat-2 data this 'ambiguity' is avoided.

*Line 160: In the case of nearest neighbour, is a correction applied to account for the effect of surface slope between the CS2 and IS2 locations? If not, why not and what are the implications? Given that ArcticDEM is already integrated into your processing flows, I assume it would be pretty simple to do this.*
So far we did not account for this effect. We initially considered that i) the maximum allowed distance is 50 m, which probably could not be significant for the LRM coverage, and ii) we would like to keep the validation independent of ArcticDEM. In the revised manuscript, we will follow the suggestion of the referee to compute the correction from ArcticDEM.

*Line 167: 'h_DEM / h_ICE2' – replace '/' with 'or' to avoid any ambiguity with a division operator.*
Agreed. This correction will be implemented.

*Line 169: Would it not make sense to also consider sensitivity to how the start of the leading edge is defined? Surely this is relevant too?*
Agreed. In the revised version, we will change the definition of $r_0$ and $r_{end}$. By doing so, we will also assess the sensitivity of the start point of our search window.

*Line 186: 'best' relative to what – I assume you mean of all methods, but it could be construed as ArcticDEM vs IS-2, so worth making clear.*
Indeed. This will be specified in the revised manuscript.

*Table 1:*
- *'Before' and 'after' are not used in the table, so I would recommend not needing to refer to them in the caption.*

Agreed. Will be removed.

- *Please state whether differences are calculated as CS2 – ref, or ref – CS2.*

CS2-ref. It will be added.

- *I'm not sure how useful it is to list all the percentiles in a table. Have you considered showing these as a cumulative distribution figure instead? I think this would be much easier for the reader to interpret.*

We agree with the suggestion of the referee. A figure will be added in the revised manuscript.

*Line 203: Suggest 'most ideal' -> 'performs the best'.*
Agreed.

*Fig 3: Comparing the LEPTA and L2I pdf's it looks like the main benefit form LEPTA is to reduce positive rather than negative differences. Any thoughts on why this might be? Could the lack of impact on the negative differences be due to the relatively large delta-r leading to DEM elevations beyond the leading edge being included – i.e. a smaller delta-r might deliver improvements here as well? I guess it would be fairly clear by looking at the full pdf in the sensitivity analysis, rather than just the central value?*

The observation of the referee is correct. We do not have a full explanation. However, based on how we defined $r_{end}$, the explanation given by the referee cannot be true. We agree with the last suggestion of the referee. In the revised manuscript, we will elaborate the analysis in this respect.

*Line 208: I would recommend using 'positive' and 'negative' elevation differences, rather than 'right' and 'left' side of the median.*
Agreed.

*Figure 4: It seems that LEPTA is much more clearly the best performer when compared to IS-2, rather than ArcticDEM in Figure 3. It is not clear to me why this is the case – is it linked to spatial coverage, differences in the timestamp of ArcticDEM relative to IS-2, or something else? I think it would be helpful for the authors to expand upon this here.*
In our view, this is mainly caused by i) the higher quality of the ICE-2 data compared to ArcticDEM, and ii) the comparison with ICE-2 data is done based on measurements that are acquired around the same time the Cryosat-2 data were acquired. Indeed, the timestamp of ArcticDEM is different. We will include this discussion in the revised manuscript.

*Line 218: Also covered in previous points. I'm not sure it is the timestamp of the optical images that is important – isn't it the data used to provide the absolute reference?*
We agree, and we will clarify it in the discussion.

*Fig's 5-7: Captions should state what it is the median and median absolute deviation of – i.e,. CS2-minus-IS2.*
This will be improved.

*Line 229: More detail needed – is this for the full dataset or a subset? Is this with outliers removed?*
The full dataset with outliers removed. This will be specified in the revised manuscript.

*Line 230: It's not very clear to me how this choice of 2-5 m actually relates to the properties of the leading edge. I think it would help to justify this choice in the minds of the readers, if the authors could describe the typical width of the leading edge, and show that delta-r is a sensible choice within this context. For example, with the current analysis as it is presented, I am left wondering how common it is for the leading edge to be less the 20% OCOG + 2 m; i.e. to lie outside of the range tested. From a theoretical expensive I could see that a delta-r value of 0.5-1 m could make sense, but there is no analysis to explain why this parameter range was not explored; nor indeed why the actual range of ranges spanned by the leading edge of each waveform was used. Did the authors evaluate what happened when delta-r < 2 metres?*
See our response to the major commens concerning the 'choice of delta-r' above.

*Line 244: I'm interested in why the sensitivity to a bias in the DEM is not symmetrical about zero. Can the authors expand upon this point; i.e. why having a biased-low DEM has very little effect, but biased-high does? Is this somehow connected to a generous choice of delta-r, i.e. that at 3.5 metres, it is actually including a significant buffer beyond the leading edge, such that when you bias the DEM low the true POCA still remains within the delta-r range? I think a slightly more in-depth evaluation and discussion for the observed behaviour would be useful*

*here in terms of understanding the method, rather than a simple 1 paragraph summary of the sensitivity results with minimal interpretation.*
Agreed. The revised manuscript will include a more in-depth evaluation and discussion.

*Line 254: Again, I think the manuscript would benefit from critical interpretation here, rather than simply reporting the bare results. For example, can the authors expand on why the point based approach degrades so quickly with increasing resolution – is it due to topographic peaks being smoothed? Wouldn't you expect the point-based approach to tend towards the slope based approach; i.e. with sufficient smoothing then you remove all high frequency topography and are just left with the long wavelength slope?*
Agreed. We will add this discussion to the revised manuscript.

*Line 259: 'reduces with 30 cm' – doesn't make sense.*
Will be changed into 'is reduced by'.

*Line 261: 'are slightly more off' – please rephrase this more precisely.*
We will rephrase this.

*Line 261: 'choice of retracker \*threshold\*' – I don't think you have compared different retrackers?*
Agreed. This will be corrected.

*Table 2: Does this suggest that LEPTA is more sensitive than the other methods to choice of threshold, for the median absolute deviation; i.e. when 50% is chosen then its performance is comparable to the point based approach? Do you think this ties into delta- r; i.e. if you choose a higher threshold then you are including more terrain at large ranges beyond the leading edge, which might degrade the LEPTA solution in a way that doesn't happen for the point based approach?*
We agree, and detailed discussion will be added to the revised manuscript.

*Line 264: Most of this opening paragraph seems simply to be repeating and summarising the results; i.e. not adding new insight, as I would expect in a discussion section.*
Agreed. This paragraph will be re-written.

*Line 266: I don't think the wording 'in terms of spatial patterns, the LEPTA method outperforms' is very clear; i.e. what it means for one pattern to outperform another. Please consider rewording.*
This paragraph will be re-written in the revised manuscript.

*Line 268: CryoSat-2 \*LRM\* height estimations'.*
This will be corrected.

*Line 268: 'Our results show moreover that the method is not very sensitive to changes in the definition of the end of the leading edge as it shows only millimetre-level uncertainties for the corrected heights when including multi-metre uncertainties on the definition of the end of the leading edge.' I don't think this statement is accurate – If I understand correctly, then I think you see mm-level differences on the median bias averaged over the whole LRM zone in 2019;*

*but this is very different to saying that the height measurements themselves will only vary by mm when you change delta-r by metres – I'm not sure that you have demonstrated this?*
Agreed. In the revised manuscript we will add some statistics on how the 3D position of the height measurements themselves will change (also lat/lon will change).

*Line 271: "the definition of the leading edge should be adjusted accordingly" – please clarify what you mean here. How do you adjust the definition of the leading edge, and what does it mean to be 'adjusted accordingly'?*
Indeed, we can only adjust the definition of the leading edge in case one knows the bias. In reality this is not the case. Hence, our statement is pointless and will be removed from the manuscript.

*Line 273: Please explain why you recommend a high resolution DEM when it has little impact on the method. Also what counts as 'high resolution'?*
This indeed looks a bit odd. From a conceptual point of view, we would opt for the highest resolution. We experimented with coarser resolutions (>900m) and found that the bias increases when the DEM resolution becomes coarser than 2 km. We will add these results to the manuscript and reformulate our statement.

*Line 276: 'the importance of \*accurately determining the\* impact point'?*
This will be corrected.

*Line 277: Do you show that your point based approach outperforms the slope based method? There seems to be a pretty large bias in the former, compared to the latter?*
The point-based approach outperforms the slope-based approach in terms of the median absolute deviation. This will be specified in the revised manuscript.

*Line 286: I am unconvinced by the attribution of differences to the Doppler slope correction – see previous major comment. Please provide more justification that this could be the source of such large differences and, if indeed this is the case, why (1) it does not affect LEPTA, and (2) you don't correct for it in your L2 processing.*
We agree with the referee. The statement originates from the discussion by Levinsen et al. (2016). In their study they used, however, Envisat Radar Altimetry-2 data. As such, the statement is not applicable and will be removed in the revised manuscript.

*Line 289: I think more explanation of how you filter waveforms is required, and how you can be sure that some of the performance improvement you see in LEPTA relative to ESA L2I is not simply due to the fact that you are applying stricter filtering criteria.*
Agreed. Please see also our response to one of the referee's major comments and the one to the comment on Line 68.

*Line 290: I'm not sure I understand this – doesn't it contradict line 136 where you say you adjust the interval if no DEM points are within the leading edge range?*
This sentence bypassed our proofreading and was not supposed to be in the manuscript. Will be removed in the revised version.

*Line 293: I don't think the opening sentence makes sense – applying to what? Also needs to make clear this is for non-interferometric data only.*
Agreed. Sentence will be rephrased to: 'Reducing slope-induced errors is of key importance when processing LRM data over ice sheets'.

*Line 299: Recommend that you do not need to start a new paragraph here.*
This will be implemented in the revised manuscript.

*Line 301: 'by the begin and end of the leading edge' – I don't think this is strictly correct; i.e. you do not limit yourself to the actual end of the leading edge.*
Agreed. Will be rephrased in revised manuscript.

*Line 304: 'almost identical' and 'good improvement' – it would be much more helpful to provide quantitative measures here.*
Agreed. Revised manuscript will include quantitative measures.

*Line 305: You don't evaluate LEPTA at the margins, as your analysis is restricted to the LRM zone only.*
Here, we referred to the margins of the LRM zone. This will be clarified in the revised manuscript.

*Line 306: 'radar altimetry'.*
This will be corrected.

*Line 306: I think somewhere you should flag that you have only assessed performance (1) in the interior, and (2) for Greenland. Therefore it still remains to be shown how the method performs over the more complex ice margin terrain and also over Antarctica. Obviously your thoughts on whether you expect comparable, better or worse performance would be of interest to the reader, and this might fit better within your discussion section.*
We agree. We have assessed LEPTA separately over Antarctica (although using a 8 x 8 km BLF), and the results are similar to Greenland i.e. closer to ICESat-2 measurements in East Antarctica but more biases in West Antarctica. However, based on the improved implementation mentioned above, this experiment could be added as supplementary material if the referee is interested and the Editor allows.

**References**
Bamber, J. L.: Ice sheet altimeter processing scheme, International Journal of Remote Sensing, 15, 925–938, https://doi.org/10.1080/01431169408954125, 1994.

Bouzinac, C.: CryoSat Product Handbook, Tech. rep., ESA, available online at https://earth.esa.int/documents/10174/125272/CryoSat_Product_Handbook, 2012, accessed Apr. 6, 2020.

Hai, G., Xie, H., Du, W., Xia, M., Tong, X., and Li, R. Characterizing slope correction methods applied to satellite radar altimetry data: A case study around Dome Argus in East Antarctica, *Advances in Space Research*, *67*(7), 2120-2139, https://doi.org/10.1016/j.asr.2021.01.016, 2021.

Levinsen, J. F., Simonsen, S. B., Sorensen, L. S., and Forsberg, R.: The Impact of DEM Resolution on Relocating Radar Altime-try Data Over Ice Sheets, IEEE Journal of Selected Topics in Applied Earth Observations and Remote Sensing, 9, 3158–3163, https://doi.org/10.1109/jstars.2016.2587684, 2016.

Nilsson, J., Gardner, A., Sandberg Sørensen, L., and Forsberg, R.: Improved retrieval of land ice topography from CryoSat-2 data and its impact for volume-change estimation of the Greenland Ice Sheet, The Cryosphere, 10, 2953–2969, https://doi.org/10.5194/tc-10-2953-2016, 2016.

Roemer, S., Legrésy, B., Horwath, M., and Dietrich, R.: Refined analysis of radar altimetry data applied to the region of the subglacial Lake Vostok/Antarctica, Remote Sensing of Environment, 106, 269–284, https://doi.org/10.1016/j.rse.2006.02.026, 2007.

Tilling, R. L., Ridout, A. & Shepherd, A.: Estimating Arctic sea ice thickness and volume using CryoSat-2 radar altimeter data, Advances in Space Research, 62, 1203-1225, https://doi.org/10.1016/j.asr.2017.10.051, 2018.

---

## Author Comment (AC2)

Response to Referee 2 on tc-2021-176

First, we would like to thank the referee for reviewing and commenting the manuscript. Please find the item-by-item reply below, with the original comments in *italics* and the responses in blue. The suggested changes will be implemented in the revised text.

*The authors present a novel method of correcting slope-induced biases in radar satellite altimetry. A major challenge in assessing elevation and elevation changes of ice sheets. First, I have to applause the authors for revisiting this challenge, which has been a considerable error source in radar altimetry since the early work by (Brenner et al., 1983). Novel strategies for dealing with this issue are of interest to the radar community, but I will let it be up to the editor to decide if the topic is within the scope of "The Cryosphere".*

*I share many of the same general concerns as the first reviewer, and the following review will mainly supply additional comments. However, first I would like to highlight a couple of common issues also raised by the first reviewer. 1) Impact of radar penetration. Operating in the LRM area of Greenland, one would expect a considerable difference between the raw elevation measurements derived by leading-edge retracking at >10-20% and a validation dataset of real surface elevation observations. Hence, before venture into assessing the biases, this needs to be addressed. I cannot see any mentioning of surface penetration in the paper. 2) Performance of LEPTA relative to other approaches. The limited description of the implementation of both the LEPTA and reference methods leaves the readers with whether the observed differences are due to the method or implementation. 3) The figures are not of publication quality:*

We thank the referee for the cronstructive review and suggestions. Based on some of the comments, we found that some important (technical) details regarding the methods and the setup of the experiments were missing. We will elaborate on this below and clarify that with relevant details in the revised manuscript. We regret that our explanations were not always clear, and will tackle that in the revised version with specific attention to the three points raised:

1) We will improve the discussion of the role of penetration in the revised manuscript (see also detailed response to Referee 1).
2) The description of the methods will be extended to better show that the results can be attributed to the methods and not their implementation (see also detailed response to Referee 1).
3) Figures will be updates based on the suggestions.

- *The figure is hard to follow from the caption. Besides the equations, the main text only offers the figure to be "methods are briefly illustrated". What is "briefly illustrated" I suggest adding an extensive description of the model flow, both in the main text and caption.*

We understand the concern of both referees. Figure 1 and the caption will be revised.

- *This figure could be one image, I guess h_ice should be h_ice2, what is d_min. Text inside the figure should be avoided as much as possible. What is the geographical location of the plot?*

Indeed, h_ice should have been h_ICE2. d_min is the distance to the nearest ICESat-2 point. We will change figure 2 in line with the suggestions of the referees.

- *This is a fine figure, however, see the following comments about grid sizes.*
- *Same as above*

Please see our later comment about grid size.

- *Could have been one figure, with a double y-axis. The curves show steps, which suggest the tested delta r values to be too coarsely spaced.*

Both panels will be integrated in one figure. Following the suggestion of Referee 1, new thresholds are added (0.5–2 m) which have changed the scale of the figure. We do not agree with the referee that the delta-r values are too coarsely spaced, as one cannot expect a continuous line when the number and locations of the points change for every choice of delta-r.

- *Work is needed to better resolve the signals in this illustration; the flat lines do not offer much information.*

We agree that the flat lines do not add a lot of information, but this is exactly what we would like to show: the slope method and point-based method are not affected by a bias in the DEM, whereas LEPTA is.

*This leaves me with the following suggestions for improvements.*
- *When assessing the performance of retrackers an informative measure is "slope vs. elevation bias". A better-performing retracker will have a flat response to an increase in the surface slope. Assessing this response would be beneficial for the paper.*

We agree with the 'better-performing retracker will have a flatter response' statement and this is actually also what we observe in Figure 4 with an increase of median absolute deviation from the interior of the ice sheets towards the edges, where the slopes generally are steeper. This increase towards the edges is especially strong for the ESA L2I and slope method and to a lesser extent for the point-based and LEPTA methods. In this comparison, the LEPTA method again performs best. This as such shows that LEPTA has flatter response vs. slope. We prefer to keep the analysis as it is (i.e., based on the maps), while simultaneously adding an additional description of better performance vs. slope based on the maps instead of adding an additional figure. The reason for this is that computing the plot asked for by the referee leaves with the ambiguity on how to define the surface slope (i.e., over which resolution it should be computed) and in which point (i.e., the nadir point or the impact point).

- *The differences between the different methods should be judged in terms of statistical significance.*

In general, we agree with this comment. Doing so to assess the significance of the differences in the median requires, however, the differences to be normally distributed. From the top panels in figures 3 and 4 we can see that this is not the case. Hence, doing a t-test does not make sense. Therefore, we prefer to show the actual distributions so the reader can see the differences instead of relying on a t-test that violates the normal distribution assumption.

- *How does the gridding of 50x50 km tiles influence the results? Why is the point-based method the only method missing data in the trunk of Jakobshavn isbrae? Why is there no data for all the methods east of the line from 79fjord to Helheim of figure 3? The data coverage seems different in figure 4.*

The difference in data coverage between figures 3 and 4 originates from a minor issue in the plotting which will be solved in the revised manuscript. The fact that we lack data in the trunk of the Jakobshavn isbrae is caused by the fact that outliers are removed (please see attached figure).

[Figure]

*Figure 1. Median (upper panels) and median absolute deviation (lower panels) with all points (both with and without outliers) included.*

Regarding our choice to use 50x50 km, this was a compromise between visualization and robustness of the statistics. Switching to 25x25 km tiles (please see map below) results in more empty tiles (because we lack data points) and enhanced pepper and salt effect, but the main conclusions remain the same.

[Figure]

*Figure 2. Comparison between using a 25 km tile (upper figure) and 50 km tile (lower figure).*

- *Table 1, for the discussion it would be informative to also have the arcticDEM vs. ICESat statistics.*

We agree this can be informative. We can't however add this to Table 1 as it would become too large. We will add a table to the Appendix.

- *Table 2, The surface penetration biases may relate to the retracking threshold chosen. How is the statistics changing between valid choices? (10%-90%)*

Agreed. Note, this was our main motivation for the sensitivity analysis presented in section 4.6. In the revised manuscript, we will use a broader range of thresholds and discuss the issue in more detail.

*Minor comments:*
*L2: What is "assessing snow/ice anomalies"*
This statement refers to the 2012 melt observed by Nilsson et al. (2015). It will be rephrased in the revised manuscript.

*L14: Is the difference between the 1cm and 0cm bias between LEPTA and ESA significant and therefore needs to be differentiated from the performances of LEPTA. I think that it is the standard deviation that is the important measure.*
This sentence will be reformulated. In our view, the results show that the median differences need a careful interpretation. Indeed, penetration plays a role. At the same time, the results suggest that part of the observed bias is due to the applied slope correction method.

*L16: Reformulated: "we recommend the LEPTA method for obtaining... "*
This will be changed in the revised manuscript.

*L17: What is complex topography? The work is done in the LRM area.*
We will rephrase 'especially in regions with complex topography.' to 'especially towards the margins of the LRM area where the surface slopes increase.'

*L23: Concerning elevation change, you could add a reference to (Hurkmans, Bamber, and Griggs, 2012)*
This will be added.

*L51-52: Suggestion to move this to the last part of the introduction.*
Agreed.

*L79: Why not use the official releases of the downsampled ArcticDEM?*
Because in this way we can define the resolution ourselves.

*L84: Is it ATL03 or ATL06 being used? From the link it seems to be ALT06, is there a bias of using the downsampled product?*
It is indeed ATL06, which has a known geolocation accuracy/bias of less than 10 m (https://nsidc.org/sites/nsidc.org/files/technical-references/ICESat2_ATL06_Known_Issues_v005.pdf, last access Dec. 3, 2021). In the revised manuscript, we will add these details.

*L89: This should be moved to the acknowledgment.*
This will be changed in the revised manuscript.

*L92: This sentence needs to be elaborated.*
Agreed. This will be changed in the revised manuscript.

*L134: 8x8 km seems small. When looking at the SARIn retracked data from ESA, relocation distances of up to 12 km from the nadir point can easily be found.*
Agreed. In line with the point-based method, we will enlarge it to approximately 14x14 km.

*L157: The bias can be evaluated in monthly intervals, but at some of 50x50 km tiles closer to the coast a seasonal difference in the bias is expected. How is this seen in your data?*
This analysis is part of our ongoing work for future publication where we focus on spatio-temporal variability in penetration depth. We agree that it is interesting, but we consider it outside the scope of this manuscript.

*L159: Why use both the nearest and natural neighbor interpolations? You give some reasoning. However, would the two algorithms not converge in your case, and thereby there is no need for adding a user-defined threshold?*
We prefer to use a natural neighbour interpolation over a nearest neighbour one. Indeed, doing so allows to compensate for the difference induced by the surface slope as the points never overlap exactly. As pointed out in the manuscript, this requires, however, that the CyroSat-2 data point is surrounded by ICESat-2 points. If this is not the case, we rely on a nearest neighbour interpolation (regarding the use of nearest neighbour, we would like to point to our response to Referee 1). The user-defined threshold the referee points to is required to avoid interpolation over large distances. We will clarify this in the revised manuscript.

*L197-198: I guess the eastside is a result of topography? Could you give some insights into the differences on the east and west-side which will be the reasoning for this reported difference.*
The highter median values on the eastern side of the ice sheet are indeed likely due to topography. We will study this in some more detail and include our analysis in the revised manuscript.

*L219: Having a setup at 50km tiles it would be rather easy to take the time-tagged ArcticDEM tiles into the analyses. This might be a large job to undertake this effort, but one or two tiles would be very informative for the analysis.*
We agree that it might be interesting to compare our results with the time-tagged ArcticDEM tiles, but that is indeed an enormous job and difficult as the tiles are not referenced and still include potential offsets and tilts (https://www.pgc.umn.edu/data/arcticdem/, last access Nov. 23, 2021). There is meta-data available to correct them based on ICESat, but then we de-facto replace the time-tag with the ICESat time-tag. Therefore, we think such a comparison with ArcticDEM tiles does not make sense (without enormous pre-processing of the tiles) and we do consider it an unrealistic experiment.

*L238: The observed change in bias is an important observation, please elaborate on this.*
See also our response to Referee 1. We will elaborate on this in the revised manuscript.

*L244: Please clarify the statement: "relative sensitive". Relative is a difficult word as it might be different for you and me.*
Agreed. This will be changed to quantative descriptions.

*L247: "although not directly visible" please improve the figure.*
The figure will be improved.

*L284: Any insights into why ESA outperforms the other methods?*
Unfortunately, we cannot give a conslusive response.

*L290: Please elaborate on this last statement.*
This statement was not supposed to be in the manuscript and will be removed in the revised version.

---

## Author Response (AR1)

**Response to Referees on tc-2021-176**

We appreciate the reviews and comments from both Referees. Please find the response to Referee 1 on pages 1-18, and the response to Referee 2 on pages 19-25.

**Response to Referee 1 on tc-2021-176**

First, we would like to thank the referee for reviewing and commenting the manuscript. Please find the item-by-item reply below, with the original comments in *italics* and the responses in blue. The suggested changes haven been implemented in the revised text.

*This manuscript describes a new approach for relocating radar altimetry measurements acquired over ice sheets; one of the most important processing steps for retrieving reliable surface elevation measurements. The authors outline the method, together with a proof-of-concept study whereby the approach is applied to one year's worth of CryoSat-2 LRM measurements over the interior of Greenland. They perform validation relative to ICESat-2 measurements and an independent DEM, alongside a sensitivity analysis to explore some of the inherent assumptions within their approach.*

*I found the manuscript very interesting; the proposed methodology is novel and definitely has the potential to improve upon current approaches documented in the scientific literature and implemented within ESA's ground segment. I therefore believe that it will be of interest to the subsection of The Cryosphere's readership that have an interest in radar altimetry processing techniques over ice sheets, ice caps and glacier surfaces. That being said, I believe that there is still some additional work required to (1) convincingly demonstrate the superior performance of the method relative to existing approaches, and (2) to provide the necessary level of methodological detail required to adequately document this promising new method. Without this, I am left feeling that I have a glimpse of an exciting new approach, but have many unanswered questions that prevent me from being fully convinced that it delivers the improvements that the authors claim. I hope that, by addressing these points, the authors will be able to provide a more compelling demonstration for The Cryosphere's readership. I have detailed these major comments below and would like to see each of them addressed in the revisions. Following these comments I have also listed a number of more minor points, which I hope will help to improve the clarity of the manuscript. Finally, I would recommend that the manuscript undergoes a thorough check for grammatical errors, as there were a considerable number throughout.*

We thank the referee for the constructive review and suggestions. We regret that our explanations were not always clear. Based on some of the comments, we found that some important (technical) details regarding the methods and the setup of the experiments were missing. We try to elaborate on this below and clarify it with relevant details in the revised manuscript. We also found a small flaw in our implementation of the point-based method (Roemer et al., 2007). This caused an error in the refined search of the impact point on a 10 meter grid and is also the reason why the performance decreased after increasing the DEM resolution. After correcting the error, we found an improved performance of the point-based method. Still, though, our LEPTA method shows the best performance when compared to ICESat-2 data. Please see our detailed responses below.

*Major comments*
- *Performance of LEPTA relative to other approaches.*

*The authors compare LEPTA to the ESA L2I product, and their own in house versions of the slope correction method and the Roemer et al. (2007) relocation method. Whilst the statistics show the superior performance of LEPTA, I am left with several important questions relating to the implementation of other approaches, which make it difficult to determine whether they have been implemented optimally; i.e. whether a better implementation could have yielded improved results more closely matching the performance of LEPTA. Specific points that I would like to see addressed are as follows:*

- *For ESA L2I – have any of the quality flags included within the product been applied? More L2I data are available than for the in house methods and this makes me wonder whether stricter quality control has been applied in the latter, e.g. the waveform filtering mentioned on line 289. In other words, that some of the improvement of LEPTA relative to L2I is not due to the method used slope correction, but simply down to the quality control applied.*

This is a good point. The quality flags indicated in Section 4.3.3 of Bouzinac (2012) are used to exclude flagged data. Furthermore, we reject a waveform if it meets one of the following empirically derived criteria i) the integrated normalised power exceeds 150, ii) the normalised power in the first 10 range bins is larger than 0.2, or iii) no peak is identified in the waveform. In the revised version of the manuscript, the latter criterion is also applied to the L2I data (this was not the case earlier). The implementation of these quality flags results in the removal of some large outliers and reduces the number of L2I data points to the same amount as obtained for the in-house processed L2 datasets. As such, the statistics of Table 1 are also improved and complemented by a cumulative distribution Figure 2 (according to the referee's comments). So, indeed, part of the improvement was due to the applied quality control. At the same time, however, LEPTA still shows significant improvement compared to ESA L2I. We added the necessary details regarding the applied quality control to Section 2.1 and updated the corresponding figures (Figures 3, 4, and Appendix B).

- *For the authors' in house 'slope correction' method – the results, e.g. as shown in Fig 4, indicate far worse performance than the ESA L2I implementation, and make me concerned that their slope correction method has been implemented sub-optimally. This, combined with point 1 above, means that I do not think that a convincing case has been made to justify the level of improved performance of LEPTA relative to the slope correction approach. This is not to say that LEPTA is not an improvement, but just that I feel that more work is required to justify this convincingly. Specifically, if the authors really believe that the difference between L2I and their in-house implementation relates to the Doppler slope correction, then I would like to see further analysis to demonstrate (1) that this really is the case (i.e. that the Doppler slope correction can be responsible for a difference of this magnitude), and (2) why it does not affect LEPTA in the same way (and should not be incorporated into the LETPA L2 processing). I would also like the authors to state the DEM resolution used for the slope correction (I couldn't seem to find it anywhere), and if it is 900 m or less, to justify why this is an appropriate choice. From my perspective, the 'resolution' should be comparable to the beam limited footprint (i.e. 10's of km), not the pulse limited footprint, because it is preferable to relocate using the large scale slope across the illuminated area. If you use the '900 m' slope at nadir, then there is the risk that*

*the slope you use will not be representative of the average slope across the illuminated area. Indeed I think you could be seeing this effect in Figure 7, where performance improves up to a resolution of 900 m, and raises the question as to whether you would see further improvements if the resolution was increased any more. As such, I would like the authors to either provide a justification to counter the above concerns, or to test this by computing the slope over a larger length scale (comparable to the beam limited footprint) and re-evaluating the performance of their slope-based method.*

Indeed, the DEM resolution we used in the original manuscript was 900 m. This choice was motivated by the fact that we wanted to use a resolution that is close to the resolution used by ESA (1 km). In addition, we relied on the results obtained by Levinsen et al. (2016) who tested resolutions up to 8 km and showed that over the interior ice sheet, the DEM resolution has little impact. Anyway, the concern of the referee is valid. Hence, in the updated manuscript, the chosen DEM resolution for the slope method is 2 km, which corresponds to the resolution with lowest median and median absolute deviation differences compared to ICESat-2. Here, we searched over an interval from 1 to 8 km with steps of 1 km. The optimal resolution we found this way was 2 km. The results of this resolution experiment have been added to the revised manuscript. Moreover, to compare with Levinsen et al. (2016), we have extended the experiment to 8 km resolution. Please find details of the experiment in the revised manuscript.

Our statement that '*the difference between L2I and our in-house implementation relates to the Doppler slope correction'* originates from the discussion section of Levinsen et al. (2016). However, in their study they used Envisat Radar Altimetry-2 data. Although they state that "conclusions from this study can be applied to observations from other sensors, such as CryoSat-2 LRM", we agree that our statement lacks supportive reference and is not applicable. It is removed in the revised manuscript. To interpret the differences between our in-house slope correction method and the ESA L2I implementation, we used the DEM suggested in CryoSat-2 Baseline D Handbook *(https://earth.esa.int/documents/10174/125272/CryoSat-Baseline-D-Product-Handbook, last access: Dec. 3 2021)* and a 25% threshold to try to obtain similar results to ESA L2I products. However, the agreement compared to the ESA L2I product did not change (please see the attached figures). This makes us conclude it does not depend on the DEM or the OCOG threshold.

At this stage, we can only state that our results are in line with Levinsen et al. (2016). They report differences (Table I+II Levinsen) between 'ESA relocation' and their slope method using a 1 km DEM of approximately 25/10 m (median) in a steep/smooth region, respectively.

[Figure]

*Figure 1. Comparison between in-house implementation using 2 km resolution ArcticDEM and 20% threshold for OCOG retracker (left panels), in-house implementation using the 1 km resolution DEM derived from Helm et al. (2014) as described in the CryoSat-2 Baseline D Handbook and 25% threshold for OCOG retracker (middle panels), and ESA L2I products (right panels). The visualised height difference is defined as CS2-ICE2.*

- *For the authors' point-based approach, I find the magnitude of the bias surprising, e.g. as shown in Figure 4, and that there is a general lack of detail or discussion required to assess whether this is due to the implementation of the approach. In particular, I cannot find any information relating to the search area that the authors have used; i.e. the illuminated area on the ground where they assume the leading edge reflection could have come from. It would be reasonable to base this upon the 3 dB beamwidth of the instrument, but it is not clear to me what the authors have used. As such, my concern is that an inappropriate choice could lead to a bias in the 'point-based' solution; for example if the criteria used is too strict, and does not allow for the POCA to be sufficiently far away from nadir. I would therefore like to see the authors (1) state what criteria is used, (2) justify why it is appropriate and not impacting the accuracy of the results, and (3) dependent upon these points, consider whether the performance of their point-based approach should be re-evaluated with a refinement to the allowed relocation distance.*

We agree that the description of the search area should be improved. In the manuscript, we assumed a square shape of the PLF (similar to Roemer et al., 2007) with width of 2 km. The BLF was assumed to be a square of 16 x 16 km. These values are close to what is reported by Hai et al., 2021: '*Considering the average altitude of 730 km, an average antenna beam widthof 1.1296° (the antenna shape of CryoSat-2 is an ellipse (Bouzinac, 2015)), the pulse length of 3.125 ns, and flat terrain, the beam-limited illuminated area (BLF) should be 14,393 m in diameter, and the smaller PLF size should be 1,654 m in diameter*'. In the revised

manuscript, we used values in line with Hai et. al., (2021): PLF = 1.65 km and BLF = *14.393 x 14.393 km*, as we consider them more accurate. We have added this information to the revised manuscript and the references. So, the bias in the point-based method is not related to inappropriate choices of the BLF/PLF.

We agree that the magnitude of the bias in the point-based method is large. This bias, though, seems to be partly introduced by a flaw in our implementation of the method. The method published by Roemer et al. (2007) includes a refined search step based on a high-resolution grid of 10x10 m which was skipped in our implementation. New results (please see the revised manuscript or the manuscript with marked-up changes) show that with the new settings and revised code, both the median differences and median absolute deviation values are lower.

- *Choice of delta-r.*
*The choice of delta-r seems rather arbitrary, yet central to the LEPTA approach, and so I would like to see some more discussion relating to this point within the manuscript:*

- *From a theoretical perspective, clearly it would make sense to let delta-r vary according to the width of the leading edge of each waveform. I assume the authors have practical considerations for why they chose not to implement this approach, and I think it would be helpful for readers if they could therefore expand on this within the manuscript, to explain why such an approach was not selected.*
Agreed. First of all, we regret that our description in the manuscript was limited. In fact, what we referred to as the 'end of the leading edge' ($r_{end}$) was defined as $r_{end}$ = min($r_{60\_rt}$, $r_{rt}$ + delta-r), where $r_{60\_rt}$ is the range obtained with a 60% threshold retracker and $r_{rt}$ is the retracked range obtained with the OCOG retracker. Here, the use of $r_{rt}$ + 3.5 was to avoid that the search window becomes too large. This typically happens in case the waveform has multiple peaks before it reaches its maximum. Hence, in most cases we already used a width that depends on the waveform as suggested by the referee.

Based on this comment and a later comment regarding the choice of $r_0$, (in the revised manuscript referred to as $r_{begin}$) we have modified the definition of $r_{begin}$ and $r_{end}$ to the following: $r_{begin}$ is defined as max($r_{1\%}$, $r_{20\%}$ - delta-r) and $r_{end}$ as min($r_{90\%}$, $r_{20\%}$ + delta-r). Here, $r_{1\%}$ and $r_{90\%}$ are the retracked ranges obtained using, respectively, a 1% and 90% threshold retracker (Davis, 1997), $r_{20\%}$ the OCOG retracked range using a 20% threshold, and $\Delta r$ is a user-defined threshold. $\Delta r$ is used to avoid the search range ($r_{end}$ – $r_{begin}$) becomes unrealistically large. For all experiments, we use a value of 1.25 m based on an empirical optimisation of $\Delta r$ (see Section 4.3). Note that based on this definition, , we only consider one parameter in the sensitivity analysis, namely $\Delta r$.

- *I appreciate this is extra work, and therefore I would not insist upon it, but given the central role that the leading edge plays in the LEPTA approach, I think it would be really valuable for the authors to provide some quantitative measures relating to the characteristics of the CryoSat-2 LRM leading edge over Greenland. For example, can you provide statistics relating to the mean and standard deviation of the range spanned by the leading edge? This would provide really helpful context for judging the validity of the range of delta-r considered.*

The mean and standard deviation of the leading edge width (in this experiment defined as the difference between bin at the peak of the waveform and the bin where the normalised waveform power exceeds 0.05) is attached (please see the figure below). The tiling resolution is again 50 x 50 km (same as height differences). It is certainty true that the leading edge width can vary per location. In the revised manuscript, we refer to these results when discussing the results of the first sensitivity analysis. Indeed, these results suggest the use of a spatially varying $\Delta r$ for future works.

[Figure]

*Figure 2. Mean and standard deviation of leading edge width (LeW) defined as $X_{peak}$-$X_{norm>0.05}$, where $X_{peak}$ is defined as the bin index where the normalised power reaches the peak and $X_{norm>0.05}$ is defined as the bin index where the normalised power exceeds 0.05. This range is larger in the margin regions of the LRM coverage than the inland regions.*

- *Without point 2 being addressed, it's not clear to me why delta-r of 2 metres is a reasonable lower bound. I would therefore like to see the sensitivity analysis expanded below 2 metres, or a justification for why this is not appropriate; as, in theory, choosing a lower threshold would seem a sensible approach to ensuring that you always identify terrain corresponding to the leading edge.*

Please see our response above regarding the definition of $r_0$ (= $r_{begin}$) and $r_{end}$. We agree that a lower delta-r ($\Delta r$) makes sense. In the revised manuscript, we have changed the range over which $\Delta r$ is varied (between 0.5-5 m instead of 2-5 m in the original manuscript). We would like to stress that 0.5 m is pretty close to the range resolution (0.469 m). The new definition has now been implemented, and the subsequent results and discussion has been added in the revised manuscript.

- *I also suspect that the optimal choice of delta-r might vary significantly spatially; yet this is impossible to assess based upon the median statistics presented. For example, that a delta-r of 2 m or lower might perform much better over simple topography. Given the central role of delta r in terms of the LEPTA approach, I think it would be interesting to produce spatial maps of the type shown in Figure 4 for a LEPTA-delta-r of 1 m and 2 m, to see the extent to which this can improve upon the 3.5 m case already plotted.*

We appreciate the suggestion, and we have added a comparison between using delta-r ($\Delta r$) = 1 m and $\Delta r$ = 2 m (please also see the discussion regarding Line 268). In new Fig. 6 of the revised manuscript, a 3D comparison between $\Delta r$ = 2 m and $\Delta r$ = 1 m has been shown. To

answer to the referee's comment, changing Δr to 2 m does not affect the results over simple topography by much. However, in the margin regions of the LRM coverage, using Δr = 2 m biases the results towards the northeast, and the derived heights become lower, compared to using Δr = 1 m. As such, we agree with the reviewer that the optimal choice of Δr might vary spatialy. We bring this forward in the discussion of the results of the first sensitivity analysis. In particular, we identified it as a potential futher improvement of the method.

- ***Impact of penetration***
*Throughout the manuscript, the issue of penetration into the snowpack is never mentioned. I do not think it requires further analysis, but I do think it would be helpful to include some discussion related to this phenomenon, and whether or not it has any implications for the LEPTA method; given that LEPTA uses range information from the leading edge, and the leading edge of LRM measurements can be modified by subsurface scattering.*
Agreed to mention the issue. The reason to not mention it was that we carefully selected the threshold in our retracker to avoid *on average* penetration. According to Davis et al., (1997), a 20% retracker is a proper choice to determine the absolute ice sheet height. The motivation behind the sensitivity experiment in which we increased the threshold to 50% was in fact to assess whether in this case we would observe a lowering of the estimated heights. As can be seen from the results, this is indeed the case. In the revised manuscript we have explicitly mentioned the issue (in particular in view of the results of the sensitivity experiment of OCOG thresholds).

***Manuscript minor comments***
*Line 2: anomalies in what – mass change, physical properties?*
Physical properties – this mainly refers to the massive melt event in Greenland in 2012, where meltwater was produced and subsequent sub-surface ice lenses were observed.

*Line 4-5: Perhaps I'm misunderstanding, but I think the 'slope' method and 'point-based' (I assume Roemer?) are correcting for undulating topography within the \*beam-limited\* footprint rather than the pulse limited footprint?*
This has been corrected in the revised manuscript.

*Line 9: Begin -> beginning.*
This has been corrected in the revised manuscript.

*Line 13: 'slope corrected' – I assume this relates to those using LEPTA? This should be made clear.*
True. This has been clarified.

*Line 13: 'Almost negligible' is rather vague – please be quantitative, especially as you quote statistics for the other methods and ESA L2. E.g. is it better than ESA L2 at 0.01 m difference?*
Agreed. This has been changed into millimetre-level in the revised manuscript. The reason why we do not provide the exact number is that the sensitivity analysis shows varying results based on delta-r.

*Line 14: Which methods exactly do 0.22 m and 0.69 m refer to? Currently it is not clear. Same applies on line 16.*

Slope and point-based methods. This has been clarified.

*Line 14: Median absolute deviation from what?*
(Now Line 15) CryoSat-2 vs. ICESat-2. This has been clarified.

*Line 31: surface \*slope\* parameters.*
(Now Line 34) This has been changed in the revised manuscript.

*Line 31: Beam limited footprint?*
(Now Line 36) This has been changed in the revised manuscript.

*Line 32: 'full height information' is not particularly clear for readers not familiar with the subject – perhaps something like 'uses a topographic model...' would be clearer?*
(Now Line 36) Changed into 'uses a topographic model'.

*Line 35: Not clear whether you are referring to pulse limited or beam limited footprint. As a more general point, I would recommend that you make sure that through-out the manuscript that it is unambiguous which you are referring to.*
Agreed.

*Line 54: Please state which CS2 product baseline was used.*
(Now Line 63) Baseline D. This has been specified in the revised manuscript. And the Bouzinac (2012) reference documenting Baseline C has been replaced by the Baseline D documentation throughout.

*Line 56: Is the data used inclusive of these end months? Important for future reproducibility.*
(Now Line 62) We use all data between Jan. 1, 2019 – 31 Dec. 2019. This has been clarified in the revised manuscript.

*Line 66: use -> used.*
(Now Line 87) This has been corrected in the revised version.

*Line 66: What is 25% more realistic than? Do you have any supporting evidence for this statement?*
(Now Lines 66-74) More realistic than 10% and 50% as discussed by Davis (1997) and Aublanc et al. (2018). We have clarified the sentence and added the references.

*Line 68: Please explain what you mean by 'a distinguishable noise' and what criteria exactly were used to identify the waveforms that failed this and the 'beginning of leading edge' tests; i.e. so that the reader has sufficient information to be able to reproduce your method, should they wish.*
(Now Lines 75-77) A waveform is rejected in case: i) the integrated normalised power exceeds 150, ii) in case the normalised power in the first 10 range bins is larger than 0.2, or iii) our algorithm to identify peaks in the waveform does not return a peak.

*Line 70: 'as \*a\* benchmark'.*
This has been removed.

*Line 71: 'which has a resolution'?*
This has been removed.

*Line 84: Please also mention which ATL- product was used.*
(Now Line 95) ATL06. This has been added to the revised manuscript.

*Figure 1. I find this figure pretty hard to interpret and I think it would benefit from some more attention:*
- *Why is the low resolution DEM only given in the slope method panel?*

Because the low-resolution DEM is only used by the slope method. The other methods rely on a 'high-resolution DEM.'
- *I don't think 'apply the satellite-terrain range' really makes sense.*

We have changed the last sentence of the caption into 'The slope method computes a correction based on the surface slopes obtained from a DEM, whereas the point-based method and LEPTA are based on the range between satellite and terrain'.
- *I think 'block mean averaged' could do with more explanation in the caption – presumably you mean the average range over either a square, rectangular or circular search window? What is the radius used in these graphs, or is it just a cartoon drawing to illustrate the concept?*

Changed 'block mean averaged' to 'averaged range over a square'.
Indeed, this figure is just a cartoon to illustrate the concept. We have replaced 'Illustration…' by 'Conceptual illustration…' and hopefully this is clearer.
- *Might be also worth annotating with the true POCA as well?*

We decided not to do so as we do not show the 'true' terrain to avoid the plots getting too busy. Moreover, it is just a conceptual illustration.

*Line 98: 'is the central angle between the satellite and Ps'. I don't think this is very clear. I'm not sure what the 'central angle' means, and also that it is correct. Doesn't it depend upon the instrument boresight, which might not necessarily be pointing at nadir?*
$\Gamma$ = asin((Rsin(slope magnitude))/$R_s$) where R is the retracted range measured by the altimeter and $R_s$ the radius of curvature of the ellipsoid at the sub-satellite point and the altitude. To avoid any confusion, we have included all the equations given by Bamber 1994, including the computation of all other angles and variables.

*Figure 2: This figure feels somewhat rough and not ready for publication:*
- *Where is this data from? A location map would be helpful.*
- *Why does the header say natural neighbour but the caption say nearest neighbour? What is h_ICE? ICESat-2 elevation? If so, how should the statistics be interpreted given that point-based method is much further from the IS-2 track than LEPTA? For example, has a correction been applied to account for the effect of surface slope between the CS2 and IS2 locations?*
- *What is d_min?*
- *How were the ICESat-2 tracks that are plotted selected?*
- *Visually, I think it would be easier for the reader to interpret if the DEM was displayed as a contour map; but this is only a recommendation, not essential.*

The figure has been removed, as it does not add useful information expect that the PLF defined by LEPTA is not rectangular.

*Line 117: It's not clear to me why you are dividing by R? Also equation is not numbered.*
(Now Eq. (11)) This is a mistake from our side; R should have been the area of the PLF ($1.65^2$ $km^2$). We have corrected the equation in the updated manuscript and added the equation number.

*Line 123: I think it would be helpful to expand upon this final sentence slightly, as I think it is important to convey this point, as it's your main argument relating to the limitation of Roemer. It is not clear what 'It' refers to in 'It also shows'. For example, I don't think that Roemer uses DEM points other than the POCA within equation 4, rather it is only in identifying the location . This distinction is not clearly articulated with the current wording.*
(Now Line 155) Agreed. 'It' refers to the not numbered equation. We have clarified the sentences in the revised manuscript.

*Line 134: Please provide justification for why 8 x 8 km is chosen as the search radius for the intersection points. It seems quite an arbitrary choice, with no justification given. For example, why not use something closer to the 3dB beamwidth, which would seem to have a much better physical justification? Otherwise, how can you be confident that you are not incorrectly locating measurements where POCA is greater than 4x4 km from nadir but still within the 3dB beamwidth, and therefore sensitive to the antenna gain pattern? At the very least, I would like to know how many measurements fail to identify DEM points within the search window?*
(Now Line 161) We agree with the referee that our justification needed improvement. In the revised manuscript, we have used a BLF that is consistent to what is used in the point-based method. That is, we have used a 14.393 x 14.393 km BLF and the result has been slightly changed (please see the manuscript with marked-up changes).

*Line 136: 'In case no DEM grid points are identified...'. Please provide a clearer explanation of what you are doing here, as it seems important but I cannot understand exactly what you are doing here. Is the interval expanded? Or is it shifted? Is this the same as finding the DEM points that are \*closest\* to the retracked range, even if they are not within the search interval? If this is the case then I would like to see some more analysis to support this approach; e.g. are these points commonly at the edge of the 8 x 8 search window? Is there a systematic bias in terms of whether the retracked range is normally higher or lower than the DEM range? I think this is required because it seems like this is somewhat at odds with the central tenet of your method which is to only use points within the leading edge interval, so it's not clear to me why this is justified. It relates to the previous point too – in that the underlying issue might be that in these cases POCA lies beyond the 8 x 8 km search window – and it isn't clear to me that what you are doing here is an appropriate way to correct for this issue.*
The interval is shifted. In the revised manuscript, we use a BLF of 14.393 x 14.393 km. In case no DEM grid points are identified, we define $r_0$ ($r_{begin}$) by the range to the closest DEM point. We clarified this in the text (Lines 169-170).

*Line 137: Do you mean here that P(x,y) is computed as the average of the x and y coordinate values? If so, is this the mean, median or mode? Using this approach, I guess you could get a*

*P(x,y) that is located outside of the LEPTA search area? Can you comment on this; e.g. how often it occurs and what the implications are?*

P(x,y) *i* is the mean of the x and y coordinate values. This point will always be in the 8x8/14.393x14.393 km search area (in the original/revised manuscript, respectively). What may happen is that the position is in between two equally large but disjoint sets of points. By assessing the minimal distance between the computed impact point and the set of identified DEM points, we found that this happens in about 5.2% of the cases. In the revised manuscript, we have assessed the impact of this on the error statistics (when the heights are compared with ArcticDEM). It turns out that the median and mean hardly change, but the median absolute deviation and the standard deviation improve (Lines 253-254).

*Line 139: Should there be a 1/K averaging in equation 5?*

(Now Eq. (15)) This has been corrected in the revised manuscript.

*Line 141: Slight aside and not essential, but do you have any statistics relating to the size of the LEPTA footprint – i.e. the intersect between the leading edge and DEM – it would be really interesting to see how much the reality diverges from the classical footprint size over a flat smooth surface.*

We agree this is very interesting. Apart from some limited case studies, we did not assess this in full detail. We would like to make the code publicly available. We welcome the referee to study this aspect.

*Line 142: 'but \*are\* outside'.*

(Now Line 179) This has been corrected.

*Line 151: Please explain what a 'conceptual assessment' actually means.*

(Lines 184-190) ArcticDEM is not an independent DEM (it is used in computing the corrections). Hence, we do not consider this comparison as a validation. Still we think it is insightful, especially when CryoSat-2 points do not always have a validating ICESat-2 point within the 50 m search range. We thought about clarifying this using the phrase 'conceptual assessment'. In principle, we meant to say that comparing to ArcticDEM is not a real validation, since it is not an independent dataset (it has been used to correct for the slope). We clarified this in the revised manuscript.

*Line 153: It's not clear to me how meaningful the median statistic is, given the effective timestamp of the ArcticDEM. I.e. isn't ArcticDEM referenced to ICESat, and in which case surely you need to account for the intervening elevation change of the surface?*

(Now Lines 225 onwards) We agree that some care is needed in the interpretation of the median difference when comparing the ArcticDEM elevations and the Crysosat-2 elevations as i) there are indeed time stamp differences (e.g. due different moment of to ArcticDEM observations, co-registration with Icesat, etc) and ii) ArcticDEM is not an independent DEM validation data set as it is being used in the corrections. Therefore, we opted to call it DEM *comparison* instead of *validation*. Nevertheless, for the reason mentioned in our previous response, we think the comparison with ArcticDEM elevations still adds useful information. Therefore, we opt to keep it in the paper. In the first line of the discussion section, though, we explicitly state 'comparison with ArcticDEM' and 'validation based on ICESat-2'. Indeed, when comparing our data to ICESat-2 this 'ambiguity' is avoided.  A correction for temporal

elevation changes is not possible at the offered ArcticDEM resolution. In Section 4.5, we have assessed the impact of potential biases in the DEM (due to constant ice sheet elevation changes).

*Line 160: In the case of nearest neighbour, is a correction applied to account for the effect of surface slope between the CS2 and IS2 locations? If not, why not and what are the implications? Given that ArcticDEM is already integrated into your processing flows, I assume it would be pretty simple to do this.*
(Now Lines 191-200) In the original manuscript we did not account for this effect. We initially considered that i) the maximum allowed distance is 50 m, which probably could not be significant for the LRM coverage, and ii) we would like to keep the validation independent of ArcticDEM. In the revised manuscript, we have followed the suggestion of the referee to compute the correction from ArcticDEM, and the presented results are from this approach.

*Line 167: 'h_DEM / h_ICE2' – replace '/' with 'or' to avoid any ambiguity with a division operator.*
(Now Line 204) Agreed. This correction has been implemented.

*Line 169: Would it not make sense to also consider sensitivity to how the start of the leading edge is defined? Surely this is relevant too?*
Agreed. In the revised version, we changed the definition of $r_0$ ($r_{begin}$) and $r_{end}$. In the new definition, we consider only one parameter in the sensitivity analysis, namely $\Delta r$. Indeed, this parameter shows up in both the definition of $r_{begin}$ and $r_{end}$. By doing so, we have also assessed the sensitivity of the start point of our search window.

*Line 186: 'best' relative to what – I assume you mean of all methods, but it could be construed as ArcticDEM vs IS-2, so worth making clear.*
(Now Line 233) Indeed. Changed from 'best' to 'better than other methods'.

*Table 1:*
- *'Before' and 'after' are not used in the table, so I would recommend not needing to refer to them in the caption.*

Agreed. These have been removed.

- *Please state whether differences are calculated as CS2 – ref, or ref – CS2.*

CS2-ref. It has been added in the caption.

- *I'm not sure how useful it is to list all the percentiles in a table. Have you considered showing these as a cumulative distribution figure instead? I think this would be much easier for the reader to interpret.*

We agree with the suggestion of the referee. The cumulative distribution curves have been added instead of percentiles in the revised manuscript.

*Line 203: Suggest 'most ideal' -> 'performs the best'.*
(Now Line 251) Agreed.

*Fig 3: Comparing the LEPTA and L2I pdf's it looks like the main benefit form LEPTA is to reduce positive rather than negative differences. Any thoughts on why this might be? Could the lack of impact on the negative differences be due to the relatively large delta-r leading to DEM elevations beyond the leading edge being included – i.e. a smaller delta-r might deliver improvements here as well? I guess it would be fairly clear by looking at the full pdf in the sensitivity analysis, rather than just the central value?*

The observation of the referee is correct. We do not have a full explanation. However, based on how we defined $r_{end}$, the explanation given by the referee cannot be true. We agree with the last suggestion of the referee. In the revised manuscript, we have added the full pdf in the appendix and tried to elaborate the analysis in this respect. However, we cannot conclude whether LEPTA really can reduce positive or negative differences, as it largely depends on delta-r (please see the figure below).

[Figure]

*Figure 3. Full histogram using delta-r = 1 m (upper), delta-r = 1.25 m (middle), and delta-r = 2 m (lower). Largest negative outliers are from delta-r = 1.25 m.*

*Line 208: I would recommend using 'positive' and 'negative' elevation differences, rather than 'right' and 'left' side of the median.*

(Now Line 238) Agreed.

*Figure 4: It seems that LEPTA is much more clearly the best performer when compared to IS-2, rather than ArcticDEM in Figure 3. It is not clear to me why this is the case – is it linked to spatial coverage, differences in the timestamp of ArcticDEM relative to IS-2, or something else? I think it would be helpful for the authors to expand upon this here.*

In our view, this is mainly caused by i) the higher quality of the ICE-2 data compared to ArcticDEM, and ii) the comparison with ICE-2 data is done based on measurements that are acquired around the same time the Cryosat-2 data were acquired. Indeed, the timestamp of ArcticDEM is different. We have included to Section 4.2 in the revised manuscript.

*Line 218: Also covered in previous points. I'm not sure it is the timestamp of the optical images that is important – isn't it the data used to provide the absolute reference?*
(Now Lines 264-268) We agree, and we try to clarify it in the discussion.

*Fig's 5-7: Captions should state what it is the median and median absolute deviation of – i.e,. CS2-minus-IS2.*
This has been specified by adding the reference to the equation used to calculate the difference.

*Line 229: More detail needed – is this for the full dataset or a subset? Is this with outliers removed?*
The full dataset with outliers removed. This has been specified in the captions of Fig. 5 and Figs. 7-9.

*Line 230: It's not very clear to me how this choice of 2-5 m actually relates to the properties of the leading edge. I think it would help to justify this choice in the minds of the readers, if the authors could describe the typical width of the leading edge, and show that delta-r is a sensible choice within this context. For example, with the current analysis as it is presented, I am left wondering how common it is for the leading edge to be less the 20% OCOG + 2 m; i.e. to lie outside of the range tested. From a theoretical expensive I could see that a delta-r value of 0.5-1 m could make sense, but there is no analysis to explain why this parameter range was not explored; nor indeed why the actual range of ranges spanned by the leading edge of each waveform was used. Did the authors evaluate what happened when delta-r < 2 metres?*
Please see our response to the major commens concerning the 'choice of delta-r' above.

*Line 244: I'm interested in why the sensitivity to a bias in the DEM is not symmetrical about zero. Can the authors expand upon this point; i.e. why having a biased-low DEM has very little effect, but biased-high does? Is this somehow connected to a generous choice of delta-r, i.e. that at 3.5 metres, it is actually including a significant buffer beyond the leading edge, such that when you bias the DEM low the true POCA still remains within the delta-r range? I think a slightly more in-depth evaluation and discussion for the observed behaviour would be useful here in terms of understanding the method, rather than a simple 1 paragraph summary of the sensitivity results with minimal interpretation.*
Agreed. The revised manuscript includes a more in-depth evaluation and discussion (Lines 305-312 and Lines 364-371). Regarding the fact that the sensitivity to a bias in the DEM is not symmetrical around zero, we have the following explanation. The impact point is typically in the area where the range between the satellite and the terrain is smallest. Lowering the DEM and thereby increasing the range to the satellite hence results in a reduced number of DEM grid points within the search range ($r_{end}$-$r_{begin}$). In case no points are found, the search range is adjusted (Section 3.1.3). Applying a positive bias, on the other hand, will result in other parts of the terrain being within the search range. We have added that to the revised manuscript.

*Line 254: Again, I think the manuscript would benefit from critical interpretation here, rather than simply reporting the bare results. For example, can the authors expand on why the point based approach degrades so quickly with increasing resolution – is it due to topographic peaks*

*being smoothed? Wouldn't you expect the point-based approach to tend towards the slope based approach; i.e. with sufficient smoothing then you remove all high frequency topography and are just left with the long wavelength slope?*

Agreed. We have expanded the analysis of the results in Section 4.6 and elaborated the discussion in Section 5. However, the degrading results for the point-based method was due to our flaw in the implementation of the method mentioned in the major points above.

*Line 259: 'reduces with 30 cm' – doesn't make sense.*
The whole section has been rewritten.

*Line 261: 'are slightly more off' – please rephrase this more precisely.*
The whole section has been rewritten.

*Line 261: 'choice of retracker \*threshold\*' – I don't think you have compared different retrackers?*
The whole section has been rewritten.

*Table 2: Does this suggest that LEPTA is more sensitive than the other methods to choice of threshold, for the median absolute deviation; i.e. when 50% is chosen then its performance is comparable to the point based approach? Do you think this ties into delta- r; i.e. if you choose a higher threshold then you are including more terrain at large ranges beyond the leading edge, which might degrade the LEPTA solution in a way that doesn't happen for the point based approach?*
The whole section has been rewritten.

*Line 264: Most of this opening paragraph seems simply to be repeating and summarising the results; i.e. not adding new insight, as I would expect in a discussion section.*
The discussion has been rewritten.

*Line 266: I don't think the wording 'in terms of spatial patterns, the LEPTA method outperforms' is very clear; i.e. what it means for one pattern to outperform another. Please consider rewording.*
The whole section has been rewritten.

*Line 268: CryoSat-2 \*LRM\* height estimations'.*
(Now Line 336) This has been corrected.

*Line 268: 'Our results show moreover that the method is not very sensitive to changes in the definition of the end of the leading edge as it shows only millimetre-level uncertainties for the corrected heights when including multi-metre uncertainties on the definition of the end of the leading edge.' I don't think this statement is accurate – If I understand correctly, then I think you see mm-level differences on the median bias averaged over the whole LRM zone in 2019; but this is very different to saying that the height measurements themselves will only vary by mm when you change delta-r by metres – I'm not sure that you have demonstrated this?*
Agreed. In the revised manuscript we have added some statistics on how the 3D position of the height measurements themselves change (also lat/lon will change) in the new Fig. 6.

*Line 271: "the definition of the leading edge should be adjusted accordingly" – please clarify what you mean here. How do you adjust the definition of the leading edge, and what does it mean to be 'adjusted accordingly'?*
This statement was indeed not possible and has been removed from the manuscript.

*Line 273: Please explain why you recommend a high resolution DEM when it has little impact on the method. Also what counts as 'high resolution'?*
This indeed looks a bit odd. From a conceptual point of view, we would opt for the highest resolution. We experimented with coarser resolutions (>900m) and found that the bias increases when the DEM resolution becomes coarser than 2 km. We have added these results to the manuscript and reformulated our statement.

*Line 276: 'the importance of \*accurately determining the\* impact point'?*
(Now Line 337) This has been corrected.

*Line 277: Do you show that your point based approach outperforms the slope based method? There seems to be a pretty large bias in the former, compared to the latter?*
The point-based approach outperforms the slope-based approach in terms of the median absolute deviation. The difference has been specified in the revised manuscript (Lines 259-263).

*Line 286: I am unconvinced by the attribution of differences to the Doppler slope correction – see previous major comment. Please provide more justification that this could be the source of such large differences and, if indeed this is the case, why (1) it does not affect LEPTA, and (2) you don't correct for it in your L2 processing.*
We agree with the referee. The statement originates from the discussion by Levinsen et al. (2016). In their study they used, however, Envisat Radar Altimetry-2 data. As such, the statement is not applicable and is removed in the revised manuscript.

*Line 289: I think more explanation of how you filter waveforms is required, and how you can be sure that some of the performance improvement you see in LEPTA relative to ESA L2I is not simply due to the fact that you are applying stricter filtering criteria.*
Agreed. Please see also our response to one of the referee's major comments and the one to the comment on Line 68.

*Line 290: I'm not sure I understand this – doesn't it contradict line 136 where you say you adjust the interval if no DEM points are within the leading edge range?*
This sentence bypassed our proofreading and was not supposed to be in the manuscript. It is removed in the revised version.

*Line 293: I don't think the opening sentence makes sense – applying to what? Also needs to make clear this is for non-interferometric data only.*
(Now Line 387) The sentence is rephrased to: 'Reducing slope-induced errors is a key correction algorithm when processing LRM data over ice sheets.'.

*Line 299: Recommend that you do not need to start a new paragraph here.*
This has been implemented in the revised manuscript.

*Line 301: 'by the begin and end of the leading edge' – I don't think this is strictly correct; i.e. you do not limit yourself to the actual end of the leading edge.*
(Lines 393-395) Agreed. Has been rephrased to 'The principle of the method is that only the points on the ground that are within range determined by a specific search range that contributes to the rise of the waveform leading edge are used to determine the impact point.'.

*Line 304: 'almost identical' and 'good improvement' – it would be much more helpful to provide quantitative measures here.*
(Now Line 398) The revised manuscript includes quantitative measures.

*Line 305: You don't evaluate LEPTA at the margins, as your analysis is restricted to the LRM zone only.*
(Now Line 400) Here, we referred to the margins of the LRM zone. This has been clarified in the revised manuscript.

*Line 306: 'radar altimetry'.*
(Now Line 404) This has been corrected.

*Line 306: I think somewhere you should flag that you have only assessed performance (1) in the interior, and (2) for Greenland. Therefore it still remains to be shown how the method performs over the more complex ice margin terrain and also over Antarctica. Obviously your thoughts on whether you expect comparable, better or worse performance would be of interest to the reader, and this might fit better within your discussion section.*
We agree. We have assessed LEPTA separately over Antarctica over a month, and please find attached the result (figure below). The altimetry data are acquired only between 1 and 31 Jan., 2019. 50x50 km tiles are applied for statistics and visualisation. The visualised statistics are again the median and median absolute deviation per tile. Tiles with less than 10 acquisitions are not visualised. Differently from Greenland, the interior of Antarctica mainly shows negative median delta-h. High median and median absolute deviation of delta-h values are mainly located in West Antarctica, megadune regions (Michel et al., 2014) and, similar to Greenland, in the margin regions of LRM coverage. Further application and potential improvement of LEPTA over Antarctica can be a focus of future works.

[Figure]

*Figure 4. Median and median absolute deviation of height difference CryoSat-2 corrected with LEPTA and ICESat-2 (delta-h as shown in the manuscript). The visualised results are between 1 Jan. 2019 and 31 Jan. 2019 instead of the whole year 2019, as a whole year*

*contains too many data points. The mapped locations are tiled by the 50x50 km grid. DEM for visualisation is the 1 km resolution DEM from the REMA project, courtesy of the Polar Geospatial Center.*

**References**

Bamber, J. L.: Ice sheet altimeter processing scheme, International Journal of Remote Sensing, 15, 925–938, https://doi.org/10.1080/01431169408954125, 1994.

Bouzinac, C.: CryoSat Product Handbook, Tech. rep., ESA, available online at https://earth.esa.int/documents/10174/125272/CryoSat_Product_Handbook, 2012, accessed Apr. 6, 2020.

Hai, G., Xie, H., Du, W., Xia, M., Tong, X., and Li, R.: Characterizing slope correction methods applied to satellite radar altimetry data: A case study around Dome Argus in East Antarctica, *Advances in Space Research*, *67*(7), 2120-2139, https://doi.org/10.1016/j.asr.2021.01.016, 2021.

Howat, I. M., Porter, C., Smith, B. E., Noh, M.-J., and Morin, P.: The Reference Elevation Model of Antarctica, The Cryosphere, 13, 665–674, https://doi.org/10.5194/tc-13-665-2019, 2019.

Levinsen, J. F., Simonsen, S. B., Sorensen, L. S., and Forsberg, R.: The Impact of DEM Resolution on Relocating Radar Altime-try Data Over Ice Sheets, IEEE Journal of Selected Topics in Applied Earth Observations and Remote Sensing, 9, 3158–3163, https://doi.org/10.1109/jstars.2016.2587684, 2016.

Michel, A., Flament, T., and Rémy, F.: Study of the Penetration Bias of ENVISAT Altimeter Observations over Antarctica in Comparison to ICESat Observations, Remote Sensing, 6, 9412–9434, https://doi.org/10.3390/rs6109412, 2014.

Nilsson, J., Gardner, A., Sandberg Sørensen, L., and Forsberg, R.: Improved retrieval of land ice topography from CryoSat-2 data and its impact for volume-change estimation of the Greenland Ice Sheet, The Cryosphere, 10, 2953–2969, https://doi.org/10.5194/tc-10-2953-2016, 2016.

Roemer, S., Legrésy, B., Horwath, M., and Dietrich, R.: Refined analysis of radar altimetry data applied to the region of the subglacial Lake Vostok/Antarctica, Remote Sensing of Environment, 106, 269–284, https://doi.org/10.1016/j.rse.2006.02.026, 2007.

Tilling, R. L., Ridout, A. & Shepherd, A.: Estimating Arctic sea ice thickness and volume using CryoSat-2 radar altimeter data, Advances in Space Research, 62, 1203-1225, https://doi.org/10.1016/j.asr.2017.10.051, 2018.

**Response to Referee 2 on tc-2021-176**

First, we would like to thank the referee for reviewing and commenting the manuscript. Please find the item-by-item reply below, with the original comments in *italics* and the responses in blue. The suggested changes have been implemented in the revised text.

*The authors present a novel method of correcting slope-induced biases in radar satellite altimetry. A major challenge in assessing elevation and elevation changes of ice sheets. First, I have to applause the authors for revisiting this challenge, which has been a considerable error source in radar altimetry since the early work by (Brenner et al., 1983). Novel strategies for dealing with this issue are of interest to the radar community, but I will let it be up to the editor to decide if the topic is within the scope of "The Cryosphere".*

*I share many of the same general concerns as the first reviewer, and the following review will mainly supply additional comments. However, first I would like to highlight a couple of common issues also raised by the first reviewer. 1) Impact of radar penetration. Operating in the LRM area of Greenland, one would expect a considerable difference between the raw elevation measurements derived by leading-edge retracking at >10-20% and a validation dataset of real surface elevation observations. Hence, before venture into assessing the biases, this needs to be addressed. I cannot see any mentioning of surface penetration in the paper. 2) Performance of LEPTA relative to other approaches. The limited description of the implementation of both the LEPTA and reference methods leaves the readers with whether the observed differences are due to the method or implementation. 3) The figures are not of publication quality:*

We thank the referee for the cronstructive review and suggestions. Based on some of the comments, we found that some important (technical) details regarding the methods and the setup of the experiments were missing. We elaborate on this below and have tried to clarify that with relevant details in the revised manuscript. We regret that our explanations were not always clear, and tried to tackle that in the revised version with specific attention to the three points raised:

1) We disagree with the reviewer that '*one would expect a considerable difference between the raw elevation measurements derived by leading-edge retracking at >10-20% and a validation dataset of real surface elevation observations*'. This is discussed by Davis (1997) and Aublanc et al. (2018). Indeed, locally differences can be observed (see Figures 3 and 4) but on average this is not expected. We elaborated on this in Section 2.1 and highlighted our finding in Section 4.4. Moreover, we changed the way we plotted the median differences per tile to hightlight the median difference per tile varies over the area..
2) The description of the methods have been extended to better show that the results can be attributed to the methods and not their implementation (please see also detailed response to Referee 1).
3) Figures are updated based on the suggestions.

- *The figure is hard to follow from the caption. Besides the equations, the main text only offers the figure to be "methods are briefly illustrated". What is "briefly illustrated" I*

*suggest adding an extensive description of the model flow, both in the main text and caption.*

We understand the concern of both referees. Figure 1 and the caption have been revised.

- *This figure could be one image, I guess h_ice should be h_ice2, what is d_min. Text inside the figure should be avoided as much as possible. What is the geographical location of the plot?*

We decided to not include this figure anymore as it does not provide useful information except that the PLF defined by LEPTA is not rectangular.

- *This is a fine figure, however, see the following comments about grid sizes.*
- *Same as above*

Please see our later comment about grid size.

- *Could have been one figure, with a double y-axis. The curves show steps, which suggest the tested delta r values to be too coarsely spaced.*

Both panels have been integrated in one figure. Following the suggestion of Referee 1, new thresholds are added (0.5–2 m) which have changed the scale of the figure. We do not agree with the referee that the delta-r (Δr) values are too coarsely spaced, as one cannot expect a continuous line when the number and locations of the points change for every choice of delta-r (Δr).

- *Work is needed to better resolve the signals in this illustration; the flat lines do not offer much information.*

We agree that the flat lines do not add a lot of information, but this is exactly what we would like to show: the slope method and point-based method are not affected by a bias in the DEM, whereas LEPTA is.

*This leaves me with the following suggestions for improvements.*
- *When assessing the performance of retrackers an informative measure is "slope vs. elevation bias". A better-performing retracker will have a flat response to an increase in the surface slope. Assessing this response would be beneficial for the paper.*

We agree with the 'better-performing retracker will have a flatter response' statement and this is actually also what we observe in Fig. 4 of the manuscript with an increase of median absolute deviation from the interior of the ice sheets towards the edges, where the slopes generally are steeper. This increase towards the edges is especially strong for the ESA L2I and slope method and to a lesser extent for the point-based and LEPTA methods. In this comparison, the LEPTA method again performs best. This as such shows that LEPTA has flatter response vs. slope. We prefer to keep the analysis as it is (i.e., based on the maps), while simultaneously adding an additional description of better performance vs. slope based on the maps instead of adding an additional figure. The reason for this is that computing the plot asked for by the referee leaves with the ambiguity on how to define the surface slope (i.e., over which resolution it should be computed) and in which point (i.e., the nadir point or the impact point).

- *The differences between the different methods should be judged in terms of statistical significance.*

In general, we agree with this comment. Doing so to assess the significance of the differences in the median requires, however, the differences to be normally distributed. From the top panels in figures 3 and 4 we can see that this is not the case. Hence, doing a t-test does not make sense. Therefore, we prefer to show the actual distributions so the reader can see the differences instead of relying on a t-test that violates the normal distribution assumption.

- *How does the gridding of 50x50 km tiles influence the results? Why is the point-based method the only method missing data in the trunk of Jakobshavn isbrae? Why is there no data for all the methods east of the line from 79fjord to Helheim of figure 3? The data coverage seems different in figure 4.*

The difference in data coverage between figures 3 and 4 originates from a minor issue in the plotting which has been solved in the revised manuscript. The fact that we lack data in the trunk of the Jakobshavn isbrae is caused by the fact that outliers are removed (please see attached figure).

[Figure]

*Figure 1. Median (upper panels) and median absolute deviation (lower panels) with all points (both with and without outliers) included.*

Regarding our choice to use 50x50 km, this was a compromise between visualisation and robustness of the statistics. Switching to 25x25 km tiles (please see map below) results in more empty tiles (because we lack data points) and enhanced pepper and salt effect, but the main conclusions remain the same.

[Figure]

*Figure 2. Comparison between using a 25 km tile (upper figure) and 50 km tile (lower figure).*

- *Table 1, for the discussion it would be informative to also have the arcticDEM vs. ICESat statistics.*

We agree this can be informative. We can't however add this to Table 1 as it would become too large. We have added a table to the Appendix.

- *Table 2, The surface penetration biases may relate to the retracking threshold chosen. How is the statistics changing between valid choices? (10%-90%)*

Agreed. This was our main motivation for the sensitivity analysis presented in section 4.4. In the revised manuscript, we have used a broader range of thresholds and discussed the issue in more detail.

*Minor comments:*

*L2: What is "assessing snow/ice anomalies"*

This statement refers to the 2012 melt observed by Nilsson et al. (2015). It has been rephrased in the revised manuscript.

*L14: Is the difference between the 1cm and 0cm bias between LEPTA and ESA significant and therefore needs to be differentiated from the performances of LEPTA. I think that it is the standard deviation that is the important measure.*

(Now L13-19) Agreed. In our view, the results show that the median differences need a careful interpretation. At the same time, the results suggest that part of the observed bias is due to the applied slope correction method. Therefore, more quantitative details added for clarification.

*L16: Reformulated: "we recommend the LEPTA method for obtaining... "*

(Now L19) This has been changed in the revised manuscript.

*L17: What is complex topography? The work is done in the LRM area.*

(Now L20) Rephrased into 'especially in regions with complex topography.' to 'especially towards the margins of the LRM coverage where the surface slopes increase.'

*L23: Concerning elevation change, you could add a reference to (Hurkmans, Bamber, and Griggs, 2012)*

This has been added.

*L51-52: Suggestion to move this to the last part of the introduction.*

Agreed.

*L79: Why not use the official releases of the downsampled ArcticDEM?*

(Now L87-93) Because in this way we can define the resolution ourselves.

*L84: Is it ATL03 or ATL06 being used? From the link it seems to be ALT06, is there a bias of using the downsampled product?*

(Now L99) It is indeed ATL06, which has a known geolocation accuracy/bias of less than 10 m (https://nsidc.org/sites/nsidc.org/files/technical-references/ICESat2_ATL06_Known_Issues_v005.pdf, last access Dec. 3, 2021). In the revised manuscript, we have added these details.

*L89: This should be moved to the acknowledgment.*

This has been changed in the revised manuscript.

*L92: This sentence needs to be elaborated.*

(New Section 3.1.1) Agreed. This has been changed in the revised manuscript.

*L134: 8x8 km seems small. When looking at the SARIn retracked data from ESA, relocation distances of up to 12 km from the nadir point can easily be found.*
Agreed. In line with the point-based method, we have enlarged it to 14.393x14.393 km.

*L157: The bias can be evaluated in monthly intervals, but at some of 50x50 km tiles closer to the coast a seasonal difference in the bias is expected. How is this seen in your data?*
This analysis is part of our ongoing work for future publication where we focus on spatio-temporal variability in penetration depth. We agree that it is interesting, but we consider it outside the scope of this manuscript.

*L159: Why use both the nearest and natural neighbor interpolations? You give some reasoning. However, would the two algorithms not converge in your case, and thereby there is no need for adding a user-defined threshold?*
(Now L191-196) We prefer to use a natural neighbour interpolation over a nearest neighbour one. Indeed, doing so allows to compensate for the difference induced by the surface slope as the points never overlap exactly. As pointed out in the manuscript, this requires, however, that the CyroSat-2 data point is surrounded by ICESat-2 points. If this is not the case, we rely on a nearest neighbour interpolation (regarding the use of nearest neighbour, we would like to point to our response to Referee 1). The user-defined threshold the referee points to is required to avoid interpolation over large distances.

*L197-198: I guess the eastside is a result of topography? Could you give some insights into the differences on the east and west-side which will be the reasoning for this reported difference.*
(Now L246) The different median values between the eastern and western sides of the ice sheet are indeed likely due to topography. However, here we would like to interpret it more from the median absolute deviation, as it shows that all methods show larger values close to Jakobshavn where the topography is more complex (even resulting in most outliers that are outside the 10[th] and 90[th] percentiles for the point-based method). These observations have been added to the revised manuscript.

*L219: Having a setup at 50km tiles it would be rather easy to take the time-tagged ArcticDEM tiles into the analyses. This might be a large job to undertake this effort, but one or two tiles would be very informative for the analysis.*
We agree that it might be interesting to compare our results with the time-tagged ArcticDEM tiles, but that is indeed an enormous job and difficult as the tiles are not referenced and still include potential offsets and tilts (https://www.pgc.umn.edu/data/arcticdem/, last access Nov. 23, 2021). There is meta-data available to correct them based on ICESat, but then we de-facto replace the time-tag with the ICESat time-tag. Therefore, we think such a comparison with ArcticDEM tiles does not make sense (without enormous pre-processing of the tiles) and we do consider it an unrealistic experiment.

*L238: The observed change in bias is an important observation, please elaborate on this.*
(Now L305-312) Please see also our response to Referee 1. We have elaborated on this in the discussion of the revised manuscript.

*L244: Please clarify the statement: "relative sensitive". Relative is a difficult word as it might be different for you and me.*
Agreed. The whole section has been rewritten.

*L247: "although not directly visible" please improve the figure.*
Figure 9 has been improved.

*L284: Any insights into why ESA outperforms the other methods?*
(L347-350) Unfortunately, we cannot give a conslusive response.

*L290: Please elaborate on this last statement.*
This statement was not supposed to be in the manuscript and has been removed in the revised version.

**References**

Nilsson, J., et al. (2015), Greenland 2012 melt event effects on CryoSat-2 radar altimetry, Geophys. Res. Lett., 42, 3919–3926, doi:10.1002/2015GL063296.

---

## Author Response (AR2)

**Response to Referees on tc-2021-176**

We appreciate the reviews and comments from both Referees. Please find the response to Referee 2 on pages 1-4, and the response to Referee 3 on pages 5-7.

**Response to Referee 2 on tc-2021-176**

First, we would like to thank the referee for reviewing and commenting the manuscript again. Please find the item-by-item reply below, with the original comments in *italics* and the responses in blue. Simple changes in the text are responded with 'done'. The suggested changes haven been implemented in the revised text.

*First, I would like to applaud the authors for the fully committed response to the initial review and the almost full rewrite of the manuscript. This revised manuscript now stands much stronger and I will recommend it for publication in The Cryosphere without any major revisions.*

*The increased focus on how the applied retracking thresholds is resulting in surface retrievals removes most of my initial concerns of possible elevation biases due to the ratio between volume and surface scattering. The method is shown for a warm year (2019) and seems to behave very nicely under these difficult conditions. I would suggest future work to look at the 2012 melt event, to see how the method is coping with the change in scattering regime in the LRM area during July. I guess from your comment to L157 of the first review this is also on your mind, and I only can support this.*

*One thing which is a bit puzzling is the good agreement between the ActicDEM and ICESat-2 (<2 cm), and then the 24 cm bias between the LEPTA and ActicDEM. This points to biases introduced in the inter-comparison of the data. Maybe something worth to be mentioning.*

Thanks for pointing us to this issue. In the previous version of the manuscript, the comparison between ArcticDEM and ICESat-2 was conducted over the entire Greenland ice sheet, that is over both the CryoSat-2 LRM and SARIn zones. In the revised version, we have conducted the comparison only over the LRM zone. The median difference is 21 cm and the standard deviation 1.13 m, which are more coherent with the LEPTA-ArcticDEM difference.

*I still find the 50x50 km grid for Greenland too coarse and as the figures are already made, I would suggest changing to the 25x25 km as the "salt and pepper" is giving insight on the retracker behavior.*

This is changed in the revised manuscript (Fig. 3, Fig. 6, Fig. C1 and Fig. C2).

*This leaves me with minor corrections:*

*L15-20: It is hard to judge the better performance as depending on the measure different methods perform similarly. Suggest simplifying and/or maybe only looking to the lower standard deviation. Or perform a significance test to see if you have statistically significant better performance. I know you argue against the t-test in the reply to my previous request, but tests might be more suitable. Otherwise, you should not use significant in L350 as you do.*

We agree that it is hard to judge the better performance. We changed the formulation in the abstract from:

'Benchmarking of the LEPTA method to the slope- and point-based method based on CryoSat-2 LRM acquisitions over Greenland in 2019 shows that heights obtained by LEPTA outperform the other methods when compared to ICESat-2 observations, both in the flat, interior regions of Greenland and in regions with more complex topography.'

to (now L9-12):

'Benchmarking of the LEPTA method to the slope- and point-based method based on CryoSat-2 LRM acquisitions over Greenland in 2019 shows that, when compared to ICESat-2 observations, the method has a stable performance both in the flat, interior regions of Greenland and in regions with more complex topography.'

We also changed the sentence:

'Although ESA Level-2 products and the point-based method have good performance in either median and median absolute deviation, LEPTA stably outperforms the other methods.'

to (now L17):

'Although ESA Level-2 products and the point-based method have good performance in either median and median absolute deviation, LEPTA shows a good performance in terms of both metrics.'

We also removed the word significant in the conclusions section. The sentence is changed from:

'Regionally, the impact may be more significant. In particular, we observe significant changes in the vertical and horizontal position of the impact points towards the margins of the LRM zone.'

to (L349):

'Regionally, the impact may be larger. In particular, we observe changes up to 1.46 m in the vertical and 231 m in the horizontal position of the impact points towards the margins of the LRM zone.'

*L28: 717km should be the real orbit of CS2.*
The numbers have been changed accordingly.

*L35: Suggest removing references as they already are mentioned above.*
Done.

*L47: (and multiple other places) This is a matter of style but suggests removing reference to a section in the manuscript.*

They have all been removed.

*L51-56: A matter of style but suggest to only keep:*
*"To assess the performance of the LEPTA method, we apply it to all CryoSat-2 LRM acquisitions over Greenland in 2019 and benchmark it to the slope and point-based methods by comparing it with laser altimeter ICESat-2 height measurements."*
We prefer to include an outline of the paper as we think it helps the reader.

*L63: Baseline E is now available, however, if you look at the release documentation this baseline does not suggest having many changes for ice sheets, and therefore are your results still valid. This should be mentioned.*
This has been added to the Discussion section (now L381-383).

*L75: You are properly right about the threshold but the reference to Bamber is rather old concerning baseline thresholds. Please clarify.*
Unfortunately, we could not find an updated reference. Hence, we decided to remove the statement.

*L100: (and L361) The link should not be the reference please refer in a more conventional way (author for the document).*
(Now L99) Done.

*L182: Remove sentence, section 3.2 is the next section.*
Done.

*L220: -> "from -7.5 to 2.5 meters"*
(Now L217) Done.

*L241: Please enlighten the reader with your explanation of this difference.*
Unfortunately, we lack such an explanation. When we compare ICESat-2 and ArcticDEM we observe a similar pattern as observed when comparing LEPTA and the point-based method to ArcticDEM. Likely, it is a combination of elevation change signal and errors in the correction methods.

*L267: But they are fitting internal despite the timing between ArcticDEM and ICESat-2*
Please see the reply of the general comment.

*L279: remove "As mentioned in Section 3.3,"*
Done.

*L351 "< 1 dm" -> "< 0.1 m" or "< 10 cm"*
(Now L348) Changed to 0.1 m.

*L392: Why is "novel" removed here, it is still in the abstract.*
The 'novel' in the abstract has been removed.

*L421 "B1-B1" -> "B1 and B2"*

(Now L410) Done.

*L421:   The underlying skewness is hard to see. The figures should be improved. If the purpose of not showing the distributions in one panel is to show min and max values, these may be mentioned in the caption. The range should be centered to something like 10 meters.*

The figures are now displayed between [-10 m, 10 m] and the captions of Figs. B1 and B2 have been changed from:

'Full probability distribution functions of heights between CryoSat-2 and ArcticDEM derived from a) ESA L2I, b) slope method, c) point-based method and d) LEPTA.'

to

'Probability distribution functions of heights between CryoSat-2 and ArcticDEM derived from a) ESA L2I, b) slope method, c) point-based method and d) LEPTA centred between [-10 m, 10 m]. To clearly show minimum and maximum values (values displayed with arrows), the curves are not displayed in the same panel.'

**Response to Referee 3 on tc-2021-176**

First, we would like to thank the referee for reviewing and commenting the manuscript. Please find the item-by-item reply below, with the original comments in *italics* and the responses in blue. The suggested changes have been implemented in the revised text. Simple changes are responded with 'corrected' or 'done'.

*General comments:*

*Li et al present a new method for correcting slope errors in pulse-limited radar altimetry data, based upon using information in the leading edge and a high resolution DEM to more accurately locate the point of closest approach. The authors provide an extensive validation comparing their product and others to ICESat-2 measurements and a DEM, and a series of sensitivity analyses to explore the robustness of their approach.*

*I enjoyed reading the manuscript, the method detailed here offers genuine improvements over those currently used, and this is well demonstrated by the authors here by their analyses. The authors have done a considerable amount of work to address previous reviewers comments, which have improved the manuscript considerably. I only have one minor comment regarding the use of different retracking thresholds, which I would appreciate if the authors could address:*

*As I understand it from P3 L75, the ESA L2 product is using a 25% OCOG retracker, and the authors are using a 20% threshold – is this an additional source of height difference when comparing LETPA and ESA L2 heights, and has this been accounted for? To my knowledge it does not seem to have been discussed anywhere in the text (if it is indeed something worth considering). Is it possible to disentangle height differences due this choice of retracking threshold from the different slope corrections?*

As pointed out by Referee 2, the reference to Bamber 1994 is quite old. We searched for an updated reference but could not find any. Therefore, we removed the statement regarding the threshold used in the ESA L2 product. To answer the referee, from our sensitivity analysis in Section 4.4, an OCOG threshold located between 30% and 40% can indeed result in a near-zero CryoSat-ICESat difference for the slope method. However, as mentioned in L345 of the revised manuscript, we cannot tell exactly why our implementation of the slope method differs largely (20 cm in median) from the ESA L2I products. Therefore, without knowing the exact retracking process of ESA, we cannot exclude that the difference between LEPTA and ESA L2I products is due to different OCOG thresholds, and the statement on L348 'Detailed analysis (not shown in this paper) shows that the differences cannot be explained by the fact that in our study we use another DEM as well as a different OCOG retracker threshold' has been changed ('as well as a different OCOG retracker threshold' is removed).

*Specific comments:*

*P1 L4 – 'Therefore different correction methods have been developed ranging from the slope method to the point-based method' – to me this seems to have little meaning in the context*

*of the abstract as they are technical terms explained later in the paper, I suggest either expanding on their meaning or this sentence could be easily removed*
We reformulated the sentence.

*P1 L9 – 'within range' > 'within the range'?*
(Now L8) Corrected.

*P1 L9 – the phrase 'impact point' is used here and throughout the paper and seems imprecise language to me, and does not seem to be properly defined anywhere in the text. I suggest the authors more clearly explaining what they mean by this where it first appears in the main text*
The phrase impact point is used by, e.g., Roemer et al. (2007) and Levinsen et al. (2016). It refers to the point from which the radar wave reflected. We have added this 'definition' to the main text (L32).

*P1 L31 – the way these sentences are worded leaves it unclear as to which are 'slope' and are 'point-based' methods, I suggest re-wording to make this more clear*
(Now L33) We reformulated the sentence. After stating that 'the most widely used methods involve both a correction to the height as well as a relocation of the satellite measurement location from nadir to the expected impact point' we say that 'Two implementations of this so-called "relocation method" are known as the "slope" and the "point-based" method (Levinsen et al., 2016).'

*P1 L41 – 'footprint that illuminates the terrain' > 'footprint illuminated by the satellite on the terrain'*
(Now L42) Done.

*P1 L50 – 'begin' > 'beginning'*
Done.

*P2 L67 – 'proper' > 'appropriate'*
Done.

*P3 L75 – not sure what is meant by 'removed in case', do the authors mean 'removed if'?*
True. All misuses of 'in case' have been changed into 'if'.

*P7 L167 – 'firn-air' interface may be more appropriate than 'snow-air' for ice sheet interiors*
(Now L165) Done.

*P7 L186 – I'm not sure what the authors mean by 'conceptual assessment' here*
We removed the sentence. We wanted to stress that the assessment based on Arctic DEM cannot be considered as a validation. This is, however, already stated in L186-187 of the revised manuscript.

*P8 L193 – Again I'm not sure what the authors exactly mean by 'In case', do they mean 'if'?*
All misuses of 'in case' have been changed into 'if'.

*P11 Fig3 – It is more work so I will not ask the authors to do it for this paper, but in future they may wish to consider looking at the differences for each technique as a function of slope, this may in particular highlight the benefits of their method*

Thanks for the suggestion. This may indeed be interesting. But so far we make no changes to the manuscript.

*P19 L348 – If the detailed analysis is 'not shown in this paper' could the authors please provide more information as to where it is shown? There are other instances of this in the discussion section. Is this information not included in the appendices?*

They are included in the response to the referees in the previous major review. However, adding the results to the appendices will be an overloading information. We agree to remove the 'not shown' statements in L253 and L354, but as for L348, we prefer to not remove it as it keeps the logic of the sentence.

---

## Author Response (AR3)

We would like to thank the Editor for the thorough check of the manuscript again and for the suggestions for further improving the manucript. We have corrected it according to most of the suggestions. However, for the suggestion regarding L336 (millimetre-level -> millimeter-level), please refer to https://en.wikipedia.org/wiki/Millimetre.